# Chondrocyte fatty acid oxidation drives osteoarthritis via SOX9 degradation and epigenetic regulation

Zixuan Mei [1,2,6], Kamuran Yilamu[1,2,6], Weiyu Ni[1,2,6], Panyang Shen[1,2,6], Nan Pan[1,2], Huasen Chen[1,2], Yingfeng Su[1,2], Lei Guo[3], Qunan Sun[4], Zhaomei Li[5], Dongdong Huang[3], Xiangqian Fang [1,2], Shunwu Fan [1,2] ✉, Haitao Zhang [1,2] ✉ & Shuying Shen [1,2] ✉

Osteoarthritis is the most prevalent age-related degenerative joint disease and is closely linked to obesity. However, the underlying mechanisms remain unclear. Here we show that altered lipid metabolism in chondrocytes, particularly enhanced fatty acid oxidation (FAO), contributes to osteoarthritis progression. Excessive FAO causes acetyl-CoA accumulation, thereby altering protein-acetylation profiles, where the core FAO enzyme HADHA is hyperacetylated and activated, reciprocally boosting FAO activity and exacerbating OA progression. Mechanistically, elevated FAO reduces AMPK activity, impairs SOX9 phosphorylation, and ultimately promotes its ubiquitination-mediated degradation. Additionally, acetyl-CoA orchestrates epigenetic modulation, affecting multiple cellular processes critical for osteoarthritis pathogenesis, including the transcriptional activation of MMP13 and ADAMTS7. Cartilage-targeted delivery of trimetazidine, an FAO inhibitor and AMPK activator, demonstrates superior efficacy in a mouse model of metabolism-associated post-traumatic osteoarthritis. These findings suggest that targeting chondrocyte-lipid metabolism may offer new therapeutic strategies for osteoarthritis.

Osteoarthritis (OA) is the most prevalent age-related degenerative joint disease globally and is characterized by cartilage degradation, synovial inflammation, osteophyte formation, and subchondral bone alterations[1]. While OA affects entire joints, its defining pathological feature is progressive cartilage destruction and extracellular matrix (ECM) loss, resulting from imbalanced cartilage homeostasis governed by anabolic–catabolic-signaling interactions within chondrocytes[2]. Excessive catabolic signaling prompts chondrocytes to generate ECM-degrading enzymes, including matrix metalloproteinases and a

disintegrin and metalloproteinase with thrombospondin motifs (ADAMTS), culminating in cartilage degradation. Under pathological conditions, anabolic signaling fails to maintain the chondrogenic phenotype. As a critical chondrogenic transcription factor[3], SRY-box transcription factor 9 (SOX9) maintains cartilage homeostasis by triggering chondrocytes to express cartilage ECM components, including type II collagen (COL2A1) and aggrecan (ACAN)[4,5]. Deficiency in SOX9 results in proteoglycan loss in healthy cartilage and exacerbates post-trauma cartilage erosion[6]. The transcriptional activity of

[1]Department of Orthopaedic Surgery, Sir Run Run Shaw Hospital, Zhejiang University School of Medicine, Hangzhou, China. [2]Key Laboratory of Musculoskeletal System Degeneration and Regeneration Translational Research of Zhejiang Province, Hangzhou, China. [3]Pooling Institute of Translational Medicine, Hangzhou, China. [4]Department of Medical Oncology, the Second Affiliated Hospital, Zhejiang University School of Medicine, Hangzhou, China. [5]Department of Geriatrics, Xiaoshan Geriatric Hospital, Hangzhou, China. [6]These authors contributed equally: Zixuan Mei, Kamuran Yilamu, Weiyu Ni, Panyang Shen. ✉e-mail: shunwu_fan@zju.edu.cn; zhtmushi@163.com; 11207057@zju.edu.cn

SOX9 is regulated by post-translational modifications (PTMs)[3,4,7]. A comprehensive understanding of SOX9 expression regulation and its PTMs might help identify suitable intervention targets.

Obesity is a preventable risk factor significantly contributing to OA onset and progression, with substantial healthcare burdens due to its increasing prevalence[8]. While obesity is associated with increased OA risk[9–11], excessive joint loading only cannot account for this[12], as the risk persists in non-weight-bearing joints[13]. Fat loss, rather than body-weight loss, closely correlates with OA improvement[14,15]. Furthermore, removal of the intra-articular adipose tissues beneficially alters the pathogenesis of OA[16], highlighting the metabolic aspect of adipose tissue in the local environment as a significant contributor in OA pathophysiology. Adipokines and lipid metabolites are significantly altered in individuals with obesity and thus may contribute to OA development. Although the roles of adipokines like leptin and adiponectin in OA are well-studied, their impacts remain intricate[17,18]. Early studies have shown cholesterol's effect on OA pathogenesis through the CH25H-CYP7B1-RORα axis by inducing matrix-degrading enzymes[19]. However, studies on serum and cartilage samples from patients with OA have also suggested a significant association with other lipid metabolites, such as fatty acids[20–22]. Previous research indicates that fatty acid accumulation in osteoarthritic cartilage occurs early in the disease before histological changes become apparent[17,23], indicating its potential driving effects on OA development. However, the precise mechanisms, including how fatty acid accumulate in cartilage and trigger OA, remain unclear. Moreover, the effects of accumulated fatty acids on chondrocyte lipid metabolism, particularly the utilization of fatty acids, remain largely unknown.

Acetyl-coenzyme A (acetyl-CoA) participates in multiple intracellular metabolic processes (including lipid metabolism) and affects protein acetylation[24,25], playing key cellular roles in metabolic activity and epigenetic regulation[26,27]. Lysine acetylation is common among metabolic enzymes involved in glycolysis, gluconeogenesis, the tricarboxylic acid cycle, fatty acid oxidation (FAO), the urea cycle, nitrogen metabolism, and glycogen metabolism and is regulated by metabolic fuel concentrations, thereby modulating enzyme activity and stability[28]. However, the acetylation patterns of metabolic enzymes in chondrocytes and their implications in OA pathophysiology remain to be explored. Our data revealed a notable elevation in fatty acid and acetyl-CoA levels in obesity-associated osteoarthritic cartilage, coinciding with altered protein-acetylation profiles. Therefore, we delved into the intricate molecular mechanisms underlying lipid metabolism in the context of obesity-related OA (ObOA) with the aim of identifying promising therapeutic targets.

## Results

### Synergistic effects of lipid stress and inflammatory-mechanical factors on chondrocyte fatty acid uptake and OA progression

To address the role of the metabolic aspect of adipose tissue in OA pathophysiology in vitro, we cultured primary mouse chondrocytes with conditioned medium (CM) from 3T3-L1-adipocytes (adipo-CM; Fig. 1a). Micromass co-culture demonstrated that adipo-CM significantly reduced chondrocyte ECM deposition (Fig. S1a). Adipocytes function largely by secreting not only various adipokines but also producing substantial amounts of fatty acids via lipolysis[29]. To identify the major factor impacting chondrocyte ECM homeostasis, we supplemented various adipokines and free fatty acids (FFAs) to CM from 3T3-L1 fibrocytes (Fibro-CM) and co-cultured them with chondrocytes, which revealed that FFA significantly reduced ECM deposition in cartilage more effectively than any other adipokine (Fig. S1b). 3D agarose culture further validated that Fibro-CM + FFA dramatically reduced the thickness of pericellular matrix, comparable to adipo-CM (Fig. 1b). Moreover, FFA treatment of mouse chondrocytes dose-dependently downregulated anabolic factors (ACAN, COL2A1, and SOX9) and upregulated catabolic factors (MMP13 and MMP3, Fig. 1c), while having

no significant effect on chondrocyte viability (Fig. S1c), indicating its pivotal role in regulating cartilage ECM metabolism. To determine whether chondrocytes can uptake FFA and elucidate its relationship to the extracellular lipid environment, we performed a fatty acid uptake assay. FFA pretreatment significantly enhanced FFA uptake compared to the BSA control (Fig. 1d). We then examined fatty acid uptake-related genes and found that Cd36, Fatp5, Fabp3, and Lipe (involved in lipolysis) were upregulated in chondrocytes following FFA treatment (Fig. S1d). Previous studies have suggested that de novo lipogenesis is elevated in OA cartilage[30], potentially contributing to intracellular lipid accumulation. To investigate whether OA-associated inflammatory and mechanical factors also regulate fatty acid uptake, we examined their effects on the previously identified genes. IL-1β and TNF-α, major proinflammatory factors in OA pathophysiology[31], upregulated Lipe and Fatp5 expression (Fig. S1e), while cyclic tensile stress, which mimics a mechanically induced osteoarthritic phenotype[32], elevated Fatp5 and Fabp3 expression (Fig. S1f). Furthermore, we treated chondrocytes with FFA and/or IL-1β, and observed a significant increase in the number and size of cellular-lipid droplets when treating with either FFA or IL-1β, with maximal synergy after combined treatment (Fig. 1e).

To explore the impact of fatty acids on OA progression in vivo, we generated a high-fat diet (HFD)-induced obesity mouse model (Fig. 1f), which has been reported to result in dramatically increased fatty acid levels in serum and multiple organs[23,33–36]. Male C57Bl6/J mice subjected to a 16-week HFD exhibited increased body weights, visceral adiposity, and impaired glucose tolerance (Fig. S1g, h). Xylene-free Bodipy 493/503 and Nile Red staining showed that lipids significantly accumulated in mouse cartilage after feeding with HFD (Fig. 1g). To more clearly elucidate the interplay between inflammatory-mechanical factors and obesity-related lipid stress and investigate its role in OA, we generated destabilization of the medial meniscus model (DMM) using HFD-fed mice, obtaining HFD-DMM mice as a model of metabolism-associated post-traumatic OA[19]. This combination demonstrated increased lipid accumulation in the cartilage of normal diet (ND)-fed mice post-DMM surgery, comparable to that in HFD-fed mice that underwent a sham operation, with maximal synergistic effects observed in the cartilage of HFD-DMM mice. To further validate this finding, we conducted lipidomic profiling using cartilage samples from the HFD-DMM model and reveled a substantial increase in saturated and unsaturated long-chain FFAs (Fig. S1i) and corresponding glycerolipids (GLs, Fig. S1j). Mild FFA and GL elevations were also noted in ND-fed mice post-DMM surgery and in HFD-fed mice following sham operations. A previous study indicated that cholesterol accumulation in chondrocytes which may play roles in OA progression[19]. However, our lipidomic data reveal that cholesterol levels remained relatively stable (Fig. S2k). We therefore explored the possible association between fatty acid levels within chondrocytes and OA pathogenesis and found that mice fed HFD exhibited increased severity of OA caused by DMM (Figs. 1h-i and S1l–m), coincident with the increase in cellular fatty acids levels. Moreover, normal diet-fed mice that received FFA arthrocentesis exhibited significantly more severe cartilage degeneration than the BSA control mice (Fig. S1n). Similar trends were further validated in human cartilage samples (Figs. 1j-l and S2a).

To investigate whether local environment provides excessive FFAs during OA development, we retrospectively analyzed lipid contents in human synovial fluid, considering the avascular nature of cartilage and its dependence on nutrient diffusion from synovial fluids[37]. A total of 120 patients with OA were categorized into three groups (mild, moderate, and severe), based on Kellgren–Lawrence (KL) grades or into two groups based on body mass index (BMI) levels (Fig. S2b). When categorized by KL grade, no significant differences were observed in sex, height, body weight, or BMI between groups, although age varied. Similarly, no notable differences were identified in age, sex, or height when categorized by BMI, although body weights

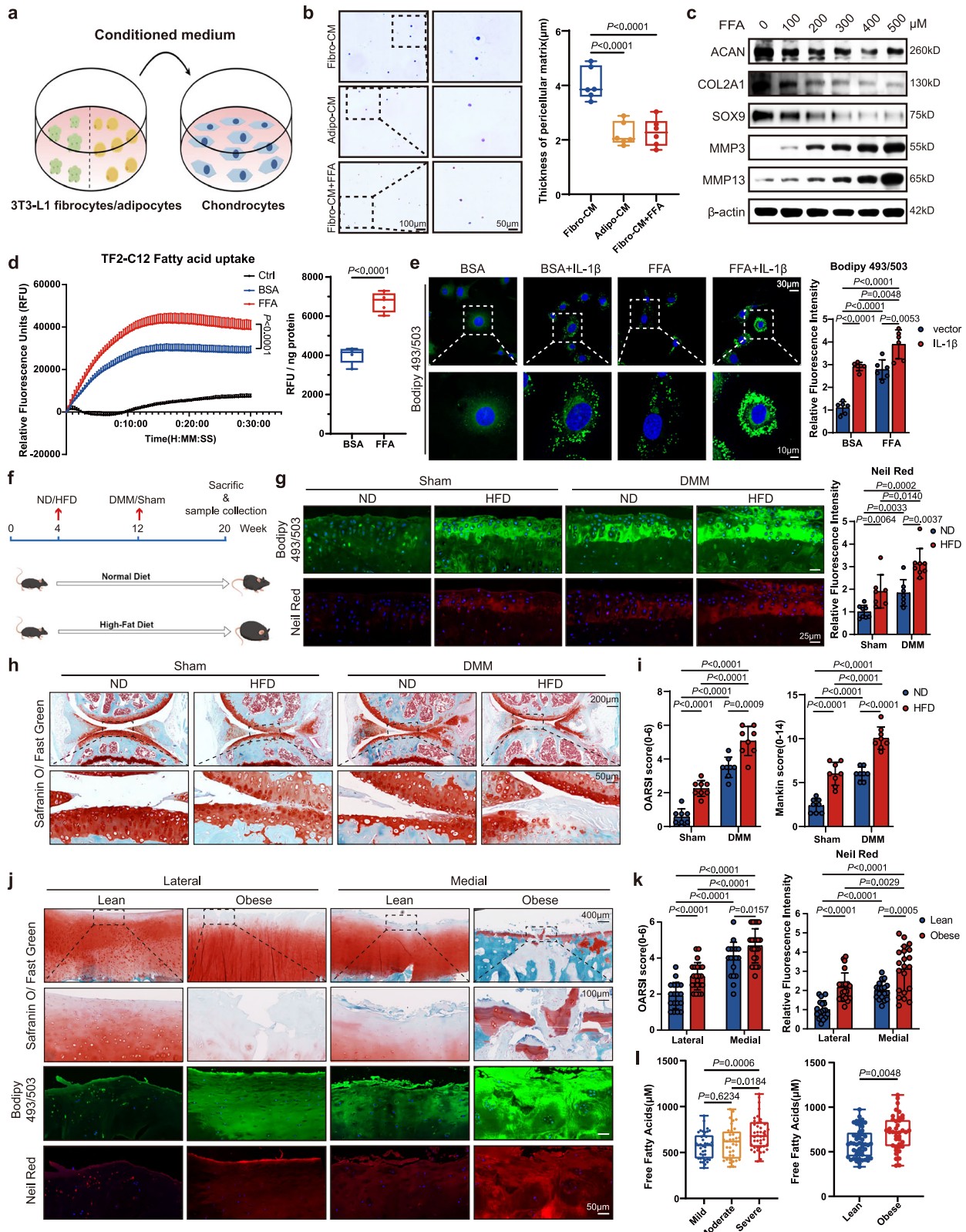

and BMI significantly differed between groups as expected. Total FFA levels within synovial fluids increased with increasing OA severity and were higher in individuals with obesity (Fig. 1m), whereas total triglyceride, total cholesterol, and free cholesterol levels showed no clinical significance (Fig. S2c, d). Together, these results indicate that OA-associated inflammatory-mechanical factors and lipid stress synergistically stimulate fatty acid uptake in chondrocytes.

## Fatty acid-derived acetyl-CoA activates chondrocyte FAO via HADHA acetylation at lysine 728

Fatty acids enter into cells and undergo mitochondrial FAO, which not only powers oxidative phosphorylation (OXPHOS) for energy generation but also yields acetyl-CoA, a pivotal metabolic intermediate connecting multiple metabolic reactions and an exclusive donor of acetyl groups that are essential for protein acetylation[25]. To investigate the

**Fig. 1 | Synergistic effects of lipid stress and inflammatory-mechanical factors on chondrocyte fatty acid uptake and OA progression. a** Illustration of the CM system, created by figdraw.com. **b** 3D-agarose culture and alcian blue staining of primary mouse chondrocytes cultured in fibro-CM, adipo-CM, or fibro-CM supplemented with 200 μM FFA (n = 6). **c** Immunoblot detection of ACAN, COL2A1, SOX9, MMP3, and MMP13 in primary mouse chondrocytes treated with FFA. **d** Fatty acid uptake in mouse chondrocytes pretreated with FFA or BSA control (n = 5). **e** Bodipy 493/503 staining of mouse chondrocytes treated with BSA or FFA in combination with or without IL-1β (n = 6). **f** Illustration of diet change and destabilization of the medial meniscus (DMM) surgery or sham operation in male C57BL/6J mice (n = 8), created by figdraw.com. **g** Bodipy 493/503 and Nile red staining of knee joints from mice indicated in this figure (**f**). (n = 7 for ND + DMM and HFD +Sham groups; n = 8 for ND+Sham and HFD + DMM groups). **h–i** Safranin O/Fast green staining of knee joints from mice indicated in this figure (**f**) (n = 8).

**j–k** Safranin O/Fast green staining and Bodipy 493/503 or Nile red staining of human knee cartilage tissues from patients after total-knee arthroplasty, categorized based on BMI values (n = 18 for Lean groups, n = 22 for obese groups, biologically independent samples). **l** Total FFA levels in synovial fluids from patients with mild (n = 40), moderate (n = 36), and severe (n = 44) OA (Left), and from patients classified as Lean (n = 52) or with obesity (n = 40) (Right). **b, d** right and **l** Data are minimum to maximum: each point represents an individual biological replicate, box from 25th to 75th percentiles; center line at median; tukey whiskers, as determined via one-way ANOVA followed by Dunn's or Tukey's multiple-comparisons test (**b** and **l**, left) or two-tailed t-test (**d**, right), or nonparametric two-tailed Mann–Whitney test (**l**, right). **d**, left and **e–k** Data are mean ± SD, as determined by two-tailed t-test or nonparametric two-tailed Mann–Whitney test. The immunoblotting data are representative of three independent experiments. Source data are provided as a Source Data file.

impact of fatty acid accumulation on chondrocyte acetyl-CoA levels, we orally administered [13]C- labeled fatty acids daily to HFD-DMM mice for 2 weeks before harvest (Fig. S3a). Ultra-performance liquid chromatography-tandem mass spectrometry (UPLC-MS/MS) revealed significantly increased [13]C-labeled acetyl-CoA in cartilage samples from HFD-DMM mice compared to that in the ND-DMM group (Fig. 2a). Although HFD alone led to a slight increase in [13]C-acetyl-CoA, the difference was not statistically significant (Fig. S3b). Enzyme-linked immunosorbent assays confirmed this elevation in HFD-DMM mice (Fig. S3c), which was also observed in mouse chondrocytes cultured with adipo-CM or FFA (Fig. S3d, e). Western blotting with an anti-acetyl lysine antibody showed higher total protein-acetylation levels in cartilage tissues from the HFD-DMM group (Fig. 2b), consistent with findings in mouse chondrocytes treated with FFA (Fig. S3f), indicating that acetyl-CoA derived from fatty acids orchestrates chondrocyte-protein acetylation dynamics. To decipher the landscape of chondrocyte-protein acetylation induced by HFD-DMM, we employed 4D label-free quantitative proteomics and acetylome analysis of knee cartilage samples from HFD-DMM and ND-DMM mice. The proteomic analysis clearly distinguished these two groups (Fig. S3g) and identified 4712 proteins, with 3952 being quantifiable and 72 showing statistically significant differences (false-discovery rate <0.01, fold-change >1.5). A volcano plot illustrated downregulated proteins involved in lipid synthesis (Fasn, Acc1, Acss2, and Thrsp) and upregulated proteins associated with fat catabolism (Ech1, Acox1, Ilvbl, and Pdk4), cholesterol efflux (Apoe, Apoa4, and Pltp), fatty acid uptake (Fabp3), and ECM metabolism (Pxdn) in response to HFD + DMM (Fig. S3h). The upregulation of APOE and APOA4 was further validated in vitro, where chondrocytes were stimulated with FFA (Fig. S3i). The acetylome analysis uncovered 4759 unique acetylation sites on 1731 proteins, with 3005 quantified sites. Importantly, 544 acetylated sites on 379 proteins were statistically significant (false-discovery rate <0.01, fold-change >1.5) after normalization to total-protein levels. Gene Ontology (GO) analysis revealed that upregulated acetylated sites were enriched in biological processes related to lipid metabolism (Fig. S3j), with molecular-function enrichment strikingly pointing to the mitochondrial trifunctional protein (TFP) subunit α, HADHA (Fig. 2c), which catalyzes the final three steps of fatty acid β-oxidation. Thus, we analyzed all differentially acetylated proteins and sites involved in lipid metabolic processes, finding that a substantial portion were associated with fatty acid β-oxidation (Fig. 2d). These findings suggest that chondrocytes sense and adapt to lipid stress by altering lipid metabolism, specifically mitochondrial FAO, via protein acetylation.

Accordingly, we considered the possibility that fatty acid-derived acetyl-coA acetylates HADHA in a substrate dependent manner and modulates its enzymatic activity. To test this, we first treated C28/I2 cells expressing HADHA-Flag with FFA or two HDAC inhibitors, trichostatin A (TSA) and nicotinamide (NAM), and found that acetylation of HADHA was markedly enhanced by each treatment (Figures S3k).

Given that non-enzymatic protein acetylation potentially depends on substrate abundance, particularly in alkaline environments, such as the mitochondrial matrix[38], we conducted in vitro acetylation assays by incubating purified Flag-tagged HADHA with increasing concentrations of acetyl-CoA, demonstrating a dose-dependent increase in acetylated HADHA (Fig. S3l). We further found that HADHA enzymatic activity increased in FFA or TSA + NAM-treated C28/I2 cells (Fig. 2e), suggesting a positive regulation of FAO by HADHA acetylation. Real-time monitoring of the cellular oxygen-consumption rate (OCR) following FFA pretreatment demonstrated increased basal and maximal OCRs. Notably, treatment with etomoxir (Eto), an FAO inhibitor targeting carnitine palmitoyltransferase 1, reduced maximal OCR levels, indicative of elevated OXPHOS fueled by FAO (Fig. S3m). Although etomoxir treatment slightly lowered basal OCR levels, it did not fully suppress the FFA-induced increase, which could be attributed to compensatory activation of alternative pathways. Therefore, we further analyzed mitochondrial fuel oxidation by sequentially inhibiting glucose, glutamine, and FAO. This approach allowed etomoxir to more effectively suppress basal OCR, particularly in the FFA pretreatment group (Fig. S3n), suggesting that FFA pretreatment enhances FAO capacity in chondrocytes.

To determine the specific acetylation site responsible for enhancing FAO activity, we performed UPLC-MS/MS analysis and found that HADHA acetylation at K728 occurred in FFA-treated C28/I2 cells but not in control cells (Fig. S4a). This acetylation site was the only one in agreement with our acetylome data from mouse cartilage samples (Fig. 2d), which is highly conserved among different species (Fig. S4b), and was validated using a site-specific acetyl-lysine antibody (anti-HADHA K728ac, Fig. S4c). In vitro acetylation assays demonstrated that HADHA K728 acetylation was enhanced with increasing acetyl-CoA levels (Fig. S4d). The level of HADHA K728ac was also increased in cartilage in humans and mice with OA and further amplified by obesity and lipid stress, coincident with in vitro observations (Fig. S4e–f). We performed HADHA-enzymatic activity assays with Flag-tagged HAHDA purified from HADHA-KO C28I/2 cells overexpressing wild-type HADHA or the HADHA K728R mutant (mimicking deacetylation). The K728R mutant showed significantly lower enzymatic activity, effectively abolishing the FFA-induced enhancement of enzymatic activity that was observed in the wild-type (Fig. 2f). Real-time monitoring of the cellular OCR validated the impact of acetylation, showing increased basal and maximal respiration rates and higher ATP production in HADHA-KO C28/I2 cells expressing wild-type HADHA than in those expressing the K728R mutant, which was blocked by Eto (Fig. S4g). Furthermore, mitochondrial fuel oxidation analysis demonstrated that FAO capacity was significantly reduced in HADHA K728R mutant chondrocytes following FFA pretreatment compared to that in wild-type HADHA cells (Fig. 2g). To uncover the structural implications, we constructed computer-homology models for wild-type HADHA, the K728R mutant, and the acetylated K728 variant (K728ac) after docking with the substrate, β-hydroxybutyryl-CoA (Fig. 2h). These models

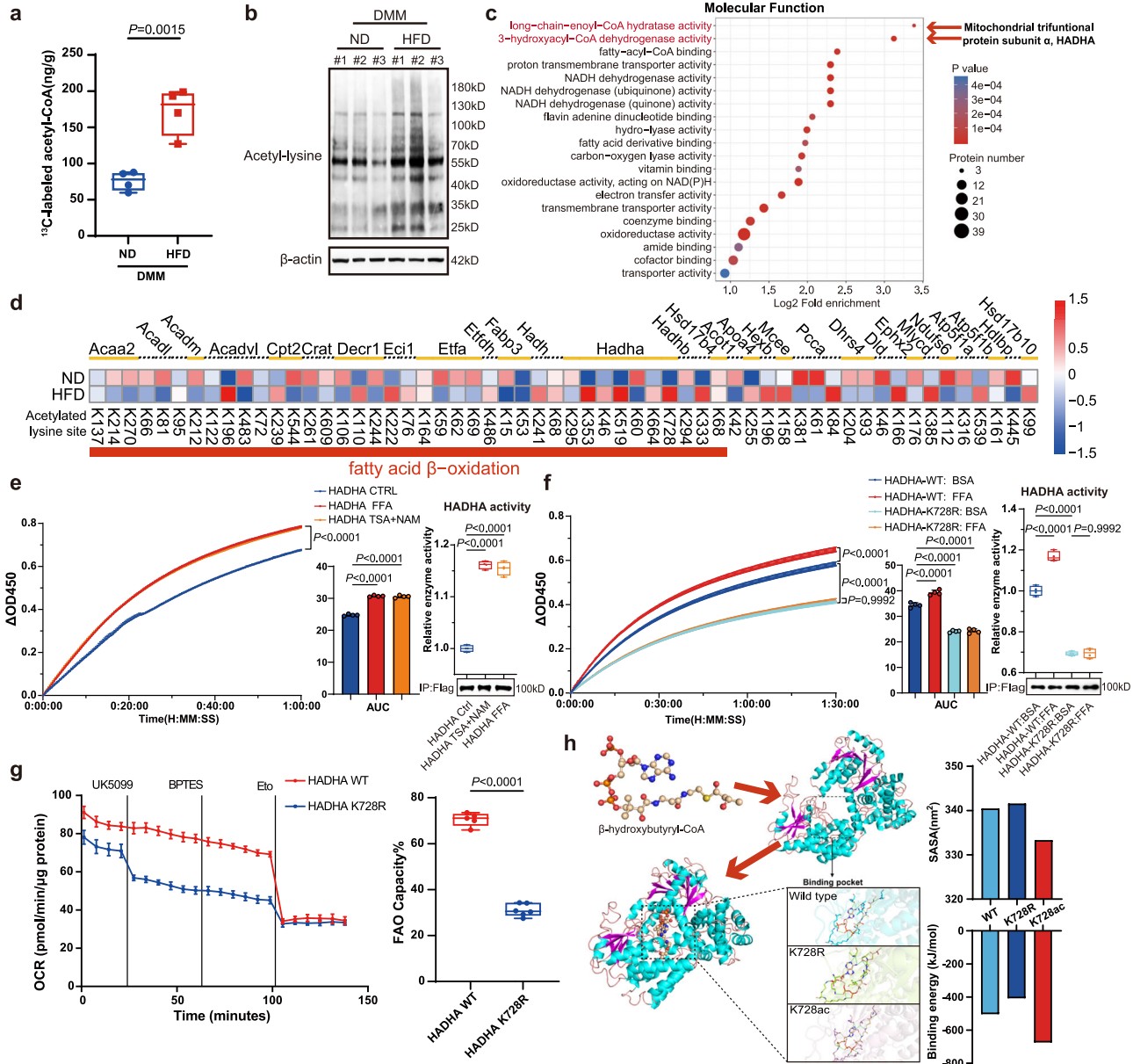

**Fig. 2 | Fatty acid-derived acetyl-CoA activates chondrocyte FAO via HADHA acetylation at lysine 728. a** Quantification of [13]C-labeled acetyl-CoA levels in cartilage samples from mice that underwent DMM surgery, indicated in Fig. S3a ($n = 4$). **b** Western blot analysis of acetyl-lysine from whole cartilage lysate obtained from mice in Fig. 1f. The immunoblotting results are representative of three independent experiments. **c** Bubble chart showing the top 20 enriched MFs among the upregulated acetylated proteins in HFD + DMM versus ND + DMM groups, as determined by a two-tailed Fisher's exact test. **d** Heatmap showing acetylated lysine sites on proteins involved in lipid metabolic process in HFD + DMM versus ND + DMM groups. **e** Real-time relative-enzyme activity of Flag-tagged HADHA purified from C28/I2 cells treated with FFA or TSA + NAM. The area under the curve (AUC) was generated by Prism 9.5.1. Relative HADHA activity was calculated at 20 min ($n = 4$). **f** Real-time relative-enzyme activity of Flag-tagged wild-type HADHA or HADHA K728R purified from *HADHA*-KO C28/I2 cells treated with FFA or BSA. The AUC was

generated using Prism 9.5.1. Relative HADHA activities were calculated at 30 min ($n = 4$). **g** Mitochondrial fuel oxidation analysis. HADHA-KO C28/I2 cells overexpressing wild-type HADHA or HADHA K728R were both pretreated with FFA for 24 h before loading. The cells were then sequentially treated with UK5099, BPTES, and Eto ($n = 5$ for HADHA WT group; $n = 6$ for HADHA K728R group). **h** Molecular interactions of wild-type, K728R, and K728ac HADHA with the substrate. The SASA and binding energy were calculated. **a**, **e** right, **f** right, and **g** right Data are minimum to maximum: each point represents an individual biological replicate, box from 25th to 75th percentiles; center line at median; tukey whiskers, as determined via unpaired two-tailed *t*-test (**a** and **g** right) or one-way ANOVA followed by Dunnett's multiple-comparisons test (**e** right), or two-way ANOVA followed by Tukey's multiple-comparisons test (**f** right). **e** left, **f** left, and **g** left Data are mean ± SD, as determined via one-way ANOVA followed by Dunnett's multiple-comparisons test. Source data are provided as a Source Data file.

revealed notable conformational shifts in HADHA, as evidenced by an increased solvent-accessible surface area (SASA) in K728R and a reduced SASA in K728ac HADHA. Lower SASA values indicated a stabler structure, correlating with higher enzymatic activity. Further insights emerged from the distinct substrate-binding configurations, with the calculated binding energy notably increasing with K728ac HADHA, implying enhanced enzymatic activity. Collectively, these

data indicate that fatty acid-derived acetyl-CoA acetylates HADHA and increases its catalytic activity and rewiring lipid metabolism in chondrocytes to adapt to lipid stress.

**Enhanced FAO exacerbates OA phenotype in vivo and in vitro**
Next, we investigated whether FAO is associated with OA. We silenced the TFP subunits *Hadha* and *Hadhb*, in primary mouse chondrocytes

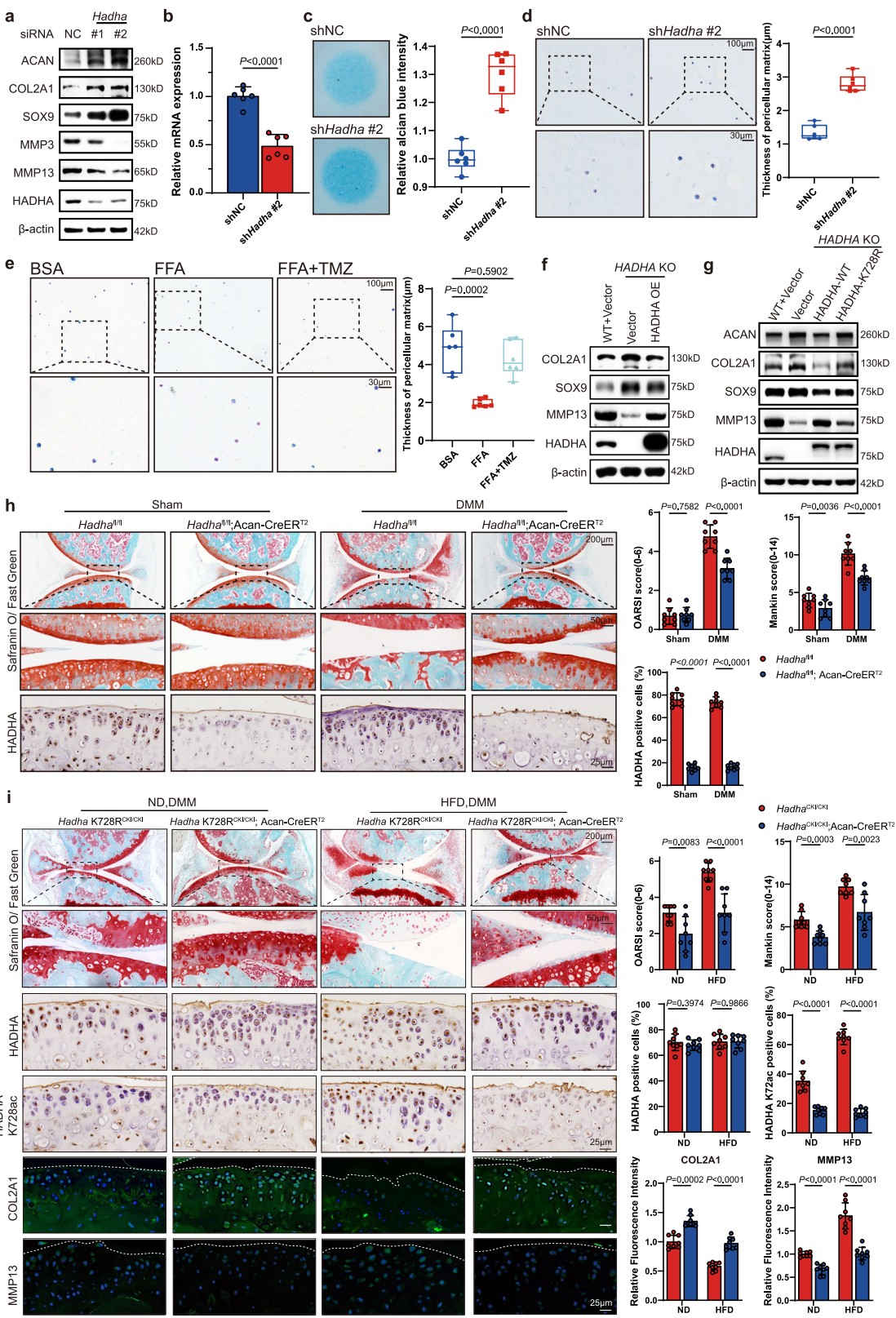

using specific siRNAs and observed upregulation of ACAN, COL2A1, and SOX9 and downregulation of MMP3 and MMP13 (Figs. 3a and S5a). Consistent findings were observed in micromass and 3D-agarose cultures, demonstrating increased chondrocyte ECM deposition when the expression of *Hadha* and *Hadhb* is suppressed (Figs. 3b–d and S5b–d). To confirm the findings from the genetic strategy, we also treated chondrocytes with trimetazidine (TMZ), a TFP complex inhibitor

blocking FAO, and observed consistent results in immunoblot detection and micromass/3D-agarose culture models (Figs. 3e and S5e, f). Moreover, we generated *HADHA*- and *HADHB*-knockout (KO) C28/I2 cells (Fig, S5g) and revealed that FAO blockage by HADHA or HADHB depletion increased COL2A1 and SOX9 expression and decreased MMP13 expression, which was reversed by HADHA or HADHB reintroduction (Fig. 3f and S5h). Furthermore, we reconstructed the

**Fig. 3 | Enhanced FAO exacerbates OA phenotype in vivo and in vitro.**
**a** Immunoblot detection of ACAN, COL2A1, SOX9, MMP3, MMP13, and HADHA in mouse chondrocytes transfected with *Hadha* siRNAs. All cells were treated with 200 μM FFA. **b** mRNA expression levels of *Hadha* in mouse chondrocytes infected with a lentivirus encoding *Hadha* short hairpin RNA #2 (*n* = 6). **c**, **d** Micromass culture (**c**) or 3D-agarose culture (**d**) and alcian blue staining of murine chondrocytes infected with a lentivirus encoding *Hadha* short hairpin RNA #2 (*n* = 6). All cells were treated with 200 μM FFA. **e** 3D-agarose culture and alcian blue staining of murine chondrocytes treated with FFA or FFA + TMZ (*n* = 6). **f** Immunoblot detection of COL2A1, SOX9, MMP13, and HADHA in *HADHA*-KO C28/I2 cells overexpressing HADHA or harboring the empty vector. All cells were treated with 200 μM FFA. **g** Immunoblot detection of ACAN, COL2A1, SOX9, MMP13, and HADHA in *HADHA*-KO C28/I2 cells overexpressing wild-type HADHA, HADHA K728R mutant, or harboring the empty vector. All cells were treated with 200 μM

FFA. **h** Safranin O/Fast green staining and IHC staining for HADHA in knee joints of *Hadha*^fl/fl and *Hadha*^fl/fl; *Acan*-creER^T2 mice that underwent DMM or sham surgery. Samples were collected 12 weeks after surgery (*n* = 8). **i** Safranin O/Fast green staining; IHC staining for HADHA and HADHA K728ac; and IF staining for COL2A1 and MMP13 in knee joints from *Hadha* K728R^CKI/CKI and *Hadha* K728R^CKI/CKI; *Acan*-creER^T2 mice that underwent DMM surgery and fed HFD or ND. Samples were collected 8 weeks after surgery (*n* = 8). **b**, **h**, and **i** Data are mean ± SD, as determined via unpaired two-tailed *t*-test, or nonparametric Mann–Whitney test. **c**–**e** Data are minimum to maximum: each point represents an individual biological replicate, box from 25th to 75th percentiles; center line at median; tukey whiskers, as determined via unpaired two-tailed *t*-test (**c**, **d**) or one-way ANOVA followed by Tukey's multiple-comparisons test (**e**). The immunoblotting data are representative of three independent experiments. Source data are provided as a Source Data file.

HADHA-K728R mutant in *HADHA*-KO C28/I2 cells. Compared with the wild-type control, FAO-quiescent chondrocytes demonstrated higher ECM anabolic signaling and lower ECM catabolic signaling (Fig. 3g).

To extend these in vitro insights, we generated *Hadha*^fl/fl and *Hadhb*^fl/fl mice and crossed them with Acan-CreER^T2 mice or Col2-CreER^T2 mice (Fig. S5i), considering the potential lethality associated with global TFP knockout[39]. Specifically deleting *Hadha* or *Hadhb* in the cartilage had no observable impact on the synovium or meniscus structure (Fig. S5j, k), but substantially inhibited cartilage destruction compared to that in wild-type mice 12 weeks after DMM surgery (Figs. 3h and S6a–c). This protective effect was more pronounced when mice were fed HFD. Cartilage-specific deletion of *Hadha* not only alleviated OA progression in the HFD-DMM group 8 weeks after DMM surgery, but also significantly prevented OA development in the HFD-sham group (Fig. S6d). To elucidate the function of FAO activity in OA and its interplay with HFD, we generated an HADHA K728R conditional knock-in (CKI) mouse model (*Hadha* K728R^CKI/CKI; Acan-CreER^T2) (Fig. S6e). The siblings were randomly assigned to either an ND or HFD group and underwent DMM surgery at 12 weeks of age. Compared with mice fed with ND, control mice fed with an HFD exhibited more severe OA manifestations 8 weeks after surgery, and these manifestations were inhibited in Hadha K728R CKI mice, which have even milder cartilage damage than control mice when fed ND (Fig. 3i). This protective effect was confirmed via IF staining, highlighting COL2A1 downregulation and MMP13 upregulation in response to HFD in control mice and counteracting effects in *Hadha* K728R CKI mice. These collective findings delineate a compelling role for chondrocyte FAO in exacerbating OA progression.

## FAO inhibition amplifies AMPK activity and stabilizes SOX9

To identify downstream mediators of FAO in OA pathophysiology, we discovered an intriguing discrepancy between SOX9 mRNA- and protein-expression levels in primary mouse chondrocytes upon increased FFA stimulation. The decline in SOX9 protein levels was more robust than the reduction in its mRNA levels (Fig. 4a). Moreover, knockdown of *Hadha* and *Hadhb* significantly increased SOX9 protein levels in chondrocytes but showed minimal effect on *Sox9* mRNA expression (Fig. S7a). Immunoblot detection revealed a significantly shorter SOX9 protein half-life in FFA-exposed murine chondrocytes (Fig. 4b), which was reversed by TMZ treatment (Fig. 4c). The proteasome inhibitor MG132 effectively halted protein degradation, whereas the lysosome inhibitor chloroquine had no effect, suggesting that the degradation of SOX9 is stimulated by FAO via the proteasome pathway. Our data suggested that ATP production increased in chondrocytes with FAO hyperactivity (Fig. S4g), which was validated by cellular-ATP assays after FFA stimulation (Fig. 4d). Recognizing the potential for inhibiting AMP-activated protein kinase (AMPK) activity via excessive ATP production through the canonical pathway[40], we employed an AMPK inhibitor and activator (dorsomorphin and A-769662), which led to decreased and increased phosphorylation,

respectively, at S79 of acetyl-CoA carboxylase (p-ACC), a canonical substrate site of AMPK[40]. We further investigated whether cellular AMPK activity is involved in SOX9 protein degradation. We found that dorsomorphin dramatically diminished SOX9 protein levels, whereas A-769662 induced it, and both exerted a minimal influence on its mRNA expression (Fig. S7b). Moreover, we observed decreased AMPK activity and SOX9 levels in mouse chondrocytes upon FFA exposure (Fig. S7c) but increased levels after TMZ treatment (Fig. S7d), which was consistently observed in HFD mouse models and human OA samples from individuals with obesity (Fig. S7e, f). Further validation involving *Hadha* or *Hadhb* knockdown, as well as *Hadha* deletion in mouse chondrocytes, demonstrated the upregulation of AMPK activity and SOX9 levels (Figs. 4e and S7g), consistent with our in vivo results of knocking out *Hadha* or *Hadhb* in the DMM mouse model (Fig. S7h–k). These data suggested that excessive FAO promotes SOX9 protein degradation, which is possibly mediated by AMPK.

Further analysis revealed increased and decreased SOX9 ubiquitination after dorsomorphin and A-769662 treatments, respectively, in C28/I2 cells (Fig. S8a). FFA treatment increased SOX9 ubiquitination, which was effectively reversed by A-769662 or TMZ (Fig. 4f). Furthermore, *Hadha* and *Hadhb* knockdown in mouse chondrocytes, and TMZ treatment effectively restored SOX9 protein expression under FFA pretreatment (Fig. S8b). AMPK is a broad-spectrum serine/threonine kinase that regulates the stability of various substrate proteins via phosphorylation–ubiquitination crosstalk[41]. UPLC-MS/MS analysis revealed phosphorylation of SOX9 at 6 serine/threonine sites (T25, T87, S98, S136, T138, and T193) in C28/I2 cells treated with A-769662 rather than with FFA (Fig. 4g). These sites were conserved across different species (Fig. S8c). MS analysis also revealed that FFA treatment significantly increased the number of ubiquitination sites (K62, K100, K166, K180, K183, K242, and K249) on SOX9 compared to that on the control and A-769662-treated groups (Fig. S8d). We mutated SOX9 by replacing these serine/threonine residues individually with alanine and found that these single mutations minimally impacted SOX9 stability. However, mutation of all 6 serine/threonine residues (ST-A, which mimics dephosphorylation) dramatically decreased SOX9 stability compared to any single mutant (Fig. S8e). When coupled with an anti-phospho-AMPK substrate motif antibody (p-AMPK motif), co-immunoprecipitation confirmed SOX9 as a potential substrate of AMPK, and the phosphorylation of the AMPK consensus motif in SOX9 significantly decreased after ST-A mutation (Fig. 4h). Ubiquitination-immunoprecipitation analysis revealed that ubiquitination was higher for SOX9 ST-A and lower for SOX9 ST-D (all the 6 serine/threonine residues were replaced by aspartate, which mimics phosphorylation) than for wild-type SOX9 (Fig. 4i). MS also showed that the E3 ubiquitin ligases, ZNF598, TRIM9, and RLIM, potentially bound to SOX9 in FFA-treated cells (Fig. S8f). Co-immunoprecipitation confirmed that FFA treatment specifically enhanced SOX9 interactions with TRIM9 (Figs. 4j and S8g). Moreover, this interaction intensified with the SOX9 ST-A mutant and diminished with the SOX9 ST-D

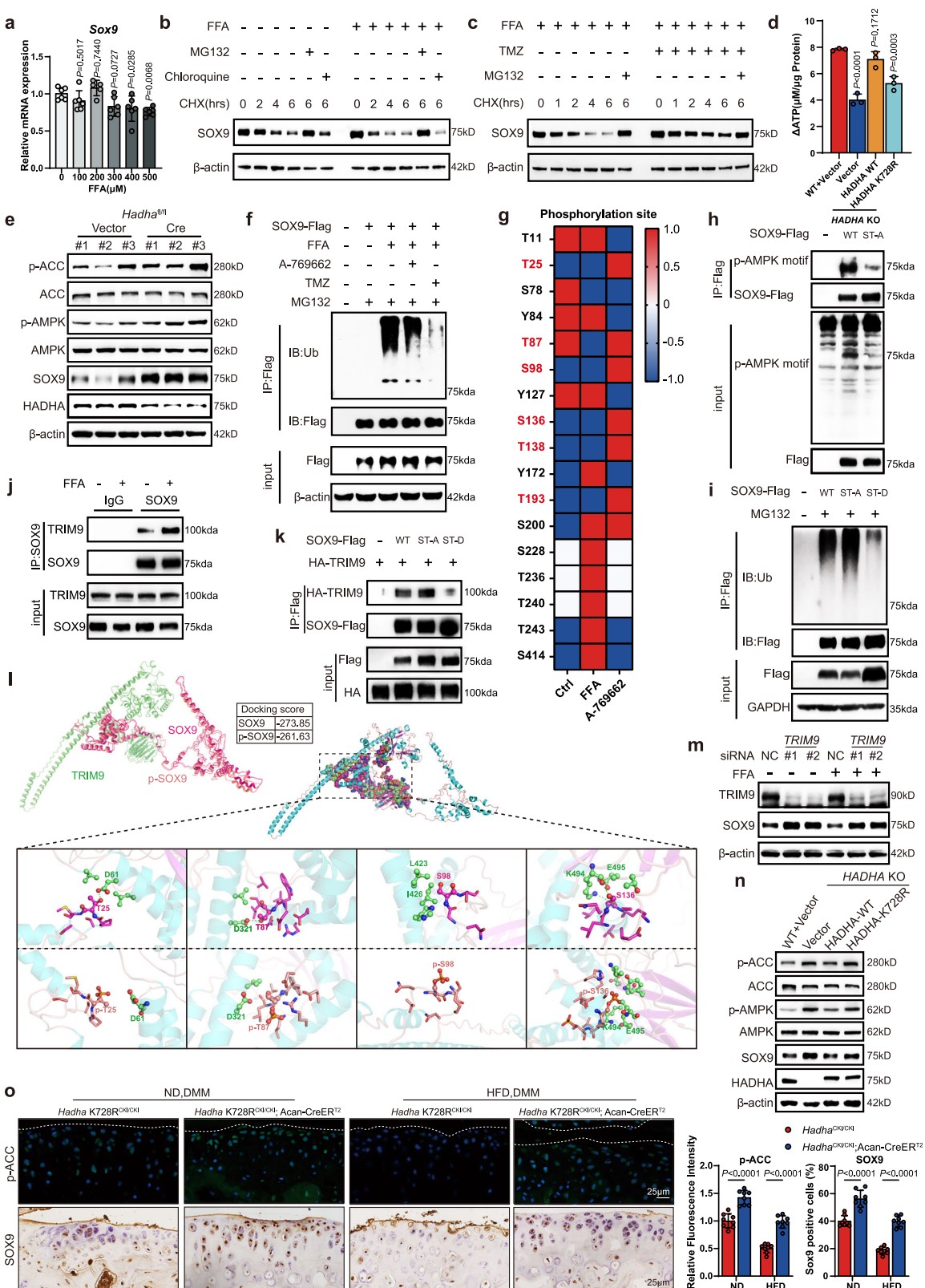

mutant (Fig. 4k). To elucidate its structural implications, molecular docking was performed (Fig. 4l). SOX9 and TRIM9 form 24 hydrogen bonds, fostering a robust interaction. Phosphorylation at these 6 serine/threonine sites on SOX9 decreased the number of hydrogen bonds with TRIM9 to 20, primarily due to phosphorylation at T25, T87, and S136. Meanwhile, a strong hydrophobic interaction existed between TRIM9 and S98 of SOX9, which was sterically hindered by S98

phosphorylation. To further investigate TRIM9's impact on SOX9 abundance, we utilized specific siRNAs to silence *TRIM9* and observed that *TRIM9* knockdown not only elevated basal SOX9 levels but also effectively reversed FFA-induced SOX9 degradation (Fig. 4m). These results indicate that AMPK-induced phosphorylation of SOX9 prevented its interaction with TRIM9, thus mitigating its ubiquitination-mediated degradation.

**Fig. 4 | FAO inhibition amplifies AMPK activity and stabilizes SOX9. a** *Sox9* mRNA levels in mouse chondrocytes treated with FFA ($n = 6$). **b**, **c** Immunoblot analysis of SOX9 in primary murine chondrocytes treated as indicated. **d** ATP production after FFA exposure in *HADHA*-KO C28/I2 cells overexpressing wild-type HADHA or HADHA K728R or harboring the empty vector ($n = 3$). **e** Immunoblot of p-ACC/ACC, p-AMPK/AMPK, SOX9, and HADHA in *Hadha*^fl/fl murine chondrocytes infected with Cre or adenovirus vector 72 h before harvesting. **f** Immunoblot detection of ubiquitinated Flag-tagged SOX9 overexpressed in C28/I2 cells via Flag immunoprecipitation. **g** Heatmap showing the results of UPLC-MS/MS analysis for identifying phosphorylation sites on the SOX9 protein purified from C28/I2 cells. The presence, absence, or lack of detection of modifications at the specified sites in the SOX9 protein are denoted by '1', '−1', and '0' respectively. **h** Immunoblot detection of AMPK phosphorylates consensus motif (p-AMPK motif) in Flag-tagged wild-type, and SOX9 ST-A mutant overexpressed in C28/I2 cells using an anti-p-AMPK-motif antibody via Flag immunoprecipitation. **i** Immunoblot detection of ubiquitinated, Flag-tagged wild-type, ST-A, and ST-D SOX9 overexpressed in C28/I2 cells via Flag immunoprecipitation. ST-A and ST-D denote SOX9 variants with six serine/threonine residues mutated to alanine and aspartate residues, respectively. **j** Co-immunoprecipitation of SOX9 with TRIM9 in HEK293T cells treated with FFA or BSA. **k** Co-immunoprecipitation of Flag-tagged wild-type SOX9 or SOX9 mutants with HA-TRIM9 in HEK293T cells. **l** Molecular docking results demonstrating the interaction between wild-type or ST-phosphorylated SOX9 and TRIM9. **m** Immunoblot detection of TRIM9 and SOX9 in HEK293T cells transfected with *TRIM9* siRNAs. **n** Immunoblot detection of p-ACC/ACC, p-AMPK/AMPK, SOX9, and HADHA in *HADHA*-KO C28/I2 cells overexpressing wild-type HADHA, HADHA K728R mutant, or harboring the empty vector. **o** IF staining for p-ACC and IHC staining for SOX9 in mouse-knee cartilage tissues indicated in Fig. 3i ($n = 8$). Data are mean ± SD, as determined by one-way ANOVA followed by Dunnett's multiple-comparisons test (**a**, **d**), or unpaired two-tailed *t*-test (**o**). The immunoblotting data are representative of three independent experiments. Source data are provided as a Source Data file.

To examine the impact of HADHA K728ac on AMPK activity and SOX9 ubiquitination in chondrocytes, we performed ubiquitination-immunoprecipitation analysis of *HADHA*-KO C28/I2 cells reconstructed with wild-type HADHA or HADHA K728R mutant, and found lower SOX9 ubiquitination in K728R mutant cells (Fig. S8h). *HADHA*-KO cells showed increased p-ACC and p-AMPK levels, along with increased SOX9 expression, which were reversed by wild-type HADHA reconstruction but not by HADHA K728R (Fig. 4n). These in vitro findings were further validated in mouse models (Fig. 4o). Collectively, our data suggest that heightened FAO induced by elevated HADHA K728 acetylation suppressed AMPK activity, subsequently triggering SOX9 degradation via phosphorylation–ubiquitination crosstalk, and that TMZ disrupted these processes by impeding FAO, thereby stabilizing SOX9.

### FAO-driven epigenetic reprogramming controls OA pathogenicity

Besides mitochondrial protein-acetylation, histone acetylation is highly sensitive to the availability of acetyl-CoA in multiple cell types[42,43]. Our acetylome data uncovered a significant 4.025-fold upregulation of histone H3.3 lysine 28 acetylation (H3K27ac) in HFD-DMM mouse cartilage tissues. Such acetylation is intricately associated with chromatin opening and transcriptional activation. Therefore, we examined the global dynamics of H3K27ac in chondrocytes stimulated with FFA. Both immunoblot detection and IF staining confirmed that FFA treatment upregulated H3K27ac level in mouse chondrocytes, which was reversed by TMZ treatment (Figs. 5a, b and S9a). Next, we conducted a comprehensive genome-wide analysis of H3K27ac via CUT&Tag sequencing in mouse chondrocytes treated with FFA. We observed distinct global H3-acetylation patterns encompassing approximately 4115 differentially acetylated regions (Fig. 5c). GO enrichment analysis revealed that genes marked by or near H3K27ac peaks were implicated in ECM structural constituents, cartilage development, and collagen fibril organization (Fig. S9B). Besides, Kyoto Encyclopedia of Genes and Genomes enrichment analysis indicated the involvement of these genes in critical signaling pathways contributing to OA pathophysiology, including ECM-receptor interactions, proteoglycan regulation, cellular senescence, and the Hif-1 signaling pathway (Fig. 5d).

Further inspection of the CUT&Tag data revealed that H3K27ac around cartilage ECM anabolic genes, such as *Acan*, *Col2a1*, *Sox5*, and *Sox6*, clearly decreased in FFA-treated chondrocytes. Conversely, acetylation around the cartilage catabolic ECM genes, *Mmp13* and *Adamts7*, clearly increased (Fig. 5e). Acetylation around the *Pdk4* gene, a metabolic regulator that decreases glucose utilization, increases fat metabolism, and induces aerobic glycolysis[44,45], was significantly elevated, consistent with our proteomic data. Concurrently, reduced acetylation around the *Hdac3* gene, known to deacetylate HADHA[46], was observed, suggesting a potential mechanism favoring HADHA acetylation in coordination with a high acetyl-CoA abundance. The alteration of H3K27ac around these genes was confirmed by CUT&Tag-qPCR (Fig. S9c), and the effect of the changes in H3K27ac on their mRNA levels were measured in FFA-treated mouse chondrocytes (Fig. S9d), where the changes in cartilage anabolic and catabolic factors were reversed by TMZ treatment (Fig. 5f). These in vivo effects were also examined in mouse HFD-DMM models (Fig. 5g) and human OA samples from individuals with obesity (Fig. S9e), providing compelling evidence that H3K27ac-dependent epigenetic reprogramming underlies FAO-induced OA progression. In addition, CUT&Tag data indicated reduced H3K27ac around genes related to cholesterol influx (*Apob*), cholesterol synthesis (*Srebf1*, *Srebf2*), and oxysterol receptors (*Esr1*) following FFA stimulation, which was further validated at the mRNA and protein levels (Figs. 5h and S9f), suggesting that FAO regulates cartilage ECM homeostasis apart from the cholesterol pathway.

### Targeted delivery of TMZ to cartilage tissues with red blood cell exosomes (RBC-Exos) alleviated OA

TMZ, a specific inhibitor of the TFP complex, has been widely used for treating ischemic cardiac disease, but also shows extracardiac benefits[47]. However, the roles of FAO vary across diverse cell types and disease backgrounds[46,48]. Indeed, FAO has recently been reported to alleviate synovitis in knee osteoarthritis by targeting fibroblast-like synoviocytes[49]. To minimize the potential side effects of TMZ on other articular tissues, we implemented a well-established chondrocyte-affinity peptide (CAP)-Exo system to enhance cartilage-specific delivery of TMZ (Figs. 6a and S10a)[50,51]. RBC-Exos were generated from volunteers who provided blood samples (Fig. S10b), and hybrids with liposomes containing a CAP (Fig. S10c) were prepared with or without TMZ loading. Hybridization minimally compromised the morphology and size distribution of RBC-Exos (Fig. S10d, e). Subsequent investigation of the impact of CAP-RBC-Exo/TMZ on the HFD-DMM-induced OA model revealed effective alleviation of cartilage destruction, not only in the HFD-DMM group but also in the ND-DMM group (Fig. 6b, c). Moreover, CAP-RBC-Exo/TMZ-treated mice exhibited notably elevated COL2A1 and reduced MMP13 expression in knee-joint cartilage (Fig. 6d). We detected increased p-ACC levels and subsequent SOX9 elevation in mouse knee cartilage tissues treated with CAP-RBC-Exo/TMZ, indicating involvement of the FAO-AMPK-SOX9 pathway. Moreover, synovitis was not exacerbated after CAP-RBC-Exo/TMZ injection (Fig. S10f), suggesting that it was an efficient and well-tolerated therapeutic strategy for ameliorating OA.

## Discussion

Various studies have suggested a correlation between fatty acids and OA[20,22,52,53]. However, the role of fatty acid metabolism in OA pathogenesis remains elusive. A recent study suggested that fatty acid synthesis in chondrocytes influences OA pathogenesis[30], indicating

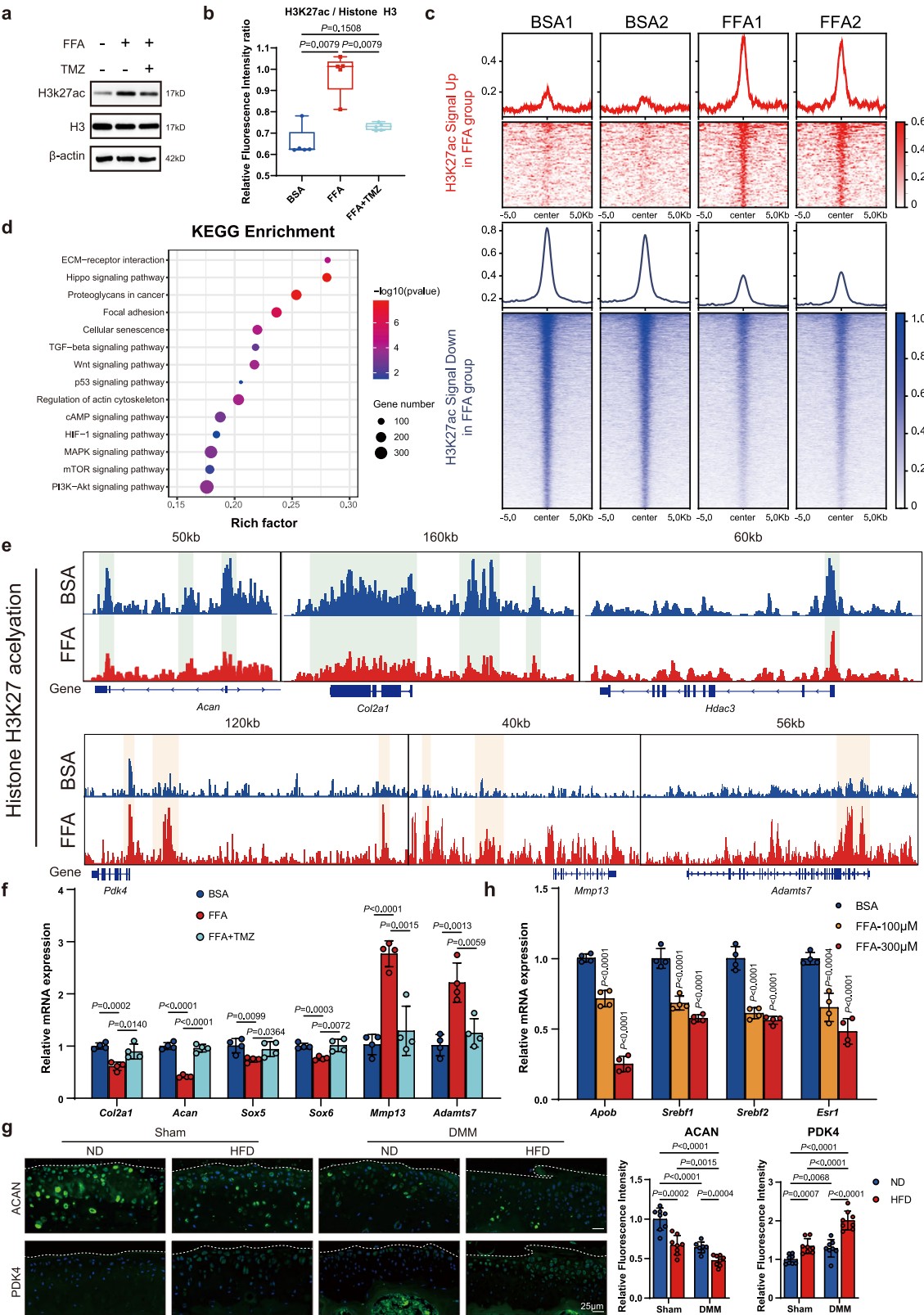

that intracellular fatty acids are a crucial regulatory factor. Here, we describe the expected correlation between cartilage degradation and fatty acid accumulation in the osteoarthritic cartilage, particularly in the context of obesity. Furthermore, we identified fatty acid uptake from the synovial fluid as a potential source of intracellular lipids, synergistically enhanced by lipid stress and inflammatory-mechanical factors, underscoring an important interplay between OA-associated

inflammatory-mechanical factors and the metabolic aspect of obesity during OA progression. Besides stimulating chondrocyte fatty acid uptake, lipid overload results in excessive FFAs in the local environment, while osteoarthritic factors disrupt the dense network established by collagen and proteoglycans in the cartilage matrix, which normally restricts the transport of large lipid molecules[23]. These effects may have more profound implications in vivo.

**Fig. 5 | FAO-driven epigenetic reprogramming controls OA pathogenicity.**
**a** Immunoblot detection of H3K27ac in murine chondrocytes treated with BSA, FFA (200 μM), or FFA (200 μM) + TMZ (300 μM) for 48 h. **b** Relative fluorescence-intensity ratio of H3K27ac to histone H3 in mouse chondrocytes treated with BSA, FFA, or FFA + TMZ. See Fig. S9a for additional details. **c** Heatmaps displaying a spike in normalized coverages of global H3K27 acetylation in mouse chondrocytes treated with FFA or a BSA control. **d** KEGG enrichment analysis of genes around significantly altered H3K27ac peaks ($\log_2$ FC > 1, $P$ < 0.01), as determined by a two-tailed Fisher's exact test followed by Benjamini–Hochberg (BH) correction. **e** Analysis of H3K27ac at gene clusters in BSA- or FFA-treated mouse chondrocytes, including *Acan*, *Col2a1*, *Hdac3*, *Pdk4*, *Mmp13*, and *Adamts7*. **f** mRNA expression levels of *Col2a1*, *Acan*, *Sox5*, *Sox6*, *Mmp13*, and *Adamts7* in mouse chondrocytes

treated with BSA, FFA, or FFA + TMZ, as assessed via RT-qPCR ($n$ = 4). **g** IF staining for ACAN and PDK4 in mouse-knee cartilage tissues represented in Fig. 1f ($n$ = 8). **h** mRNA expression levels of *Apob*, *Srebf1*, *Srebf2*, and *Esr1* in mouse chondrocytes treated with the indicated concentrations of FFA, as assessed via RT-qPCR ($n$ = 4). **b** Data are the minimum to maximum: each point represents an individual biological replicate, box from 25th to 75th percentiles; center line at median; tukey whiskers, as determined via nonparametric two-tailed Mann–Whitney test. **f–h** Data are mean ± SD, as determined by two-tailed $t$-test (**f–g**) or one-way ANOVA followed by Dunnett's multiple-comparisons test (**h**). The immunoblotting data are representative of three independent experiments. Source data are provided as a Source Data file.

However, in contrast to the concept that lipotoxicity is associated with lipid overload and accumulation, the utilization of fatty acids (particularly FAO) is crucial for understanding the overall pathophysiological contribution of lipid metabolism to OA. Skeletal progenitor cells suppress FAO by inducing SOX9 mRNA expression, which favors adaptation to an avascular environment and maintains the chondrogenic fate; otherwise, they differentiate into osteogenic cells[54]. This recognized adaptability in chondrocyte metabolism during development hints at a more nuanced perspective of lipid metabolism in mature chondrocytes in terms of OA pathophysiology. Moreover, SOX9 has been extensively studied in OA as a central cartilage ECM regulator. Its newly identified function as a crucial mediator of lipid metabolism has established a novel connection between lipid metabolism and chondrocyte ECM regulation. Here, we provide evidence that articular chondrocytes adapt to an elevated FAO state in response to lipid stress, which subsequently destabilizes SOX9 protein, consequently disrupting ECM homeostasis and accelerating OA progression. Furthermore, the recognition of SOX9 as an FAO inhibitor during chondrogenesis implies that decreased SOX9 levels may reciprocally enhance FAO in mature chondrocytes when exposed to excess lipid stress.

AMPK is a crucial cellular energy sensor. Excess energy leads to dephosphorylation of AMPK at Thr-172 on its α subunit, resulting in inhibition of the AMPK complex activity[55,56]. Normal articular chondrocytes exhibit AMPK expression and potent AMPKα phosphorylation, whereas decreased AMPK expression and Thr-172 phosphorylation were observed in the osteoarthritic cartilage[57]. Mechanistically, AMPK inhibited chondrocyte catabolic responses to inflammatory stress via various signaling pathways[58,59]. However, the underlying mechanism of the reduced activation of AMPK in OA chondrocytes, as well as its role in cartilage ECM anabolic processes, has not been clearly determined. Our findings indicate that FFAs accumulating in chondrocytes inhibited AMPK activity due to energy overproduction. This inhibition leads to decreased SOX9 phosphorylation, promoting its degradation, and a compromised cartilage ECM anabolism. Importantly, our data firstly identify SOX9 as a potential substrate protein for AMPK. The AMPK-induced phosphorylation of SOX9 impedes its interaction with TRIM9, which may extend to other E3 ubiquitin ligases, thereby playing a crucial role in regulating the stability of the SOX9 protein.

In addition to fueling OXPHOS and thereby generating cellular energy, FAO generates a substantial amount of the metabolic end product, acetyl-CoA[25]. Acetyl-CoA serves as a central metabolic intermediate connecting multiple metabolic reactions and controls key cellular processes by influencing the acetylation profiles of various proteins[60]. The acetylation of substrate protein can be influenced by the concentrations of metabolic fuels and the abundance of intracellular acetyl-CoA, especially in alkaline environments, such as the mitochondrial matrix[24,28]. Our findings provide evidence that excess fatty acids lead to acetyl-CoA accumulation in chondrocytes and rewire mitochondrial lipid metabolism by acetylating the key FAO enzyme HADHA at lysine 728, thereby enhancing its catalytic activity.

Increased FAO subsequently generates more acetyl-CoA and establishes a positive-feedback loop. These findings provide novel insights into how chondrocytes sense and respond to excess fatty acids in the context of obesity.

Besides mitochondrial protein-acetylation, global alterations in intracellular-acetyl-CoA levels can influence histone acetylation. The covalent addition of acetyl groups disrupts ionic interactions between DNA and histones, thus promoting euchromatin formation and gene expression. Metabolites derived from cellular metabolic pathways may act as sensors of nutrient states and mediate the corresponding epigenetic changes[61,62], highlighting the connections between metabolic disruption and epigenetic regulation underlying various diseases. An important study found that glutamine-derived acetyl-CoA epigenetically regulated chondrogenic gene expression, particularly *Acan* and *Col2*, through histone acetylation during bone development, thereby controlling chondrocyte identity and function[63]. Here, our data suggest that FAO-derived acetyl-CoA controls the complex metabolic-transcriptional network in mature chondrocytes, governed by epigenetic remodeling. In addition to downregulating histone acetylation levels around cartilage anabolic ECM genes, excessive FFA induces histone acetylation around matrix-degrading enzyme genes, including MMP13 and ADAMTS7. This epigenetic induction disrupted ECM metabolism and eventually led to cartilage destruction. Additionally, the FAO-driven epigenetic reprogramming enabled chondrocyte adaptation to lipid stress. Elevated histone acetylation around the *Pdk4* gene induced Pdk4 at both the mRNA and protein levels in chondrocytes, consequently introducing a metabolic shift from glucose oxidation to FAO[64] and tuning the balance between lipid synthesis and lipid utilization[65], as well as contributing to the development of the senescence-associated secretory phenotype[66]. These findings further emphasize that FAO-driven epigenetic regulation governed the overall OA progression by affecting both ECM homeostasis and chondrocytes senescent.

There are a number of recognized limitations in our study. Despite revealing the important role of fatty acids and FAO in OA, the impact of other obesity-related metabolic factors on OA requires further exploration. A previous study highlighted the potential role of cholesterol in OA[19]. However, in our research, despite the comparable severity of OA induced by HFD, cholesterol levels remained relatively stable. Our existing data suggest that under excessive fatty acid accumulation, chondrocytes may favor increased cholesterol efflux while suppressing cholesterol influx and synthesis. This metabolic shift could partially explain why cholesterol levels in mouse cartilage remained stable. More importantly, this disparity may arise from the differences in experimental models, given that obesity itself is a highly complex condition. Future research efforts should focus on elucidating how chondrocytes modulate the balance between fatty acid metabolism and cholesterol metabolism and whether each pathway predominates in specific subtypes of obesity-related OA. It is crucial to clearly distinguish these subtypes during clinical practice to implement more precise therapeutic strategies. In addition, our in vivo data demonstrate that HFD alone induces mild fatty acid accumulation

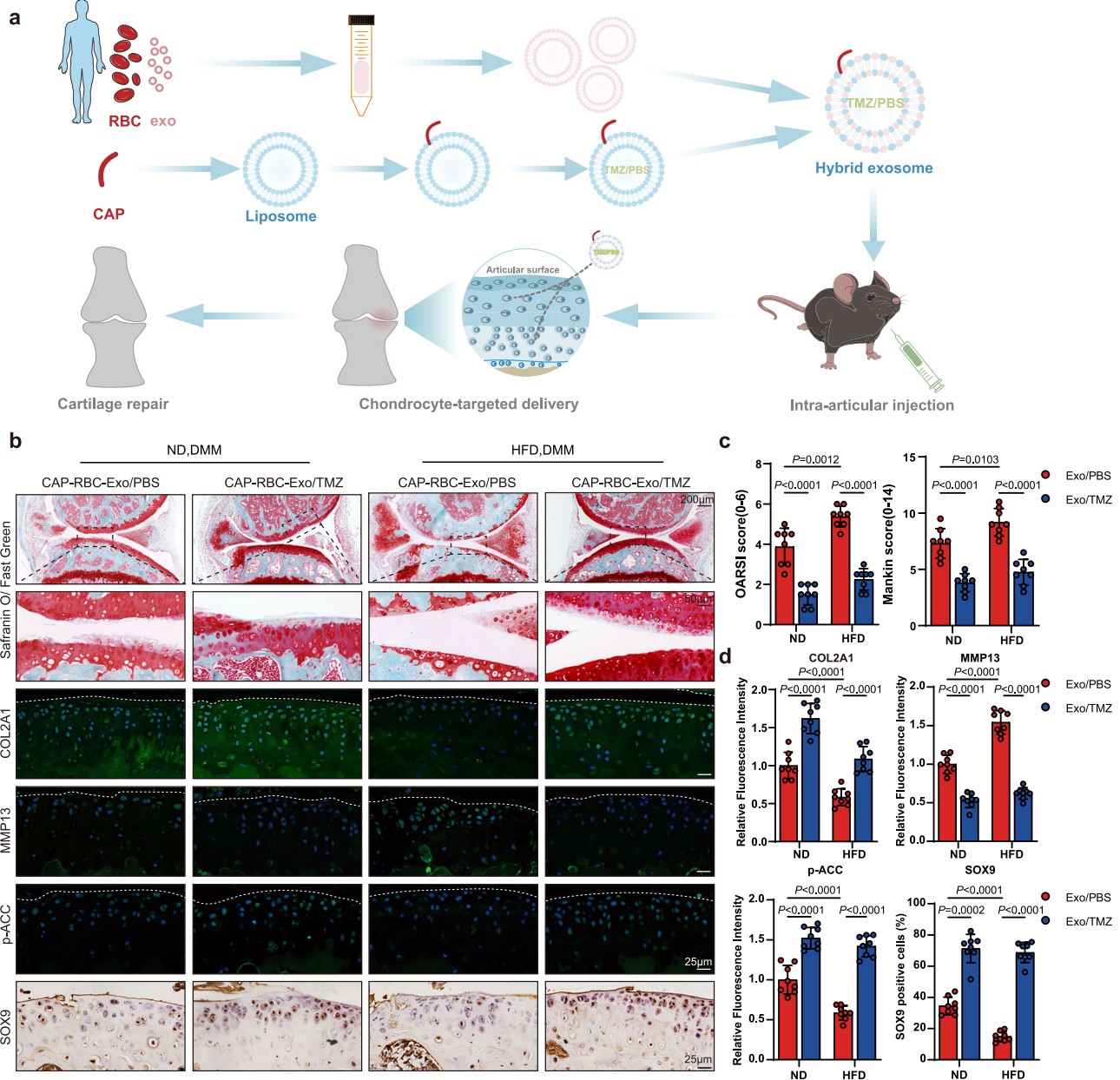

**Fig. 6 | Targeted delivery of TMZ to cartilage tissues with red blood cell exosomes (RBC-Exos) alleviates OA. a** Schematic illustration of CAP-RBC-EXOs used for the targeted delivery of TMZ to chondrocytes for OA treatment, created by figdraw.com. **b–d** Safranin O/Fast green staining; IF staining for COL2A1, MMP13, and p-ACC; and IHC staining for SOX9 in knee joints from mice fed an HFD or ND that underwent DMM surgery, followed by CAP-RBC-Exo/TMZ or CAP-RBC-Exo/PBS arthrocentesis ($n = 8$). The data represent the mean ± SD, as determined by two-tailed $t$-test or nonparametric two-tailed Mann–Whitney test. Source data are provided as a Source Data file.

within the cartilage and a moderate OA phenotype. This could be attributed to the relatively short duration (16 weeks) of HFD feeding. With prolonged HFD feeding, these finding could be more pronounced in the HFD-Sham group. A previous study reported that 30-weeks HFD feeding is sufficient to promote intracellular lipid deposition and cartilage degeneration in mice[23]. Due to timeline considerations and to more clearly detect the impact of HFD on OA, we incorporated the DMM model. Through stringent HFD-DMM vs. ND-DMM comparison, we discovered that fatty acid accumulation accelerates OA progression by enhancing FAO, which was further validated in vitro using single treatment with fatty acids. We believe that this design does not affect the main conclusions, because the HFD-DMM model better mimics the conditions of patients with obesity-related OA in the real world, who

not only experience lipid stress, but also increased joint mechanical stress due to weight gain.

Collectively, our data systematically elucidate the underlying mechanisms linking fatty acid metabolism to OA, relying on the fatty acid-FAO-epigenetic regulatory network (Fig. 7). FAO-induced SOX9 degradation coupled with H3K27ac exacerbated OA, and acetyl-CoA played a pivotal role in mediating these processes. This multifaceted understanding opens new avenues for targeted therapeutic interventions against OA associated with metabolic disturbances. The cartilage-targeted delivery of TMZ, acting as both an AMPK agonist and FAO antagonist, demonstrates efficacy not only in obesity-related OA but also in DMM-induced OA, providing important preliminary evidence for translating our findings into clinical applications.

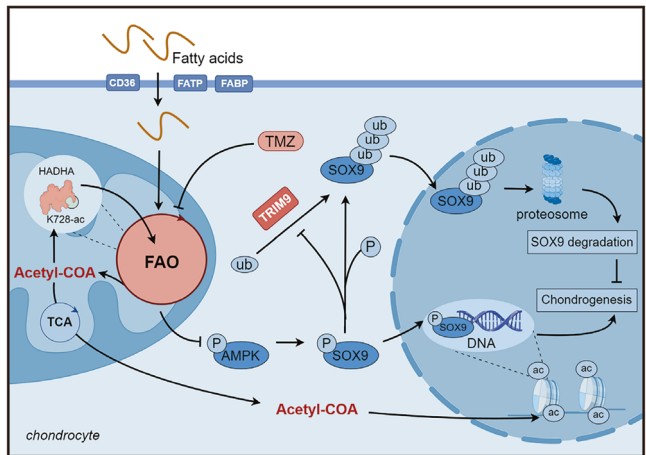

**Fig. 7 | Schematic representation of the fatty acid-FAO-epigenetic regulatory network in the context of ObOA.** Fatty acids that accumulate in chondrocytes enter the FAO pathway and lead to acetyl-CoA accumulation, which reshapes the chondrocyte protein acetylation profile. Mitochondrial acetyl-CoA facilitates the acetylation of HADHA, enhancing its enzymatic activity and creating a positive feedback loop that further amplifies FAO. Concurrently, excessive FAO suppresses AMPK activity, resulting in reduced phosphorylation and increased ubiquitination-mediated degradation of SOX9. In parallel, nuclear acetyl-CoA alters histone acetylation patterns, promoting transcriptional activation of ECM catabolic genes such as MMP13 and ADAMTS7 while suppressing ECM anabolic genes including ACAN and COL2A1. These combined effects disrupt ECM turnover and accelerate cartilage degradation. Pharmacological inhibition of FAO using TMZ effectively interrupts this pathological cascade and mitigates OA progression (created by figdraw.com).

## Methods

### Cells

Primary murine articular chondrocytes were extracted from the tibial plateaus of 7-day-old mice (wild-type or $Hadha^{fl/fl}$) through digestion with 0.5% collagenase II[67]. Cells were cultured in Dulbecco's modified Eagle's medium (DMEM) supplemented with 10% fetal bovine serum (FBS) and 1% penicillin-streptomycin. The C28/I2 chondrocytes were sourced from Sigma-Aldrich, while HEK293T cells and 3T3-L1 pre-adipocytes were obtained from the American Type Culture Collection. C28/I2 and HEK293T were cultured in DMEM, containing 10% FBS and 1% penicillin-streptomycin. 3T3-L1 were cultured in DMEM with 10% Newborn Calf Serum (NBCS). All the cells were cultured in 5% $CO_2$ and 2.5% $O_2$ at 37 °C. The collection and utilization of primary mouse chondrocytes were ethically approved by the Institutional Animal Care and Use Committee of Zhejiang University (grant number ZJU20230529) and adhered to institutional guidelines.

### Animal model

C57BL/6 mice were acquired from Hangzhou QiZhen Laboratory Animal Technology Center. ACAN-CreER$^{T2}$ mice were kindly gifted by Prof. Ximei Wu (Zhejiang University School of Medicine). Col2-CreER$^{T2}$ mice were obtained from the Jackson Laboratory. $Hadha^{fl/fl}$, $Hadhb^{fl/fl}$, and $Hadha^{CKI/CKI}$ (p. K728R Conditional Knock in) mice were purchased from Cyagen Biology Technology. Mice were matched for age and sex, randomly assigned to each experimental group, and housed in pathogen-free barrier facilities. All animal experimental protocols were approved by the Institutional Animal Care and Use Committee of Zhejiang University (grant number ZJU20230529), in accordance with institutional guidelines, and all experiments were conducted in accordance with the ARRIVE guidelines for reporting animal research.

For the establishment of the HFD-induced obesity model, mice were housed in a controlled environment with a temperature maintained at around 21 °C, a 12-h light/dark cycle, and ad libitum access to food and water. Previous studies have indicated that the articular cartilage of mouse knee joints undergoes fundamental development by approximately 3–4 weeks of age, characterized by microscopic differentiation into four zones[68,69]. Thus, 4-week male mice were assigned to either a normal diet (ND) or a high-fat diet (HFD) for a duration of 16 weeks as previously described[46]. The body weight was measured weekly and the Inguinal adipose tissue was collected and weighed post-euthanasia. When indicated, the intraperitoneal glucose tolerance test was conducted 3 days before euthanasia.

For the generation of mice with $Hadha$, $Hadhb$ conditionally knock-out, and $Hadha$ (p. K728R) Conditional knock-in (CKI) in chondrocytes, $Hadha^{fl/fl}$, $Hadhb^{fl/fl}$, and $Hadha^{CKI/CKI}$ mice were bred with Acan-CreER$^{T2}$(Agc$^{Cre/Cre}$) or Col2-CreER$^{T2}$ mice. Where indicated, 8-week-old Acan-CreER$^{T2}$ mice and 4-week-old Col2-CreER$^{T2}$ mice were intraperitoneal injected with Tamoxifen (100 μg/g body weight) for 5 days to induce target gene recombination[70]. Importantly, Agc$^{Cre/Cre}$ mice exhibit a dwarfism phenotype[71]. Therefore, only the Agc$^{+/Cre}$ mice should be used for conditional deletion of a target gene. Moreover, Col2-CreER$^{T2}$ system could achieve targeted gene knockout in cartilage by administering tamoxifen to mice between 2 and 8 weeks of age, with earlier administration better ensuring gene recombination efficiency[72]. Therefore, we chose to inject tamoxifen at 4 weeks of age, a time when the joints are about fully developed, to maximize recombination efficiency without significantly impacting joint development.

The destabilization of the medial meniscus (DMM) model was induced to establish the OA model, according to a previous study[73]. Briefly, mice aged 12 weeks were anesthetized intraperitoneally with pentobarbital sodium salt. A skin incision exposed the patella, which was then laterally dislocated using tweezers. Using microsurgical scissors under a microscope, the medial meniscotibial ligament (MMTL) was carefully transected, consequently releasing the medial meniscus. Post rinsing the joint with sterile saline, sutures were applied. Control mice were sham operated without MMTL transection.

For CAP-RBC-Exo/TMZ treatment, DMM operated mice received arthrocentesis with CAP-RBC-Exo/TMZ, CAP-RBC-Exo/PBS, TMZ, or PBS at a dose of 10 mM TMZ, with a volume of 10 μl per treatment. The administration of this treatment occurred weekly, starting one week after the surgery. The arthrocentesis was performed using a microsyringe equipped with a 32 G needle (Hamilton Company, USA).

The mice were sacrificed 8 or 12 weeks (where indicated) following DMM or sham surgery, and the articular joints were harvested for histological analysis. In instances specified, the cartilage tissues of the tibial plateau were isolated and collected according to a previous study[74]. Briefly, the hindlimbs of the euthanized mice were dislocated and washed with iced-cold PBS. Articular cartilage from the tibia was isolated by trimming around 1 mm of tissue from the tibial bone ends at the knee joint, with careful removal of surrounding connective and bone tissues under a microscope. Each experimental group consisted of mixed cartilage tissue from 3 to 5 mice. The cartilage was initially flash-frozen in liquid nitrogen and then preserved at −80 °C for subsequent analysis.

All animals were monitored daily for signs of pain, distress, or illness by trained staff from the Department of Laboratory Animal Science at Zhejiang University. Animal welfare assessments were conducted in accordance with institutional guidelines. For all surgical procedures and prior to sacrifice, mice were anesthetized via intraperitoneal injection of 1% sodium pentobarbital at a dose of 0.1 ml per 100 g body weight to minimize pain and discomfort. Humane endpoints were established, and animals showing signs of severe distress or meeting euthanasia criteria were promptly and humanely euthanized using approved methods.

### Clinical samples

Human cartilage samples were obtained from 36 patients (female 31 and male 5) who underwent total knee arthroplasty, and categorized

 

into four groups, the Lean Lateral and the Lean Medial were obtained from individuals with lower BMI (< 24, $n = 18$). The Obesity Lateral and the Obesity Media were obtained from individuals with higher BMI (> 28, $n = 22$). The medical records of each patient were carefully reviewed and listed in Supplementary Table 1. The WOMAC[75], Kellgren & Lawrence Grade[76] (upon preoperative imaging), and OARSI grade[77] (after histological staining) were performed for evaluation of OA severity.

Human synovial fluid samples were collected from patients diagnosed with OA and underwent arthrocentesis or arthroplasty, following the clinical criteria outlined by the American College of Rheumatology ($n = 120$, female 86 and male 34)[78]. Each patient's medical records were meticulously reviewed, including age, gender, height, body weight, radiographic findings, and laboratory analyses, with comprehensive details provided in Supplementary Table 2. After centrifugation (-12,000 × $g$, 4 °C, 5 min), the supernatants of each sample were gathered and preserved at −80 °C. Patients underwent classification into three stages guided by the Kellgren & Lawrence Grade[76]. The mild stage involves KL grades 0 and 1, indicating minimal OA or normal conditions ($n = 40$). The middle stage comprises KL grade 2, representing moderate OA ($n = 36$). The severe stage encompasses KL grades 3 and 4, signifying advanced or end-stage OA ($n = 44$). Furthermore, patients were classified based on their BMI levels. The "Lean Group" included individuals with a BMI < 24 ($n = 52$), whereas the "Obese Group" encompassed those with a BMI > 28 ($n = 40$).

Red blood cells (RBCs) were isolated from peripheral blood obtained from two healthy male volunteers aged between 22 and 28 years.

All the human samples including human cartilage, synovial fluid, and peripheral blood, were obtained and utilized strictly adhered to guidelines and approved by the Ethics Committee of Sir Run Run Shaw Hospital (grant number SRRSH20230743). The written informed consent was obtained from all patients and volunteers, including consent to publish information that could potentially identify individuals (e.g., age, sex, or hospital name).

## Histology, immunohistochemistry, and immunofluorescence

The mouse articular joint or human cartilage samples underwent a series of steps: fixation, decalcification, dehydration, paraffin embedding, sectioning at 5 µm, stained with safranin-O and fast green, and scored by two blinded pathologists.

The human cartilage samples were sectioned following optimization of established methods[79]. Briefly, the region of interest (ROI) was defined as the tibial plateau cartilage, excluding the meniscus cover, with dimensions of approximately 1 cm in width and 2 cm in length along the midline of the medial and lateral tibial plateau, where macroscopic osteoarthritis (OA) lesions were most evident. Coronal osteochondral specimens from the ROI were paraffin-embedded and sequentially sectioned from the anterior to the posterior. These sections were then deparaffinized in xylene and rehydrated using graded ethanol. For safranin O/fast green staining, the sections were first stained with 1% fast green (dissolved in 50% alcohol) for 3 min, triple rinsed with 1% acetic acid, and then stained with 1% safranin O (dissolved in 95% alcohol) for 5 min. Mounting with neutral balsam, representative images were obtained using a microscope (Nikon, Japan). The sections that transected the center of the most severe OA lesion in each tibial plateau were scored according to Osteoarthritis Cartilage Histopathology (OACH) assessment system[77,79].

The mouse joint samples were sectioned following optimization based on previous protocols[80]. Briefly, Joints were embedded in paraffin with the medial aspect down. Sagittal sections were started at the medial margin of the joint, and sections were taken through the entire joint at 50-µm intervals, with intervening sections stored for additional stains or immunohistochemistry. Typically, each knee produced

16–22 slides for scoring. These slides were stained with safranin-O and fast green as described above and scored by blinded pathologists according to the OARSI grading system[80], the Mankin histology histopathology grading system (MHHS)[81]. Osteophyte formation was scored as previously described[82]. Where indicated, Krenn's synovitis score[83] and the Meniscus score[84] were utilized to assess histological changes in the synovium and meniscus, respectively. The most severely damaged area, usually on the medial tibial plateau and the corresponding femoral condyle, was used for statistical analysis as the final score of each sample. The femur and tibia were both scored, but only the maximum score was used for statistical analysis. The final scores by each pathologist were averaged for each sample. The OACH, OARSI and MHHS were detailed in Supplementary Tables 4–6.

Immunohistochemistry was conducted using the Super Vision IHC kit. Briefly, the sections near the most damaged region were deparaffinized and hydrated. After antigen retrieval using Tris-EDTA buffer (PH 9.0), the sections were incubated with 3% $H_2O_2$ and 5% BSA at room temperature and with the indicated antibodies at 4 °C overnight. Subsequently, a secondary antibody conjugated with streptavidin-horseradish peroxidase (HRP) was added to the sections, followed by 3,3'-diaminobenzidine (DAB) staining. The sections were finally stained with hematoxylin and mounted. The representative images were obtained using a microscope (Nikon, Japan), and the proportion of positive cells was evaluated using Image J v1.50.

The procedures for immunofluorescence mirror those of immunohistochemistry until incubation with the primary antibody concludes. Subsequently, secondary Alexa Fluor 488 goat anti-rabbit IgG (1:500). Slides were mounted with the antifade mounting medium with DAPI. Representative images were captured using a fluorescence microscope (Olympus, Japan), and the relative fluorescence intensity was quantified by using Image J v1.50. The key steps for this analysis are as follows: First, images were captured under consistent exposure settings to ensure comparability between samples. Next, the images were converted to 8-bit grayscale for the relevant channel, with background subtraction applied to minimize noise. The ROI was then manually outlined for each sample by experienced pathologists, ensuring consistency across comparable samples whenever possible. Subsequently, areas of positive staining within the total ROI were quantified by setting appropriate intensity thresholds to assess the abundance of staining. Finally, mean fluorescence intensity was measured within the selected ROI after establishing the signal thresholds.

## Lipidomic profiling

Cartilage specimens were harvested as described above ($n = 4$ four each group, biological replicates), and homogenized using a mixture of methanol, internal standard, and methyl tertiary butyl ether in a volume of 1 ml. The resulting suspension was resuspended in 0.2 ml pure water and centrifuged (-12,000 × $g$, 4 °C,10 min). The supernatant underwent nitrogen drying, and the resulting powder were subjected to LC-MS/MS analysis. This analysis utilized an LC-ESI-MS/MS system (UPLC, Shim-pack UFLC SHIMADZU CBM A system; MS, SCIEX 6500 + QTRAP® System) conducted by Wuhan Metware Biotechnology Co., Ltd. The analytical conditions were as follows, UPLC: column, Thermo Accucore™ C30 (2.6 µm, 2.1 mm × 100 mm i.d.); solvent system, A: acetonitrile/water (60/40,V/V, 0.1% formic acid, 10 mmol/L ammonium formate), B: acetonitrile/isopropanol (10/90 V/V, 0.1% formic acid, 10 mmol/L ammonium formate); gradient program, A/B (80:20, V/V) at 0 min, 70:30 V/V at 2.0 min, 40:60 V/V at 4 min, 15:85 V/V at 9 min, 10:90 V/V at 14 min, 5:95 V/V at 15.5 min, 5:95 V/V at 17.3 min, 80:20 V/V at 17.3 min, 80:20 V/V at 20 min; flow rate, 0.35 ml/min; temperature, 45 °C; Injection volume: 2 µl. The effluent was alternatively connected to an ESI-triple quadrupole-linear ion trap (QTRAP)-MS. LIT and triple quadrupole (QQQ) scans were acquired on a triple quadrupole-linear ion trap mass spectrometer (QTRAP), QTRAP® LC-MS/MS System, equipped with an ESI Turbo Ion-Spray

interface, operating in positive and negative ion mode and controlled by Analyst 1.6.3 software (Sciex). The ESI source operation parameters were as follows: ion source, turbo spray; source temperature 500 °C; ion spray voltage (IS) 5500 V(Positive), −4500 V(Neagtive); ion source gas 1 (GS1), gas 2 (GS2), curtain gas (CUR) were set at 45, 55, and 35 psi, respectively; the collision gas (CAD) was medium. Instrument tuning and mass calibration were performed with 10 and 100 μM poly-propylene glycol solutions in QQQ and LIT modes, respectively. QQQ scans were acquired as MRM experiments with collision gas (nitrogen) set to 5 psi. DP and CE for individual MRM transitions was done with further DP and CE optimization. A specific set of MRM transitions were monitored for each period according to the metabolites eluted within this period. Relative levels of lipid metabolites were selected to pro-duce heatmap in R 4.1.2. All metabolites were listed in Supplemen-tary Data 1.

## 4D-label free proteomics and acetylome

Cartilage specimens were harvested as previous described ($n$ = 3 four each group, biological replicates), and proteins were extracted fol-lowing established methods[85]. Briefly, the samples were placed in a mortar containing liquid nitrogen and ground into a fine powder. Each sample was mixed with a lysis buffer [100 mM Tris-HCl, 8 M urea, 3 μM TSA, 50 mM NAM, 1 × protease inhibitor, pH 8.5] and sonicated indi-vidually. Following centrifugation (12,000 × $g$, 4 °C, 10 min), the resulting supernatant was aspirated to a fresh tube. After BCA assay, equivalent protein quantities from each sample underwent volume standardization using the lysis buffer and enzymatic digestion. Gra-dual addition of Trichloroacetic acid (Sigma-Aldrich) to a final con-centration of 20%, followed by gentle vortexing and precipitation at 4 °C for 2 h, was conducted. Subsequent steps involved centrifugation at 4500 × $g$ for 5 min, removal of the supernatant, and triple washing of the precipitate with pre-cooled acetone. The air-dried precipitate was reconstituted in a 200 mM solution of Tetraethylammonium bromide (Sigma-Aldrich), followed by sonication. Subsequently, it underwent overnight digestion with trypsin (Promega, enzyme: protein = 1:50, w/w). The solution was then reduced by introducing DL-Dithiothreitol (Sigma-Aldrich) to achieve a final concentration of 5 mM, incubated for 30 min at 56 °C. This process was finalized by adding iodoacetamide (IAM, Sigma-Aldrich) to reach a final concentration of 11 mM and incubating at room temperature for 15 min, shielded from light. For acetylated peptides enrichment, the peptide was suspended in the IP buffer [50 mM Tris-HCl, 100 mM NaCl, 1 mM EDTA, 1% (v/v) Nonident-P40, pH 8.0]. The supernatant was moved to pre-cleaned acetylation resin (PTM-104, PTM Bio) and gently shaken for overnight incubation at 4 °C. Subsequently, the resin was washed with the IP buffer and ultrapure water for 4 times and 2 times respectively. Afterward, 0.1% trifluoroacetic acid (TFA, Sigma-Aldrich) elution buffer was employed to release the resin-bound peptide for 3 times. The eluate was sub-jected to vacuum freeze-drying, desalted using C18 ZipTips, and then vacuum freeze-dried again for further analysis. The LC-MS/MS analysis was conducted using a timsTOF Pro mass spectrometer (Bruker) coupled to Nanoelute (Bruker Daltonics). The tryptic peptides were dissolved in solvent A, directly loaded onto a home-made reversed-phase analytical column (25-cm length, 100 μm i.d.). The mobile phase consisted of solvent A (0.1% formic acid, 2% acetonitrile in water) and solvent B (0.1% formic acid in acetonitrile). Peptides were separated with following gradient: 0–70 min, 6%–24%B ; 70–84 min, 24%–35% B ; 84–87 min, 35%–80%B ; 87–90 min, 80%B, and all at a constant flow rate of 450 nl/min on a NanoElute UHPLC system (Bruker Dal-tonics). The peptides were subjected to capillary source followed by the timsTOF Pro mass spectrometry. The electrospray voltage applied was 1.75 kV. Precursors and fragments were analyzed at the TOF detector, with a MS/MS scan range from 100 to 1700. The timsTOF Pro was operated in parallel accumulation serial fragmentation (PASEF) mode. Precursors with charge states 0–5 were selected for

fragmentation, and 10PASEF-MS/MS scans were acquired per cycle. The dynamic exclusion was set to 30 s. For acetylated peptides, the tryptic peptides were dissolved in solvent A, directly loaded onto a home-made reversed-phase analytical column (25-cm length, 100 μm i.d.). The mobile phase consisted of solvent A (0.1% formic acid, 2% acetonitrile in water) and solvent B (0.1% formic acid in acetonitrile). Peptides were separated with following gradient: 0–44 min, 6%–22% B ; 44–54 min, 22%–30%B ; 54–57 min, 30%–80%B ; 57–60 min, 80% B, and all at a constant flow rate of 450 nl/min on a NanoElute UHPLC system (Bruker Daltonics). The peptides were subjected to capillary source followed by the timsTOF Pro mass spectrometry. The electro-spray voltage applied was 1.7 kV. Precursors and fragments were ana-lyzed at the TOF detector, with a MS/MS scan range from 100 to 1700. The timsTOF Pro was operated in parallel accumulation serial frag-mentation (PASEF) mode. Precursors with charge states 0–5 were selected for fragmentation, and 10PASEF-MS/MS scans were acquired per cycle. The dynamic exclusion was set to 30 s.

The resulting MS/MS data were processed using MaxQuant search engine (v.1.6.15.0). Tandem mass spectra were searched against the Mus_musculus_10090_SP_20201214.fasta (17063 entries) concatenated with reverse decoy and contaminants database. Tryp-sin/P was specified as cleavage enzyme allowing up to 2 missing cleavages. Min. peptide length was set as 7 and max. number of modification per peptide was set as 5. The mass tolerance for pre-cursor ions was set as 20 ppm in first search and 20 ppm in main search, and the mass tolerance for fragment ions was set as 20 ppm. Carbamidomethyl on Cys was specified as fixed modification, and acetylation on protein N-terminal and oxidation on Met were speci-fied as variable modifications. False discovery rate (FDR) of protein, peptide and PSM was adjusted to <1%. For acetylated peptides, the resulting MS/MS data were processed using MaxQuant search engine (v.1.6.15.0). Tandem mass spectra were searched against the Mus_-musculus_10090_SP_20201214.fasta (17063 entries) concatenated with reverse decoy and contaminants database. Trypsin/P was spe-cified as cleavage enzyme allowing up to 4 missing cleavages. Min. peptide length was set as 7 and max. number of modification per peptide was set as 5. The mass tolerance for precursor ions was set as 20 ppm in first search and 20 ppm in main search, and the mass tolerance for fragment ions was set as 20 ppm. Carbamidomethyl on Cys was specified as fixed modification. Acetylation on protein N-terminal, oxidation on Met and acetylation on Lys were specified as variable modifications. False discovery rate (FDR) of protein, peptide and PSM was adjusted to <1%. And the acetylome data was normal-ized according to proteomics results.

The proteomics and acetylome datasets were provided in Sup-plementary Datas 2 and 3, respectively.

## Differentiation of 3T3-L1 adipocytes

3T3-L1 preadipocytes were cultured to 100% confluence. At day 0, cells were cultured in DMEM with 10% FBS and 1 μM insulin for 2 days. For the initiation of differentiation, the medium transitioned DMEM con-taining 10% FBS, 0.5 mM isobutyl-methylxanthine, 2.5 μM dex-amethasone and 1 μM insulin for 2 days, succeeded by incubation with 1 μM insulin for another 2 days. The achievement of differentiation was observed upon altering the medium to DMEM supplemented with 10% FBS for 4 days, aligning with prior literature findings[86].

## Conditioned medium system

The medium from the 3T3-L1 fibroblast or 3T3-L1 adipocyte was removed. The plates were gently washed twice with sterile PBS. Sub-sequently, each 10 cm dish received 4 mL of fresh DMEM containing 2% fatty acid-free BSA, incubating for 12 h. Following incubation, the conditioned medium was harvested, subjected to centrifugation (500 × $g$, 22–25 °C, 10 min), and then passed through a 0.22 μm filter to eliminate cellular debris. The conditioned medium was augmented

with 1% FBS and employed in chondrocyte micromass culture and 3D agarose culture.

## Free fatty acid preparation

Free fatty acid (FFA) stock solutions were formulated by conjugating free fatty acid with fatty acid-free BSA as reported[33]. In detail, 211.08 mg palmitic acid was dissolved in 4 ml ethanol at 70 °C, constituting the PA stock solution. Meanwhile, 464.51 mg oleic acid was dissolved in 8 ml of 0.1 mM NaOH water solution at 90 °C, serving as the OA stock solution. Subsequently, 0.4 ml of the PA stock solution and 0.8 ml of the OA stock solution were diluted in 15 ml of PBS containing 12%(w/v) BSA to produce the resultant FFA stock solutions, to give a final fatty acid concentration of 15 mM with an OA: PA ratio of 2:1. The stock FFA solutions underwent filtration through a 0.45 μm filter and were stored at 4 °C. Control solution were composed of equivalent ethanol, NaOH, and BSA but lacked lipids.

## Micromass culture

Micromass culture served as the method for assessing chondrogenesis and ECM deposition, consistent with previous studies[87,88]. Primary mouse chondrocytes were adjusted to a density of $2.0 \times 10^7$ cells/mL in culture medium. Micromass were established by dispensing 15 μl droplets of the mixture into individual wells of a 12-well plate. After 2-h attachment, 1 mL of culture medium with specified treatment was added. Micromasses were cultured for 1 week, with the medium and the treatment refreshed every 2 days. Harvesting involved conducting BCA protein assays or fixing the micromasses with 4% paraformaldehyde (PFA) and staining them with alcian blue at pH 0.2. Quantification of ECM deposition was performed using Image J v1.50, with the results normalized based on the protein concentration of corresponding wells.

## 3D agarose culture

3D agarose culture of primary mouse chondrocytes followed established methods[89]. Briefly, a mixture of 4% low melting agarose solution and 2× culture medium (comprising 4% FBS, 2% Hepes, and 2% penicillin-streptomycin) was prepared at a 1:1 ratio. Chondrocytes were then resuspended into this mixture, reaching a final concentration of $2 \times 10^6$ cells/ml. Subsequently, 500 μl of this mixture was transferred to individual wells of a 24-well plate, allowing half 20 min for gel formation. The culture medium and specified treatment were subsequently added to the wells and replenished daily. Following one week of culture, the agarose construct underwent fixation using 4% PFA, embedding in paraffin, sectioning at 5 μm, and staining with alcian blue and nuclear fast red. The thickness of the ECM was quantified by using Image J v1.50.

## Fatty acid uptake assay

The fatty acid uptake assay was performed according to the manufacturer's instructions. Briefly, primary mouse chondrocytes were seeded in 96-well plates, and pretreated with 300 μM FFA or BSA control for 24 h. The cells were then incubated in serum-free DMEM for 1 h, followed by replacement with fresh culture medium containing the TF2-C12 FA probe at a final concentration of 0.5 μM. Fluorescence was measured immediately (excitation: 485 nm, emission: 515 nm) using an Agilent Synergy H4 microplate reader (BioTek, USA). The fluorescence signal was recorded every 20 s for a total of 30 min and normalized to the protein level.

## Cell treatment

Cells were treated with 10 μM interleukin-1β (IL-1β), 50 μM tumor necrosis factor-α (TNF-α), FFA, and Trimetazidine (TMZ) as indicated. To inhibit deacetylation activity, cells were treated with 5 μM Trichostatin A (TSA) and 10 mM Nicotinamide (NAM) for 12 h. To escalate or diminish AMPK activity, cells received 50 μM A-769662 or 10 μM

Dorsomorphin respectively. Cells were harvested 48 h post-treatment unless otherwise indicated.

## Cyclic tensile stress of chondrocytes

Primary murine chondrocytes were introduced into silicone chambers coated with poly-D-Lysine, with each chamber accommodating a density of $5 \times 10^5$ cells. The dimensions of the culture surface within each chamber were $3 \times 3$ cm. Subsequently, a 24-h after seed, cyclic tensile strain at 0.5 Hz and 10% elongation was administered for 30–120 min using a CELL TANK system (CELL&FORCE, China) situated in a $CO_2$ incubator. Control cells were introduced into the same chambers and cultivated without exposure to tensile stress. Cells were cultured for another 24 h before harvest.

## Adenovirus and lentivirus infection

For adenovirus infection, indicated cells were cultured until reaching approximately 30–40% confluence and then exposed to either Cre or control adenovirus (HANBIO TECH, China) for 4 h, with $5 \times 10^7$ PFUs in 1 ml of medium. Following this, an additional 1 ml of medium was introduced, the cells were infected for an additional 4 h. Finally, the cells were then cultured in fresh medium.

For lentivirus infection, Human wild-type or mutated *HADHA* and *HADHB* cDNA were inserted into the pLVX vector and packaged. The *HADHA*-KO or *HADHB*-KO C28/I2 cells in a 10 cm dish were infected with specified lentivirus for 2 days. The cells were then cultured in fresh medium. Polybrene, at a final concentration of 10 μg/ml, was used for all virus infections.

## RNA Intervention

For small-interfering RNA (siRNA) transfection. Cells were seeded into 6-well plates and cultured until reaching 40% confluence. Following the manufacturer's instructions, siRNA transfection was carried out using Lipofectamine RNAiMAX. RNA and protein were extracted 48 h after transfection. The specified siRNA and control siRNAs were chemically synthesized by Genepharma, China.

To achieve stable silencing of *Hadha* and *Hadhb*, lentivirus infection was utilized. Short hairpin RNA (shRNA) sequences, designed based on the siRNA sequences, were subcloned into the PLKO.1 vector and packaged. Cells in a six-well plate were cultured until reaching 50% confluence and then infected with lentivirus for 2 day, with polybrene added at a final concentration of 10 μg/ml. The cells were then cultured in fresh medium.

## *HADHA* AND *HADHB* knockout in C28/I2

The specific gRNA sequences targeting *HADHA* and *HADHB* were designed by the online sgRNA Design Tool (Broad Institute, USA) and subcloned into the PEP-KO vector. The constructed plasmid was transfected into C28/I2 cells using the Hieff Trans® Liposomal Transfection Reagent according to the manufacturer's guidelines. Cells underwent puromycin selection 48-h post-transfection and were subsequently re-seeded in a 10-cm dish (100 cells per dish). The resulting single colonies were isolated and expanded. Immunoblot analysis and genome DNA extraction, followed by Sanger sequencing alignment were employed to verify gene knockout efficiency.

## Co-Immunoprecipitation

After cellular overexpression and exposure to specific treatments, cells were lysed with mild lysis buffer (MLB) [10 mM Tris-HCl, 2 mM EDTA, 100 mM NaCl, 50 mM NaF, 1 mM PMSF, 1× protease inhibitor cocktail and 1% (v/v) Nonidet-P40, pH 7.5]. After centrifugation (12,000 × *g*, 4 °C, 10 min), the supernatants underwent incubation with either Anti-DYKDDDDK IP Resin or Anti-HA magnetic beads, or specified antibody bind with protein A/G beads at 4 °C for 4 h. The resulting immunocomplexes were triple washed with MLB, boiled with 1× loading buffer, and then subjected to further analysis.

For acetylation-immunoprecipition, cells were pretreated as indicated, lysed in MLB buffer (containing 3 µM TSA and 50 mM NAM), sonicated, and centrifuged. Immunoprecipitation was conducted using Anti-DYKDDDDK IP Resin as described above.

For ubiquitination- or phosphorylation-immunoprecipitation, wild-type or mutated SOX9-Flag were overexpressed in C28/I2 cells using Hieff Trans® Liposomal Transfection Reagent. Twelve hours before harvest, cells received specified pretreatments. Eight hours pre-harvest, cells were exposed to 1 µM MG132. Cells were collected and boiled in MLB containing 1% SDS for 10 min. Subsequently, the cellular lysate was diluted with MLB to a concentration of 0.1% SDS. After centrifugation, the supernatants were incubated with Anti-DYKDDDDK IP Resin at 4 °C for 1.5 h, followed by washes, elution, and subsequent analysis.

## Identification of acetylation, phosphorylation, and ubiquitination sites

HADHA-HA or SOX9-Flag were overexpressed in C28/I2 cells. The cells underwent treatment with specific agents followed by Immunoprecipitation. Subsequently, proteins were eluted with 1× loading buffer and separated on 8% SDS-PAGE gel. Protein bands were visualized via Coomassie brilliant blue staining, and identical bands were cut into sections.

For in-gel tryptic digestion, the gel piece was dehydrated with 100% acetonitrile for 5 min, and incubated with 10 mM TCEP at 37 °C for 30 min. Then, the gel piece was again dehydrated with 100% acetonitrile and incubated with 25 mM iodoacetamide (chloroacetamide for ubiquitination sites identification[90]) at room temperature for 30 min in dark. After that, Gel piece was washed with 50 mM $NH_4HCO_3$ and dehydrated with 100% acetonitrile. Finally, the gel piece was rehydrated and digested with 2 µg trypsin in 50 mM $NH_4HCO_3$ at 37 °C overnight for protein in-gel digestion. After digestion, Peptides were extracted from the gel piece with 50% acetonitrile/0.1% formic acid. The extracted peptides were dried in SpeedVacuum concentrator and resuspended in 0.1% formic acid for LC-MS/MS analysis.

The peptide samples were dissolved in mobile phase A (0.1% formic acid) and separated by EASY nLC-1200 (Thermo Scientific). The Nano liquid chromatography gradient was kept at a constant flow rate of 400 nl/min and comprised of an increase from 2% to 7% mobile phase B (0.1% formic acid in 80% acetonitrile) over 1 min, 7% to 35% for 35 min, 35% to 55% for 9 min, climbing to 100% in 7 min, and held at 100% for the last 8 min. The isolated peptides were subjected to Nano source followed by Q Exactive HF-X mass spectrometer. The electrospray voltage applied was 2.0 kV, and intact peptides were detected in the Orbitrap at a resolution of 60,000. Peptides were then selected for MS/MS using NCE setting as 27 and the fragments were detected in the Orbitrap at a resolution of 30,000. The Mass Spectra were acquired in data-dependent scan mode included selection of the 20 most abundant precursor ions of each MS spectrum for MS/MS analysis with 20 s dynamic exclusion.

Mass spectra underwent processing and search procedures using Proteome Discoverer 2.4 against the human Swissprot protein database (release 2023_09). Trypsin/P or relevant enzymes were used as cleavage enzymes allowing up to 2 missing cleavages. Precursor ions had a mass tolerance of 10 ppm, while fragment ions had a tolerance of 0.01 Da. Carbamidomethyl on Cysteine were specified as fixed modification, while oxidation on Methionine, acetyl on Lysine and protein N-terminal, phosphorylation at Serine/Threonine/Tyrosine and ubiquitination at Lysine were designated as variable modifications.

## In vitro acetylation assay

*HADHA*-KO C28/I2 cells overexpressed with HADHA-Flag underwent immunoprecipitation as described above. Subsequently, the resulting immunocomplexes were triple washed with MLB and suspended in PBS containing indicated concentrations of acetyl-CoA. Incubated at

37 °C for 1 h, the immunocomplexes were boiled in 1× loading buffer and subjected to analysis via western blotting.

## Anti-HADHA-K728ac antibody preparation

HADHA-K728ac-specific antibody was prepared by Huaan Biotechnology (Hangzhou, China). The HADHA-K728ac peptide (CFVDLYGAQK[ac]IVDRL) was first coupled to keyhole limpet hemocyanin and BSA, followed by multiple subcutaneous injections to immunize New Zealand white rabbits. Blood was then collected, and the presence of HADHA-K728ac-specific antibodies in the serum was evaluated using enzyme-linked immunosorbent assay (ELISA) and western blot. For final antibody purification, the serum was eluted through the HADHA-K728ac peptide column and then passed through a non-HADHA-K728ac peptide column to ensure specificity.

## Immunoblot analysis

Cartilage protein was extracted as described above. Cell protein was extracted using the pkill buffer [50 mM Tris-HCl, 1% SDS, pH 6.8] and then boiled. Protein concentrations were measured using BCA assay. Equivalent protein amounts were separated via SDS–PAGE gels and transferred to nitrocellulose membranes. The membranes underwent a 1-h blocking and subsequent incubation with primary and secondary antibodies. Immunoblot detection was performed using FDbio-Dura ECL regents.

For proteolysis investigation, cells were pre-treated with either 1 µM MG132 or 10 µM chloroquine for 30 min. After a 12-h pre-treatment with FFA, cells were exposed to 1 µM cycloheximide (CHX) for the indicated duration, and subsequently harvested for immunoblot detection.

## Nile red and Bodipy staining

Due to xylene's ability to dissolve lipids within tissues, we employ a Specialized deparaffinized Reagent kit designed for lipid staining following the manufacturer's protocols. Subsequently, sections were stained with 5 µM BODIPY 493/503 or 1 µM Nile red for 10 min at 20–25 °C, avoiding light exposure, and mounted. Representative images were captured using a fluorescence microscope (Olympus, Japan), and the relative fluorescence intensity was quantified by using Image J v1.50.

## Computational modeling and molecular docking

The HADHA protein's structure underwent a homology search utilizing the Human HADHA amino acid sequence, resulting in a protein structure displaying 100% homology and similarity sourced from the protein database (PDB entry: 6DV2). This served as the foundation to construct wild-type and mutant or acetylated HADHA protein models for subsequent substrate docking studies. The initial structures underwent processing using AutoDock Tools 1.5.6[91]. maintaining the native charges of the proteins for subsequent docking analyses. The chemical structure of the substrate, β-hydroxybutyryl-CoA, was obtained from PubChem and optimized using MOPAC[92]. Subsequently, it was docked at the interaction region as observed in the crystal structure of HADHA protein (PDB entry: 6YSW) using AutoDock 4.2.6[93]. Docking was energy-optimized, and the solvent accessible surface area (SASA) and the binding energy were calculated.

The protein structures of wild-type and phosphorylated SOX9, along with TRIM9, were retrieved from the protein database and optimized using I-TASSER[94]. The docking between SOX9 and TRIM9 was conducted using HDCOK[95,96]. Docking was energy-optimized, and the docking score was calculated based on the ITScorePP or ITScorePR iterative scoring function.

## HADHA enzyme activity assay

The 3-Hydroxyacyl-CoA dehydrogenase activity of HADHA was assessed based on previous studies[46,97,98] and subsequently optimized.

HADHA-Flag proteins were purified via Immunoprecipitation from *HADHA*-KO C28/I2 cells overexpressed with either wild-type or mutated HADHA-Flag as described above. In specified instances, cells were pre-treated with designated agents. The immunocomplexes were eluted using 3 × Flag peptide in a reaction buffer [75 mM Tris-HCl, 75 mM KCl, pH 10]. A portion of the proteins was boiled with 1/4 volume of 5× loading buffer for HADHA-Flag detection via immunoblotting, while the remaining proteins were adjusted to an equal concentration in the reaction buffer, guided by the immunoblot results. Each well contained 160 μl of reaction buffer with an equivalent amount of HADHA-Flag proteins, the reaction was initiated by adding 40 μl of substrate buffer [125 μM β-hydroxybutyryl-CoA, 12 mM NAD, 200 μM 1-MPMS, and 5 mM WST-8]. The NADH production was tracked at 450 nm in a Synergy H1 Multimode Reader (Agilent). In the presence of 1-mPMS, WST was efficiently reduced by NAD(P)H to produce a formazan product, assessed by absorbance at 450 nm. Thus, the absorbance at 450 nm corresponds directly to the NADH concentration. Control assays were conducted without overexpression in *HADHA*-KO C28/I2 Cells.

## Extracellular flux analysis

Oxygen consumption rate (OCR) was measured using a Seahorse XFe96 Extracellular Flux Analyzer (Agilent) following manufacturer's guidelines. Briefly, C28/I2 cells were plated at a density of $5 \times 10^4$ cells per well and cultured overnight. Subsequently, cells were initially incubated in XF DMEM (Agilent) containing 25 mM glucose, 1 mM pyruvate, and 2 mM L-glutamine(GIBCO) for one hour at 37 °C without $CO_2$. To measure OCR, 4 μM Etomoxir (Eto), 1.5 μM oligomycin A (oligo), 1 μM FCCP, and R&AA (1 μM and 5 μM, respectively) were loaded as indicated.

For FAO assessment, XF Palmitate-BSA FAO Substrate (Agilent) was employed. Cells were cultured in the substrate-limited DMEM containing 0.5 mM glucose, 1 mM L-glutamine, 0.5 mM carnitine, and 1% FBS overnight. The next day, the medium was replaced with XF DMEM containing 0.5 mM glucose, 0.5 mM carnitine, and 200 μM Palmitate-BSA. The subsequent procedures were performed consistently and as described above. Data were collected and analyzed using Wave 2.4.0 (Agilent), with the results normalized based on the protein concentration of each well.

For mitochondrial fuel oxidation analysis, cells were pretreated with FFA or BSA for 24 h, followed by incubation in XF DMEM containing 25 mM glucose, 1 mM pyruvate, and 2 mM L-glutamine for one hour at 37 °C without $CO_2$. OCR measurements were taken with the sequential addition of 2 μM UK5099, 3 μM BPTES, and 4 μM Eto as indicated. FAO capacity was calculated according to the formula (1):

$$\text{Capacity\%} = \left(1 - \frac{\text{Baseline OCR} - \text{Other 2 inhibitors OCR}}{\text{Baseline OCR} - \text{All inhibitors OCR}}\right) * 100 \quad (1)$$

## Cellular ATP determination

Specific C28/I2 cells were cultured in the substrate-limited DMEM containing 0.5 mM glucose, 1 mM L-glutamine, 0.5 mM carnitine, and 1% FBS overnight, and then treated with either FFA or BSA control for 12 h. Subsequently, the medium was aspirated and cells were triple washed using PBS. The cellular ATP level was determined using the ATP Chemiluminescence Assay Kit following the manufacturer's protocols. Additionally, cell lysates underwent the BCA assay to quantify protein concentration. The obtained results were normalized according to the protein concentration. The ATP produced by FAO was deduced by subtracting the results of the BSA group from those of the FFA group.

## Immunocytochemistry and confocal microscopy

Specified cells were cultured and pre-treated for 24 h on glass coverslips. The cells were then fixed with 4% PFA at room temperature for 15 min, and permeabilized with 0.1% Triton X-100. After blocking with 3% BSA, cells were incubated with primary antibodies at room temperature for one hour. Subsequently, after triple washes using PBS. Cells were incubated with secondary antibodies at room temperature for another hour. Triple washed with PBS, cells were mounted with antifade mounting medium with DAPI. For Nile red and Bodipy staining, cells were fixed, washed, directly stained with 1 μM Nile red and 1 μM BODIPY 493/503 at 20–25 °C for 10 min and mounted. Imagines were captured using a Nikon A1R confocal microscope and analyses were carried out using Image J v1.50.

## Measurement of FFA, TG and cholesterol levels

Human synovial fluid was obtained and processed as described above. Under the manufacturer's instructions, The NEFA assay kit, Triglyceride Assay Kit, Total Cholesterol Assay Kit, and Free Cholesterol Assay Kit were utilized to measure the concentrations of total FFAs, triglycerides, total cholesterols and free cholesterols, respectively, in the synovial fluid.

## Acetyl-CoA assay

The acetyl-CoA concentrations were determined according to Elabscience's protocol. Cartilage samples were ground into a powder using a mortar cooled with liquid nitrogen. Each tissue sample was weighed, mixed with PBS (100 μl per 100 mg tissue), sonicated, and centrifuged ($12,000 \times g$, 4 °C, 15 min). The resultant supernatants underwent acetyl-CoA assays, normalized against tissue weight. For cell samples, $5 \times 10^6$ specified cells were suspended in 500 μl PBS. Subsequently, following sonication and centrifugation, the supernatants were gathered for acetyl-CoA assays and normalized based on cell counts.

## $^{13}$C-labeled acetyl-CoA detection

Mice were fed and operated following the protocol outlined in Fig. S2a. Cartilage tissues were harvested and processed into a powder as described above ($n = 4$). Each tissue sample was weighed, and mixed with 400 μl 25% methanol-water solution containing 1 ng/ml 4-chlorophenylalanine (4-CPA) as the internal standard. Subsequently, the samples were sonicated and centrifuged ($12,000 \times g$, 4 °C, 15 min.). Finally, 40 μl of the resulting supernatant underwent UPLC-MS/MS for detailed analysis and the results were normalized against tissue weight.

The LC-MS/MS analysis was performed using a TSQ Quantis™ mass spectrometer (Thermo Fisher Scientific) coupled with a Vanquish UPLC (Thermo Fisher Scientific). LC separation was achieved on an Atlantis Waters XSelect® HSS T3 cloumn (2.1 mm×100 mm, 2.5μm). Sample aliquot was injected with the column temperature maintained at 30 °C, the flow rate was set at 0.3 ml/min. Mobile phase A was 2.5 mM ammonium acetate in $H_2O$ and Mobile phase B was acetonitrile. The elution gradient started with 2% phase B: 0–5 min: 2%–95%B; 5–7 min: 95%B; 7–7.1 min: 95%–2%B; 7.1–10 min: 2%B. The sample manager temperature was set at 4 °C. Experiments were conducted in positive ionization mode and quantification was carried out in multiple reaction monitoring mode. Mass spectrometric conditions, including: collision energy (CE) and RF lens voltage were optimized automatically. The mass spectrometric conditions for analytes and the internal standard were shown in Supplementary Table 7. The ion source conditions were listed as follows: Spray voltage, 4000 V; the ion transfer tube and the vaporizer temperature, 350 °C; sheath gas and auxiliary gas of nitrogen (purity ≥99.99 %) at the flow rate of 35 arb and 10 arb, respectively. Data were analyzed using the Thermo Scientific Xcalibur workstation (Version 4.5.445.18).

### Cleavage under targets and tagmentation (CUT&Tag) assay

The Hyperactive Universal CUT&Tag Assay Kit for Illumina was used adhering to the manufacturer's instructions. In summary, fresh cultures were harvested and $5 \times 10^5$ cells were counted for each sample, followed by nucleus extraction. The nucleus was then bound to Concanavalin A beads for 10 min at 22–25 °C. Next, the liquid was aspirated, and the nucleus was incubated with the primary antibody in Antibody Buffer at 4 °C for overnight. Next, the liquid was aspirated, and the secondary antibody was added with Dig-Wash Buffer, incubating for 1 h at 22–25 °C. Washed CUT&Tag pA/G-Tn5 Transposomes were bound for 1 h at 22–25 °C. Post-washing, tagmentation occurred at 37 °C for 1 h, followed by mixing with DNA spike-in (1 pg/105 cells) at 55 °C for 10 min, and subsequent DNA extraction using DNA extraction beads. Finally, i7 and i5 Indexed Primers were combined for PCR amplification, followed by cleanup for quantification and Illumina sequencing. The raw data was trimmed by fastp (v0.20.1) to remove adapter sequences and low-quality reads. The clean reads were aligned to the mouse genome (mm10) using Bowtie2 (v2.2.5) with following options: −no-unal −no-mixed −nodiscordant −local −very-sensitive-local −phred33 -I 10 -X 700. For peak calling, we merged timepoint replicate BAM files before using macs2 (v2.1.2) callpeak: -p 1e − 5 -f BAMPE −keep-dup all -g mm. The peak visualization in the genome and chromosomal distribution was shown by Integrative Genomics Viewer (v2.6.3) and Circos (v0.69-6), respectively. The bigwig files were generated with bamCoverage function in deepTools (v3.4.3) with default parameter settings. The peak annotations were performed with R package ChIPseeker (v1.28.3). At each reprogramming timepoint, the co-localized sites were defined as 100 bp intersecting the extended summits between two histone modifications. R package DEseq2 (v1.36.0) was used to perform the differential analysis for histone modification peaks. The differentially expressed gene dataset was used for GO and KEGG enrichment analyses using DAVID (v2023q4). Statistical significance was determined using Fisher's exact test followed by Benjamini–Hochberg (BH) correction. The CUT&Tag-sequencing data was listed in Supplementary Data 4.

### Quantitative PCR analysis

Total mRNA was extracted utilized TRIzol and the SteadyPure Universal RNA Extraction Kit following respective manufacturer protocols. Subsequent cDNA synthesis employed Evo MMLV RT Premix. Real-time qPCR and CUT&Tag-qPCR employed SYBR Green Master Mix in the QuantStudio 5 Flex Real-Time PCR System (Thermo Fisher Scientific), following manufacturer guidelines. The Ct values of each gene was normalized using the Ct value of β-actin for Real-time qPCR or DNA spike-in for CUT&Tag-qPCR. Primers were obtained from Tsingke and detailed in Supplementary Table 3.

### Isolation of RBC-exosomes (RBC-Exo) and production of CAP-RBC-Exo/TMZ

RBC-Exo were obtained and then conjugated with CAP peptides using an established method[99,100]. Initially, RBCs were isolated from plasma via centrifugation ($500 \times g$, room temperature, 10 min) and passed through a leukodepletion filter. Isolated RBCs were cultured in RPMI-1640 medium and treated with 2 mM calcium ionophore for 2 days. Following this, RBCs and cell debris were removed by gradient centrifugation ($600 \times g$, 4 °C, 10 min), ($2000 \times g$, 4 °C, 20 min), and ($100,000 \times g$, 4 °C, 30 min). RBC-Exo were pacificated using ultracentrifugation with an SW32Ti rotor (Beckman Coulter, $100,000 \times g$, 4 °C, 70 min). Finally, the Exo were dissolved in sterile PBS and stored at −80 °C.

For the production of CAP-RBC-Exo/TMZ. DSPE-PEG2000-Mal (3.981 mg) and CAP-Cys peptide (>95% purity, 3 mg) underwent an overnight reaction at pH = 6.8–7.4. Employing dialysis tubing with an 8000 Da cutoff facilitated the removal of free CAP-Cys and salt ions, yielding DSPE-PEG2000-CAP peptide (CAP amino acid sequence:

DWRVIIPPRPSA). Liposome compounds were synthesized using an alkyl epoxy compound reaction with amines. A 20 mL glass vial containing a magnet hosted the reaction of 1,3-propanediamine (1 mmol) and 1,2-epoxyhexane (½N − 1 equivalent, N=sum of secondary amines plus twice the number of primary amines in the starting material) underwent stirring at 90 °C for 2.5 days. The crude reaction product underwent purification through $CH_2Cl_2$/methanol/$NH_4OH$(aq) (75 : 22 : 3), resulting in the desired liposomes. The previously prepared liposomes (14.93 mg), along with DOPC (11.321 mg), cholesterol (16.182 mg), DMG-PEG2000 (1.99 mg) dissolved in ethanol (10 mL) underwent rotary evaporation (37 °C, 20 min). The resulting film was hydrated with the prepared DSPE-PEG2000-Mal-CAP solution and TMZ drug (3.329 mg) in 1 × PBS, generating CAP-Lipo/TMZ. Subsequently, CAP-Lipo/TMZ and 500 μL RBC-Exo underwent low-speed vortexing for 30 s, extruded sequentially through 400 nm, 200 nm PC membranes, dialyzed in 8000KDa PBS buffer to remove free TMZ, and finally dried at 2–8 °C, culminating in the final product of CAP-RBC-Exo/TMZ. Control experiments replicated the CAP-RBC-Exo production steps without TMZ.

### Statistics & reproducibility

For in vitro experiments, cultures were randomly assigned to different treatments. For mouse experiments, mutant and WT littermates were grouped based on sex, age, and genotype. Male mice were selected for the experiment in order to avoid concerns about hormonal effect in female mice. Mice were randomly allocated to high-fat diet, DMM surgery, or intra-articular injection groups without subjective judgment. Blinded individuals performed cartilage destruction analysis, synovial inflammation scoring, and immunohistochemical or immunofluorescence analysis, unaware of the specific mouse strains, treatment groups, or control versus experimental status. While no statistical methods predetermined sample sizes for in vitro and in vivo analyses, preliminary experiments estimated assay variances to determine sufficient sample sizes, which are similar to those reported in other publications. No data were excluded from the analyses. Animals were closely monitored for persistent lameness and discomfort post-surgery, with no animals meeting the criteria for exclusion from the study. All experiments were performed independently at least three times. Data were presented as mean ± SD or SEM, or as minimum to maximum, and were analyzed using GraphPad Prism 9.2.1. Statistical analysis was performed using unpaired two-tailed Student's $t$ test for two groups, or ANOVA followed by multiple comparation tests for multi-group comparison, or nonparametric test for converted data that did not adhere to a normal distribution. Kyoto Encyclopedia of Genes and Genomes (KEGG) analysis and Gene Ontology (GO) analysis were performed using DAVID (v2023q4). Detailed statistical information for all experiments is provided in the figure legends.

### Reporting summary

Further information on research design is available in the Nature Portfolio Reporting Summary linked to this article.

## Data availability

All data needed to evaluate the conclusions in the paper are present in the paper and/or the Supplementary Materials. The lipidomic data generated in this study have been deposited in the OMIX database under OMIX008076. The proteomics and acetylome data generated in this study have been deposited in the iProX database under IPX0011615000. The CUT&Tag-seq data generated in this study have been deposited in the GSA database under CRA020878. The electronic structure calculations data generated in this study have been deposited in the Figshare database under [https://doi.org/10.6084/m9.figshare.28741640.v1]. Source data are provided with this paper.

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

## Acknowledgements
We thank Bin Zhao and Yingsheng Zhang for discussing and revising this manuscript. We thank Chen Pan, Zhexun Chi, and Sheng Chen from the Core Facilities, Zhejiang University School of Medicine for their technical support. This work was supported by the National Natural Science Foundation of China (grant number 82422044 and 82272522 to S.S., 82330077 and U21A20351 to S.F., U22A20282 to X.F., and 3230050202 to H.Z.), the Natural Science Foundation of Zhejiang Province (grant number LR22H060001 to S.S. and Q23H060019 to H.Z.), and the "Pioneer" and "Leading Goose" R&D Program of Zhejiang (grant number 2023C03091 to S.F.). We would also like to thank Editage (www.editage.com) for English language editing.

## Author contributions
S.S. and S.F. designed and supervised the research. H.Z. cosupervised the study and revised this manuscript. Z.M., K.Y., W.N., and P.S. conceived, designed, and performed most experiments. Z.M. wrote the manuscript. N.P., H.C., Y.S., L.G., and D.H. performed the experiments. Z.M. and H.Z. analyzed data and performed bioinformatic analyses. K.Y. and W.N. assisted with conducting the in vivo experiments. P.S. and H.C. performed bioinformatic analyses. Q.S., Z.L., and X.F. performed pathological analysis of recipient mice.

## Competing interests
The authors declare no competing interests.
