## [Peer review file · Nature Communications]

Chondrocyte fatty acid oxidation drives osteoarthritis via SOX9 degradation and epigenetic regulation

Corresponding Author: Professor Shuying Shen

Version 0:

Reviewer comments:

Reviewer #1

(Remarks to the Author)

Overall Consideration:

In the manuscript, "Fatty acids-mediated acetylation drives osteoarthritis via fatty acid oxidation-induced SOX9 degradation and epigenetic regulation" the authors show that high fat diet feeding leads to increased fatty acid levels in articular cartilage, acetyl-CoA accumulation, and exacerbates development of OA. The fatty acid oxidation enzyme HADHA was identified as differentially regulated following high fat diet feeding. The authors then perform genetic studies to assess the role of HADHA during OA development and progression. These studies are complemented with in vitro work demonstrating that fatty acid oxidation leads to proteasome mediated degradation of Sox9. Strengths on the manuscript include high rigor of the studies performed. The in vitro data describing the cellular mechanisms are particularly compelling and well executed. Likewise, human translational data are included within the work, supporting the negative impact of fatty acid accumulation on joint health. Delivery of a FAO inhibitor to the joint also prevented OA development in a preclinical model. Weaknesses of the manuscript include oversights of technical tissues within the genetic studies. The negative impact of long chain saturated fatty acids on joint health, the role of Sox9 in FAO by chondrocytes, and epigenetic alterations of chondrocyte transcription by acetyl-CoA accumulation are likewise already described within the literature. Weakness of the manuscript are discussed below in a point-by-point fashion:

Major comments:

1. The main concern with the manuscript is the choice of controls for the genetic experiments using the Col2-CreERT and the Agc-CreERT. The Agc-CreERT mouse has a basal phenotype induced by generation of an Agc hypomorphic allele after tamoxifen administration (PMID 28921880). In the work described within the manuscript, the authors use HADHA^{fl/fl} Cre-animals for comparison. Col2-CreERT is thought to be only active during development. At the very least, the authors need to acknowledge these limitations.
2. The mechanistic data showing that FAO alters SOX9 stability are well developed, but may represent an incremental advancement.
3. Are there changes within the synovium or to the meniscus conferred by HADHA deletion and/or FAO inhibition?

Minor Comments:

1. Typo, line 611: "tow" should be two; double check line 838: "1Xa loading buffer"
2. The order of the pictures showing the histology for the DMM experiments and the bar graphs summarizing the scoring is not consistent. This caused some confusion while reading. I would suggest keeping the order of presentation the same – group either like genotypes or like surgeries.

Reviewer #2

(Remarks to the Author)

In this manuscript, Mei et al. elucidate how obesity and osteoarthritic factors synergistically drive increased fatty acid uptake by chondrocytes, promoting fatty acid oxidation (FAO) and resulting in acetyl-CoA accumulation and alteration of protein acetylation profiles. This process leads to the suppression of AMPK signaling and facilitates SOX9 degradation and osteoarthritis (OA). In addition, the authors further explored the therapeutic potential by targeted delivery of trimetazidine, an FAO inhibitor and AMPK activator, in OA treatment. Overall, the study presents a substantial amount of data to address an important question in the OA field. To improve the quality of this manuscript, I have several comments and suggestions listed below.

1) Previous studies have demonstrated the significance of metabolism-mediated acetylation in hepatic cells, enhanced OA pathology by fatty acid synthesis in aging chondrocytes, and increased FAO accelerating SOX9 degradation, what is the novel information provided by the current study.

2) The authors concluded that free fatty acid (FFA) is an important metabolite contributing to obesity-associated cartilage degeneration (Fig. 1b-1c, s1b). To further support this conclusion, *in vivo* experiments are needed to demonstrate similar cartilage changes in normal diet (ND) mice treated with FFA arthrocentesis.

3) Figure S1I uses the WOMAC score to show that obese patients have more severe clinical symptoms compared to lean patients, reflecting the correlation between obesity and OA severity. However, this is not directly related to the main objectives of the study, and other parts of the research do not address pain assessment. Therefore, I would suggest the authors to remove this data.

4) Figure 4m showed that TRIM9 knockdown effectively elevated SOX9 protein levels. However, it is not known if the similar results could be obtained under high-fat diet (HFD) stimulation?

5) Figure 5e shows a significant decrease in H3K27ac at 3 peaks in the *Acan* gene locus, but only 2 of them are confirmed by CUT&Tag-qPCR as shown in Fig. S6c. A similar issue is also related to the *Col2a1*, *Pdk4*, and *Mmp13* gene loci. What is the reason for this discrepancy? Additionally, gene symbols in Fig. 5e should be italicized, and this change should be applied consistently throughout the manuscript.

6) The authors stated that fatty acid accumulation and OA manifestations could be more pronounced in the HFD-Sham group with prolonged HFD feeding. Actually, previous reports have shown that a 30-weeks HFD is sufficient to promote cartilage degeneration and intracellular lipid deposition in mice (J Orthop Res. 2022, PMID: 35279877).

7) The authors need to explain why cholesterol levels remained relatively stable among the groups (line 489-490)?

8) Why did the authors sacrifice Hadha and Hadhb knockout mice 12 weeks after DMM surgery, but sacrifice Hadha K728R knockin mice 8 weeks after DMM surgery. What is the rationale for this experimental design?

9) TMZ specifically inhibits the TFP complex, and CAP-RBC-Exo/TMZ is expected to alleviate cartilage destruction in HFD-DMM-induced OA mice. However, similar efficacy was also observed in ND-DMM mice (Fig. 6b-d). The authors need to explain this finding.

10) Why is human blood used for RBC-Exo generation, instead of mouse blood, given that it is applied to a mouse model?

11) All H3K27ac peaks (chromosomal location) and assigned gene should be provided as supplemental data.

12) All differential expression protein data and all normalized differentially acetylated proteins and sites data should be provided as supplemental data.

13) Line 32: "obesity-associated OA" should be changed to "metabolism-associated post-traumatic OA" to ensure consistency with the discussion section.

14) The right panel of Fig. S2I needs adjustment as the x-axis appears to be misaligned.

15) A recent publication by Wei et al. "Risk of metabolic abnormalities in osteoarthritis: a new perspective to understand its pathological mechanisms" Bone Res. 2023 Dec 6;11(1):63. is closely related to the current study and need to be cited.

Reviewer #3

(Remarks to the Author)

The authors report that increased fatty acid oxidation in articular chondrocytes disturbs their energy status leading to SOX9 degradation and that the increased acetyl-CoA levels also epigenetically enhance catabolic pathways. These novel findings certainly increase our insight in the role of diet on the development of osteoarthritis. The strength of the manuscript is that both *in vivo* and *in vitro* models are used to substantiate the conclusions. One aspect that is less corroborated is the (causative) link between fatty acid supply, cellular energy status and SOX9 degradation.

The authors report that adding FFA increases FAO and OXPHOS (Figure S2k): basal OCR is increased by FFA, but this is not blocked by Etomoxir, indicating that the FFA-induced increase in OCR is not due to CPT-1 inhibition and questioning the contribution of FAO in basal condition. Etomoxir has only an effect during maximal respiration in the presence of FFA, suggesting that FAO is used in this condition. Prove should be provided that FAO-linked OXPHOS is increased after FFA supply. The same comment applies to the HADHA-K728 mutant.

The authors highly focus on AMPK activation to explain the regulation of SOX9. Although some of the data report changes in pAMPK, other data only show an effect on p-ACC, which is regulated by pAMPK but also by several other parameters including citrate, long chain FFA and hormones. Therefore, an effect on p-ACC levels is not similar to energy status, but likely reflects an effect on *de novo* lipogenesis. Either the data should be interpreted in this way or levels of pAMPK,

reflecting energy status, should be provided for the different conditions tested (Fig S4c, d, i).

How do the authors explain that the overlap of FFA-induced phosphorylation sites of SOX9 and A-769662 (AMPK)-induced phosphorylation sites is rather low (Fig 4g), as these data indicate that FFA-induced effect on SOX9 levels is different from AMPK/energy status.

To further prove that FFA supply regulates SOX9 levels through FAO, the authors should show that the effect of FFA supply has no effect when Hadha gene expression/activity is blocked.

The authors report that adding free fatty acids (FFA) changes chondrocyte functioning (Figure 1a-c), but the concentrations of FFA used (between 100-500 μ M) are rather high as concentrations higher than 200 μ M can become lipotoxic for some cell types. The authors should therefore provide evidence that the used FFA concentrations do not induce toxicity and that the observed effects are specific. In addition, Western Blots should be quantified, not only in Figure 1 but in all the experiments.

Immunohistological analysis (Col2, MMP13, ACC,...) forms an important part of the study, but no information is provided on how the intensity is quantified and whether also the abundance of the signal was quantified. Related to the latter aspect, it is rather surprising that Col2 staining of the articular cartilage is very weak, detected only at certain spots and often intracellular (nuclear), also in the control condition (Sham, normal diet). Normally you should expect positive staining of almost the entire extracellular matrix. Staining should be reassessed, as this information is important for the conclusions.

The authors demonstrate that adding FFA results in increased number of lipid droplets and increased gene expression of Lipe and Cd36 and conclude that FFA uptake is increased (Fig 1d and Suppl fig 1c). However, these findings do not prove undeniably that FFA uptake is increased, as lipid droplets can be formed by many mechanisms.

The lipidomic analysis shows increased FA in the HFD-CCM (Fig S1a). Are these FA found in triglycerides (lipid droplets) or more as phospholipids or other types of lipids which do not function as storage but rather as membrane component, signaling or energy source.

Minor

No information is provided on which ¹³C-Fatty Acids are used in the in vivo model (Fig 2a).

Validation of gene deletion and levels of overexpression of Hadha or Hadhb should be provided, both for the in vitro and in vivo models.

Reviewer #4

(Remarks to the Author)

The manuscript describes the correlation between fatty acid and the change in metabolic pathway (acetylation, fatty acid oxidation, etc) that induces the severity of osteoarthritis. After careful consideration, there are some issues which authors can consider:

1. The role of some compounds, such as acetyl-CoA, free fatty acid, cholesterol, etc, have been widely reported in various scientific literatures, including the detail mechanism of those pathways. Thus, what is the crucial aspect of this manuscript compared to others? Does this manuscript provide new information regarding OA pathogenesis or progression?
2. The authors stated that cholesterol levels remained stable in OA induced by HFD. However, it is known that the elevation of cholesterol level plays a crucial role in OA. How would you explain your findings regarding this issue?
3. Why did the authors use OA:PA ratio of 2:1 for FFA solutions? Also, is there any difference in FFA-induced OA if the type/chain number of FFA is different?
4. Figure 1f: it is unclear if the mice subjected to ND and HFD diet was carried out in weeks or months. The authors should put that information in the figure itself or in the caption. Please recheck also the text as the authors said that the "mice subjected to a 16-week HFD diet."
5. The data shown in the manuscript is too much for one figure. Please reselect the main data and put some data in the supporting information. Also, please consider the font size or clarity of the font in the figures. For example, p values can be just marked by different asterisk, and the actual values can be mentioned in the captions or text.
6. The resolution of the figures is low. Please ensure the clarity of the figures.
7. There are many typos in the manuscript. Please revise it.

Reviewer #5

(Remarks to the Author)

This study by Mei et al. investigates the role of fatty acid oxidation (FAO) in OA. The authors report considerable in vitro results implicating the mechanisms of fatty acid induce FAO mediates acetylation of HADHA, which modifies the AMPK activity leading to degradation of SOX 9 and increases production of cartilage matrix degradation enzymes. They confirm their finding in vivo using tissue-specific KO of Hadha, Hadhb, and Hadha k728R conditional knock in mice. Finally, target delivery of Trimetazidine (TMZ) in the joint to alleviate the OA. Overall, this is a comprehensive and well-designed study that presents a large body of work. However, there are concerns.

Major concern

- 1.OA caused by obesity/HFD is different from OA caused by injury. Combining the two is not a good model and could have exacerbated the finding. Fig. 1 clearly showed that HFD+ DMM accumulates lipid in joints compared to HFD+sham and ND+sham.
- 2.Does loss of Hadha/- prevent mice from developing OA on HFD?
- 3.It is not clear why only palmitate was chosen when other pro-inflammatory FFAs are known to play roles in obesity/HFD-derived pathways.
- 4.Fig. 2. C14 labeling data was presented for ND+DMM and HFD+DMM. What about ND and HFD alone?
- 5.Figure 3 invitro studies, cells were not treated with FFA?

Other concerns

How long were the cells fed with FFA in Fig. 1 and elsewhere?

Fig. 2, panel D, is misrepresented.

Fig. 2 panel H: How was HADHA k728acetylation detected in cartilage tissue? HADHA-specific antibodies were generated? For in vitro studies, the authors mentioned that they used general K728 antibodies. Is this true for tissues too?

Fig. 3 and in other figures. What are lane 1 and 2?

Page 8 line 2..upregulated SOX9 protein expression. Word expression is inappropriate here.

Figure panels should have capital alphabets. For example, Figure 2I should be Figure 2L. And also, the panel should follow an order. It is confusing.

Reviewer #6

(Remarks to the Author)

The manuscript "Fatty acids-mediated acetylation drives osteoarthritis via fatty acid oxidation-induced SOX9 degradation and epigenetic regulation" is well written and contains a wealth of interesting data and results. It is easy to follow and results are presented in a clear and insightful way.

Several experiments use mass spectrometry-based bottom-up proteomics to identify differences in acetylation of proteins in chondrocytes as a consequence of a high fat diet (HFD).The authors used 4D label-free proteomics to identify 72 differentially regulated proteins in medial meniscus model (DMM) mice exposed to HFD. The identification of the acetylene identified 3005 acetylated sites in 1731 proteins of which 379 were significantly regulated. The mitochondrial trifunctional subunit alpha (HADHA) protein was over-represented in the enrichment.

Major concerns:

Figure 5g shows only the phosphorylation sites but does not indicate the results for the ubiquitination sites. Please include the data for the ubiquitination of the lysine sites.

The description of the UPLC-MS/MS analysis p24I971ff is too short: What were the MS/MS analysis conditions and how were 13C acetylated peptides distinguished from 12C acetylated peptides? The mass difference is 1 Da so that the 13C mono isotopic peptide mass overlaps perfectly well with the first mono-isotopic mass of the 12C acetylated peptides. More details are needed to allow the reader sufficient insight into the experiment and its results.

Minor concerns:

One minor concern is that the ubiquitination of SOX9 was detected with mass spectrometry wherein samples were treated with DTT to reduce Cysteines to sulfhydryl moieties that then were modified with iodoacetamide according to the reference in the methods section pointing to the initial expanded description of the 4D Proteomics experiment. However, iodoacetamide treatment of proteins has been shown to result in 2 acetoamidoacetamide adducts on lysine sites mimicking the di-glycine mass modification of an ubiquitinylation with a subsequent endoproteolytic digestion with trypsin (Nat Methods 5, 459–460 (2008). <https://doi.org/10.1038/nmeth0608-459>). Can the authors clarify how they confirmed the ubiquitination sites (other than mutation) or be more specific on the sample preparation protocol used for detecting the ubiquitination sites?

p8I320f 'aspirate' -> 'aspartate'

move figure 2g to supplemental data

Version 1:

Reviewer comments:

Reviewer #1

(Remarks to the Author)

In this revised manuscript, the authors essentially address all prior concerns regarding experimental design and data interpretation. They have also clarified the novelty of the study. Strengths on the manuscript include high rigor of the studies performed. The in vitro data describing the cellular mechanisms are particularly compelling and well executed. Likewise, human translational data are included within the work, supporting the negative impact of fatty acid accumulation on joint

health. Delivery of a FAO inhibitor to the joint also prevented OA development in a preclinical model. Prior noted weaknesses of the study have been addressed (e.g., details of genetic controls for experiments, examination of synovium/meniscus).

Reviewer #2

(Remarks to the Author)

The authors have carefully addressed all questions and comments. The manuscript is now acceptable for publication.

Reviewer #3

(Remarks to the Author)

The authors responded to the raised questions and outstanding issues by performing several additional experiments and adapting the manuscript accordingly. I do not have any additional concerns.

Reviewer #4

(Remarks to the Author)

The authors have address all reviewer's questions with sufficient data and comprehensive explanations. Therefore, this manuscript can be accepted in this journal.

Reviewer #5

(Remarks to the Author)

Authors have addressed this reviewer's concerns in the revised manuscript.

Reviewer #6

(Remarks to the Author)

REVIEWER COMMENTS

Reviewer #1 (Remarks to the Author):

Overall Consideration:

In the manuscript, "Fatty acids-mediated acetylation drives osteoarthritis via fatty acid oxidation-induced SOX9 degradation and epigenetic regulation" the authors show that high fat diet feeding leads to increased fatty acid levels in articular cartilage, acetyl-CoA accumulation, and exacerbates development of OA. The fatty acid oxidation enzyme HADHA was identified as differentially regulated following high fat diet feeding. The authors then perform genetic studies to assess the role of HADHA during OA development and progression. These studies are complemented with in vitro work demonstrating that fatty acid oxidation leads to proteasome mediated degradation of Sox9. Strengths on the manuscript include high rigor of the studies performed. The in vitro data describing the cellular mechanisms are particularly compelling and well executed. Likewise, human translational data are included within the work, supporting the negative impact of fatty acid accumulation on joint health. Delivery of a FAO inhibitor to the joint also prevented OA development in a preclinical model. Weaknesses of the manuscript include oversights of technical tissues within the genetic studies. The negative impact of long chain saturated fatty acids on joint health, the role of Sox9 in FAO by chondrocytes, and epigenetic alterations of chondrocyte transcription by acetyl-CoA accumulation are likewise already described within the literature.

Response: We would like to sincerely thank you for your thoughtful and constructive feedback on our manuscript. We appreciate your recognition of the rigor in our studies, especially the in vitro data and human translational results, as well as the potential therapeutic implications of FAO inhibition in OA.

Regarding the concerns with the genetic studies, we have now clarified the limitations of the Agc-CreER^{T2} and Col2-CreER^{T2} systems, providing additional context and references to justify our approach. Your feedback has helped us strengthen this aspect of our study.

In terms of the innovation of our findings, we have provided a detailed clarification to better highlight how our study provides new insights and differentiates itself from previous research, especially concerning the role of FAO in regulating SOX9 protein stability and influencing OA progression. A detailed explanation can be found in our response to your major Comment #2.

Once again, thank you for your insightful comments, which have greatly contributed to the improvement of our manuscript. We believe the revisions have effectively addressed your concerns, and we hope the updated version meets your expectations.

Major comments:

1. The main concern with the manuscript is the choice of controls for the genetic experiments using the Col2-CreERT and the Agc-CreERT. The Agc-CreERT mouse has a basal phenotype induced by generation of an Agc hypomorphic allele after tamoxifen administration (PMID 28921880). In the work described within the manuscript, the authors use HADHAfl/fl Cre- animals for comparison. Col2-CreERT is thought to be only active during development. At the very least, the authors need to acknowledge these limitations.

Response: Thank you for pointing out this important concern. We acknowledge the potential limitations in our genetic experiments using the Agc-CreERT and Col2-CreERT systems.

Regarding Agc-CreERT, we were aware of the basal phenotype induced by Agc-Cre+/+, which results in dwarfism as noted in previous study (PMID 28921880), while which also shows Agc-Cre+/- mice exhibit no overt skeletal phenotype. In our current study, we used Hadha fl/fl;Agc-Cre+/- mice crossed with Hadha fl/fl;Agc-Cre-/- mice. The littermates could be either Hadha fl/fl; Agc-Cre+/- mice or Hadha fl/fl; Agc-Cre-/- mice, but it won't include Hadha fl/fl; Agc-Cre+/+ mice. We then injected tamoxifen at 8 weeks of age to induce gene deletion. We believe this strategy minimizes the limitations of the Agc-Cre system while maintaining rigorous controls.

As for Col2-CreERT system, we acknowledged that its activity is predominantly during the active developmental stage. According to a study by its inventor, Professor Di Chen, this system can achieve targeted gene knockout in articular cartilage by administering tamoxifen to mice between 2 to 8 weeks of age, with earlier administration better ensuring gene recombination efficiency (PMID: 25340803). Therefore, in this study, we chose to inject tamoxifen at 4 weeks of age, a time when the joints are about fully developed, to maximize recombination efficiency without significantly impacting joint development.

We greatly appreciate your concern, as it has prompted us to clarify the experimental design in greater detail. We have added the information discussed above and cited the two referenced studies in the METHOD section. The revised sections have been highlighted in red for your convenience.

2. The mechanistic data showing that FAO alters SOX9 stability are well developed, but may represent an incremental advancement.?

Response: Thank you for your insightful feedback. Besides this concern, we have also taken note of your overall consideration regarding the innovation of our study, particularly your comments on the negative impact of long-chain saturated fatty acids on joint health, the role of SOX9 in FAO by chondrocytes, and the epigenetic alterations of chondrocyte transcription due to acetyl-CoA accumulation. We would like to clarify how our research differentiates itself

from previous studies and highlight the innovative aspects:

1. **Specific Mechanistic Insights:** While existing literature has addressed the broader implications of long-chain saturated fatty acids on joint health (PMID: 38052778), the precise mechanisms, including how fatty acids accumulate in cartilage and trigger osteoarthritis (OA) remain unclear and limited. Our study delves deeper into the specific mechanisms by which fatty acid metabolism, particularly FAO, affects OA pathophysiology. We demonstrate that FAO modulates SOX9 stability through AMPK activity and identify SOX9 as a potential substrate protein for AMPK for the first time. Additionally, we find that FAO induces epigenetic alterations via H3K27ac, leading to the transcriptional activation of the matrix-degrading enzymes *Mmp13* and *Adamts7*. Furthermore, our study identifies a novel mechanism of fatty acid uptake from the synovial fluid as a potential source of intracellular lipids in chondrocytes, in contrast to *de novo* lipogenesis, which has already been preliminarily reported (PMID: 34987154). We have addressed this information in the second paragraph of the INTRODUCTION and in the first to third and fifth paragraphs of the DISCUSSION.

2. **Contextualizing FAO's Role in Mature Chondrocytes:** Although SOX9's role in regulating FAO during chondrogenesis has been documented in an important research (PMID: 32103177), which described a phenomenon during development process, wherein SOX9 functions as a regulator of cellular metabolism by inhibiting FAO. This regulatory mechanism enables skeletal progenitor cells to adapt to lipid scarcity and an avascular environment, ultimately leading to their differentiation into chondrocytes. Our study was actually designed based this finding, which led us to explore a new question: What happens if mature chondrocytes need to elevate their FAO levels under pathological conditions? Our data indicates that in mature chondrocytes, fatty acid overload induces escalating FAO, which accelerates SOX9 degradation and OA progression, which may reciprocally further enhance FAO in chondrocytes, potentially establishing a vicious circle given the similarity between chondrocytes and skeletal progenitor cells. We sincerely acknowledge that this represents an incremental advancement. However, this finding is still significant, as it provides a new perspective on the role of FAO in mature articular cartilage during OA progression, especially in the context of obesity. We have addressed this information in the second paragraphs of the DISCUSSION.

3. **Specific role of FAO-derived acetyl-CoA in epigenetically modulating OA phenotype:** We acknowledge that the relationship between acetyl-CoA and transcriptional regulation in chondrocytes has been explored (PMID: 32470321), with the key finding of that study showing that glutamine-derived acetyl-CoA epigenetically regulates chondrogenic gene expression, particularly *Acan* and *Col2*, through histone acetylation during bone development. We have

discussed and cited this important study in the fifth paragraph of our DISCUSSION, which has been highlighted in red. While our study provides a new perspective in the pathological context of obesity-related osteoarthritis, demonstrating that FFA-derived acetyl-CoA epigenetically regulates cartilage ECM metabolism, particularly through the transcriptional activation of matrix-degrading enzymes *Mmp13* and *Adamts7*, contributing to the progression of OA. Moreover, our data showed that FFA-derived acetyl-CoA stimulated Pdk4 expression in chondrocytes. PDK4 is well recognized as an important metabolic regulator that shifts metabolism from glucose oxidation to FAO, and modulates the balance between lipid synthesis and utilization. This finding further suggests how chondrocytes adapt to lipid stress through FAO-induced epigenetic modulation in the context of obesity. Recently, PDK4 was found to contribute to the development of the senescence-associated secretory phenotype via modulating lactate metabolism (PMID: 37903887), establishing a potential connection between PDK4 and age-related degenerative diseases such as OA. Therefore, our findings also provide a novel direction for future research. We have addressed this information in the fifth paragraph of the DISCUSSION.

By addressing these points, we hope to clarify the innovative contributions of our study to the existing body of literature.

3. Are there changes within the synovium or to the meniscus conferred by HADHA deletion and/or FAO inhibition?

Response: Thank you for your valuable comment. To assess histological changes within the synovium, we performed H&E staining and evaluated synovitis using Krenn's synovitis score (PMID: 12092767). The results showed that while DMM surgery slightly exacerbated synovial inflammation, *Hadha* deletion did not affect synovial inflammation in the knee joint. Similarly, FAO inhibition via CAP-RBC-Exo/TMZ treatment also had minimal impact on synovial tissue in the knee joint. We have included these results in the revised **Figure S5j** and **Figure S10f**.

For the histological evaluation of the meniscus, we assessed Meniscus scores (PMID: 26585241) using Safranin O/Fast Green-stained sections. Since DMM surgery structurally impacts the meniscus, leading to considerable systematic error in evaluating the meniscus post-DMM surgery. Therefore, we restricted the Meniscus assessment to the Sham group, where the results indicated that cartilage-specific *Hadha* deletion had minimal impact on the histological structure of the meniscus. These findings have been included in the revised Figure S5k.

Although our histological evaluation of the synovium and meniscus suggests that *Hadha* deletion or FAO inhibition have limited effects on these tissues, your comments are highly significant. The exploration of this question underscores the possibility that FAO in chondrocytes may play a more critical role in OA pathogenesis. We greatly appreciate your constructive feedback, which has provided valuable insights for this study.

Minor Comments:

1. Typo, line 611: "tow" should be two; double check line 838: "1Xa loading buffer"

Response: Thank you for pointing out these typographical errors. We have corrected "tow" to "two" in line 611 and revised "1Xa loading buffer" to "1× loading buffer" in line 838 to ensure accuracy. These changes have been highlighted in red. We appreciate your careful review.

2. The order of the pictures showing the histology for the DMM experiments and the bar graphs summarizing the scoring is not consistent. This caused some confusion while reading. I would suggest keeping the order of presentation the same – group either like genotypes or like surgeries.

Response: Thank you for pointing out the inconsistency between the order of the histology images and the corresponding bar graphs. We have adjusted the order of all the images to align with their respective bar graphs, ensuring a consistent presentation. We appreciate your valuable feedback, which has significantly enhanced the clarity of our data presentation.

Reviewer #2 (Remarks to the Author):

In this manuscript, Mei et al. elucidate how obesity and osteoarthritic factors synergistically drive increased fatty acid uptake by chondrocytes, promoting fatty acid oxidation (FAO) and resulting in acetyl-CoA accumulation and alteration of protein acetylation profiles. This process leads to the suppression of AMPK signaling and facilitates SOX9 degradation and osteoarthritis (OA). In addition, the authors further explored the therapeutic potential by targeted delivery of trimetazidine, an FAO inhibitor and AMPK activator, in OA treatment. Overall, the study presents a substantial amount of data to address an important question in the OA field. To improve the quality of this manuscript, I have several comments and suggestions listed below.

Response: We would like to sincerely thank you for your thoughtful and constructive feedback on our manuscript. We greatly appreciate your recognition of the significant contributions this study makes to the field of osteoarthritis and the potential therapeutic implications of FAO inhibition in OA treatment.

In response to your comments and suggestions, we have addressed each point carefully and made several important revisions to improve the quality and clarity of the manuscript. We believe the revisions have effectively addressed your concerns and enhanced the overall quality of the manuscript. We hope the updated version meets your expectations. Once again, we thank you for your insightful and constructive feedback, which has greatly contributed to the improvement of our manuscript.

1) Previous studies have demonstrated the significance of metabolism-mediated acetylation in hepatic cells, enhanced OA pathology by fatty acid synthesis in aging chondrocytes, and increased FAO accelerating SOX9 degradation, what is the novel information provided by the current study.

Response: Thank you for your valuable comment. We acknowledge the significance of studies related to metabolism-mediated acetylation in hepatic cells (PMID: 20167786), enhanced OA pathology by fatty acid synthesis (PMID: 37640697; PMID: 34987154), and the role of SOX9 in FAO by chondrocytes (PMID: 32103177). However, compared to this existing body of literature, we believe our study offers further discoveries and innovative aspects. We summarize these as follows:

1. Acetylation in Chondrocytes: Acetylation is indeed widely studied in the fields of OA and chondrocyte biology, with most studies focusing on acetyltransferases and deacetylases. Zhao et al. demonstrated that the concentration of metabolic fuels significantly influences the acetylation status of metabolic enzymes in hepatocytes, which in turn plays a major role in metabolic regulation (PMID: 20167786). Our study not only confirms that this metabolism-mediated acetylation regulatory mechanism also exists in

chondrocytes but also, importantly, provides the first comprehensive landscape of acetylation in osteoarthritic chondrocytes **under high-fat stress**, not only PTM but also epigenetic. These alterations in acetylation not only regulate chondrocyte metabolism but also mediate changes in various biological processes through epigenetic regulation. These findings further broaden the understanding of the role of metabolic abnormalities in the mechanisms underlying OA.

2. Fatty Acid Utilization and OA: The hypothesis that fatty acid synthesis in senescent chondrocytes may exacerbate OA progression has received preliminary support from several studies (PMID: 34987154), including contributions from our group (PMID: 37640697). **However, these studies have not clearly elucidated why fatty acid overload in chondrocytes exacerbates OA or the underlying mechanisms.** Our current research advances this understanding by revealing that chondrocyte fatty acid utilization (FAO) is a key factor in exacerbating OA. By targeting this process, we observed that cartilage-targeted delivery of trimetazidine, an agent widely used in ischemic heart disease treatment, showed superior efficacy in a mouse model of obesity-associated OA, suggesting potential for clinical translation. Additionally, our current study identifies a novel mechanism of fatty acid uptake from the synovial fluid as a potential source of intracellular lipids in chondrocytes, which may have more profound implications in the obesity-related OA subtype.

3. FAO and SOX9: While previous studies have highlighted the critical role of SOX9 as a metabolic regulator that inhibits FAO during bone development to maintain chondrocyte identity (PMID: 32103177), our study uncovers a distinct mechanism in a different biological context. Specifically, we are the first to demonstrate that in mature chondrocytes, increased FAO accelerates SOX9 degradation through AMPK-mediated phosphorylation-ubiquitination crosstalk. This finding provides new insights into OA pathogenesis and highlights the dynamic role of SOX9 in chondrocyte metabolism.

We believe these new aspects enhance the understanding of metabolic regulation in OA and complement the findings of previous studies.

2)The authors concluded that free fatty acid (FFA) is an important metabolite contributing to obesity-associated cartilage degeneration (Fig. 1b-1c, s1b). To further support this conclusion, in vivo experiments are needed to demonstrate similar cartilage changes in normal diet (ND) mice treated with FFA arthrocentesis.

Response: Thank you for your valuable comment. To further support our interpretation, we conducted additional in vivo experiments. 4-week-old C57BL/6J wild-type mice were fed a normal diet. At 8 weeks of age, their left and right knees were randomly assigned to the BSA or FFA group and received

weekly intra-articular injections of 10 μ L of 5mM FFA or BSA control. At 20 weeks of age, the mice were sacrificed, and knee joint samples were collected for histological evaluation as previously described.

The results demonstrated that mice treated with FFA exhibited significantly greater cartilage degeneration compared to the BSA group. This finding provides more direct evidence that FFA is a critical metabolite driving obesity-associated cartilage degeneration and aligns with our *in vitro* findings in Fig. 1c. We have included these results in the revised **Figure S1n**. Thank you again for your constructive suggestion, which has strengthened the robustness of our conclusion.

3) Figure S1l uses the WOMAC score to show that obese patients have more severe clinical symptoms compared to lean patients, reflecting the correlation between obesity and OA severity. However, this is not directly related to the main objectives of the study, and other parts of the research do not address pain assessment. Therefore, I would suggest the authors to remove this data.

Response: Thank you for your thoughtful suggestion. We agree that the WOMAC score data presented in Figure S1l, while demonstrating the correlation between obesity and OA severity, is not directly aligned with the main objectives of our study. As the other parts of our research do not focus on pain assessment, we have removed this data from the revised manuscript. We appreciate your insight on this matter.

4) Figure 4m showed that TRIM9 knockdown effectively elevated SOX9 protein levels. However, it is not known if the similar results could be obtained under high-fat diet (HFD) stimulation?

Response: Thank you for your insightful comment. We have indeed observed that TRIM9 knockdown effectively reverses the high fat-induced downregulation of SOX9. This data has now been included in the revised **Figure 4m**. We appreciate your suggestion, which has helped clarify this point in the manuscript.

5) Figure 5e shows a significant decrease in H3K27ac at 3 peaks in the *Acan* gene locus, but only 2 of them are confirmed by CUT&Tag-qPCR as shown in Fig. S6c. A similar issue is also related to the *Col2a1*, *Pdk4*, and *Mmp13* gene loci. What is the reason for this discrepancy? Additionally, gene symbols in Fig. 5e should be italicized, and this change should be applied consistently throughout the manuscript.

Response: Thank you for your thoughtful comment. The discrepancy arises primarily from technical limitations. According to the manufacturer's instructions (Vazyme Biotech; Cat#TD904-02), half of the sample is reserved for sequencing, while the other half is used for CUT&Tag-qPCR validation. However, due to the limited sample volume (approximately 15 μ L), we had to prioritize selecting the most relevant loci for validation. Additionally, for certain loci, primer design posed challenges, making it difficult to include all peaks in the qPCR analysis. Thus, these considerations led us to make adjustments in our experimental design. We have also italicized the gene symbols in Figure 5e and applied this consistently throughout the manuscript, as you suggested.

6) The authors stated that fatty acid accumulation and OA manifestations could be more pronounced in the HFD-Sham group with prolonged HFD feeding. Actually, previous reports have shown that a 30-weeks HFD is sufficient to promote cartilage degeneration and intracellular lipid deposition in mice (J Orthop Res. 2022, PMID: 35279877).

Response: Thank you for pointing this out. We agree with your observation and have now further cited the findings from the study you mentioned into the discussion section of the revised manuscript. The relevant text has been added and highlighted in red for clarity. We appreciate your suggestion, which has helped enhance our discussion.

7) The authors need to explain why cholesterol levels remained relatively stable among the groups (line 489-490)?

Response: We sincerely appreciate your valuable comments. Given the significant impact of cholesterol in OA progression (PMID: 30728500), it is essential to apply extra effort and care in analyzing our observations to uncover the underlying relationship between cholesterol metabolism and other lipid metabolic processes. With this acknowledgement, we carefully reviewed our existing data and identified potential indications that obesity-related OA may

involve alternative lipid metabolic pathways, particularly in the context of excessive fatty acid accumulation, in addition to cholesterol accumulation. First, our proteomic analysis (Figure S3h) shows that cholesterol efflux-related proteins, such as APOA4 and APOE, were upregulated in cartilage from mice subjected to HFD+DMM compared to ND+DMM. We further validated the upregulation of these proteins *in vitro*, where FFA were used to stimulate chondrocytes. Moreover, our CUT&Tag-sequencing data revealed that H3K27ac around genes related to cholesterol influx (*Apob*), cholesterol synthesis (*Srebf1*, *Srebf2*) are significantly reduced upon FFA stimulation, suggesting downregulation of these genes, which we confirmed at the mRNA level (Figure 5h). We have also included additional data in the revised manuscript showing that protein levels of SREBF1, SREBF2, and APOB were downregulated in chondrocytes upon FFA stimulation. These data have now been included in the revised Figure S3i and Figure S9f.

Additionally, we measured free and total cholesterol levels in previously collected synovial fluid samples. The data showed no significant changes in cholesterol levels, either with increasing OA severity or in the obese group. This suggests that in obesity-related OA, cholesterol level changes in synovial fluid may lag behind alterations in free fatty acids. These data have now been included in the revised Figure S2d.

These collected findings suggest that, in the context of excessive fatty acid accumulation, chondrocytes may favor a state of increased cholesterol efflux

while inhibiting cholesterol influx and synthesis. This shift may partially explain why cholesterol levels in the cartilage of mice remained stable in our HFD-DMM model. However, given that obesity itself is a highly complex condition. Future research efforts should focus on elucidating how chondrocytes modulate the balance between fatty acid metabolism and cholesterol metabolism and whether each pathway predominates in specific subtypes of obesity-related OA. It is crucial to clearly distinguish these subtypes during clinical practice to implement more precise therapeutic strategies. We have included this discussion in the "Limitations of Study" section of the revised manuscript, located in the penultimate paragraph of the Discussion.

8) Why did the authors sacrifice Hadha and Hadhb knockout mice 12 weeks after DMM surgery, but sacrifice Hadha K728R knockin mice 8 weeks after DMM surgery. What is the rationale for this experimental design?

Response: Thank you for your insightful comment. In our experimental design, we intentionally extended the DMM period to 12 weeks for the Hadha and Hadhb knockout mice to better assess the protective effects of FAO inhibition on OA progression. For the *Hadha* K728R CKI mice, as indicated by our data in Figure 1i the HFD-DMM model typically leads to significant arthritis (OARSI score ~5-6) 8 weeks post-DMM surgery. This duration was sufficient to observe the effects of the knockin mutation. Additionally, to ensure the welfare of the experimental animals, we opted to collect samples at 8 weeks post-DMM surgery.

9) TMZ specifically inhibits the TFP complex, and CAP-RBC-Exo/TMZ is expected to alleviate cartilage destruction in HFD-DMM-induced OA mice. However, similar efficacy was also observed in ND-DMM mice (Fig. 6b-d). The authors need to explain this finding.

Response: Thank you for thoughtful comment. As indicated by prior studies (PMID: 36971866; PMID: 26892523) and supported by our data (Figure S7d), TMZ, in addition to inhibiting FAO, also acts as an AMPK activator. AMPK plays a significant role in protecting against OA, whether through the AMPK-SOX9 axis we identified or other pathways (**Ann Rheum Dis. 2020**, PMID: 32156705), We have included this information in the third paragraph of the discussion and cited the relevant study. Another possible explanation is that DMM surgery enhances fatty acid uptake by chondrocytes, thereby activating FAO to some extent. In this scenario, using TMZ to inhibit FAO could also provide protection against OA, even in ND-DMM mice. A similar situation was observed in previous research by Choi et al. (PMID: 30728500), where inhibition of cholesterol metabolism through the CH25H-CYP7B1-ROR α axis yielded protective effects in DMM-induced OA models in mice on a normal diet. Therefore, it is reasonable that TMZ would also offer protective effects in the ND-DMM model.

10) Why is human blood used for RBC-Exo generation, instead of mouse blood, given that it is applied to a mouse model?

Response: Thank you for your insightful comment. Our approach of using human blood for RBC-Exo generation is based on prior research (PMID: 37343564), where exosomes derived from human blood were successfully used to treat obesity-induced inflammation in mice. While mouse blood could have been used, our focus on human blood-derived exosomes aligns with the goal of developing translational therapies. Red blood cells from human donors are widely available, and their long-established safety profile further supports the strong potential for clinical application of this method.

11) All H3K27ac peaks (chromosomal location) and assigned gene should be provided as supplemental data.

Response: Thank you for comment. We have provided all H3K27ac peaks, including their chromosomal locations and assigned genes, in "Supplementary Data 4. CUT&Tag-sequencing data" for your reference. We appreciate your suggestion, which has enhanced the transparency and accessibility of our data.

12) All differential expression protein data and all normalized differentially acetylated proteins and sites data should be provided as supplemental data.

Response: Thank you for comment. We have included all differential expression protein data as well as all normalized differentially acetylated proteins and sites data in "Supplementary Data 2. Proteomics data." and "Supplementary Data 3. Acetylome data." for your reference. We appreciate your suggestion, which has improved the comprehensiveness and accessibility of our dataset.

13) Line 32: "obesity-associated OA" should be changed to "metabolism-associated post-traumatic OA" to ensure consistency with the discussion section.

Response: Thank you for your helpful suggestion. We have revised the term "obesity-associated OA" to "metabolism-associated post-traumatic OA" in the revised manuscript to ensure consistency. The changes have been highlighted in red for your reference.

14) The right panel of Fig. S2I needs adjustment as the x-axis appears to be misaligned.

Response: Thank you for your careful observation. We have corrected the misalignment in the x-axis of the right panel in revised Figure S3m, which has been included in the updated manuscript. We appreciate your attention to this detail.

15) A recent publication by Wei et al. "Risk of metabolic abnormalities in osteoarthritis: a new perspective to understand its pathological mechanisms"

Bone Res. 2023 Dec 6;11(1):63. is closely related to the current study and need to be cited.

Response: Thank you for bringing this to our attention. We agree that the study by Wei et al. (PMID: 38052778) provides a comprehensive review of recent advances in the relationship between metabolic abnormalities and OA pathology, especially regarding fatty acid metabolism, which forms an important basis for our own research. At the time of writing the previous version of our manuscript, this study had not yet been published, and we apologize for not citing it earlier. We have now referenced this work in both the introduction and discussion sections of the revised manuscript. Your suggestion has undoubtedly strengthened the content of this manuscript, and we appreciate your valuable input.

Reviewer #3 (Remarks to the Author):

The authors report that increased fatty acid oxidation in articular chondrocytes disturbs their energy status leading to SOX9 degradation and that the increased acetyl-CoA levels also epigenetically enhance catabolic pathways. These novel findings certainly increase our insight in the role of diet on the development of osteoarthritis. The strength of the manuscript is that both in vivo and in vitro models are used to substantiate the conclusions. One aspect that is less corroborated is the (causative) link between fatty acid supply, cellular energy status and SOX9 degradation.

Response: We sincerely thank you for your thoughtful and constructive feedback. Your recognition of our study's contributions to the understanding of diet's impact on osteoarthritis development is greatly appreciated. We also value your acknowledgment of our use of both in vivo and in vitro models to support our conclusions.

Regarding your concern about the causative link between fatty acid supply, cellular energy status, and SOX9 degradation, we have made several important revisions and additions to further clarify and strengthen our data. Specifically, we have provided additional experimental evidence to demonstrate that FAO capacity is increased under high-FFA concentration. We have also added new data to demonstrate that FAO in chondrocytes indeed alters energy status, which in turn affects AMPK phosphorylation and its activity. We further clarified how FAO-induced de-phosphorylation of SOX9 impacts its stability and the role of AMPK activity in this process. Additionally, we provide evidence that FFA regulates SOX9 stability through FAO, as FAO-block effectively reverses FFA-induced SOX9 degradation. Moreover, we have included additional validation of our experiments, including controls for FFA toxicity and quantification of Western blot data, and have reassessed our immunohistological analysis to ensure accuracy and reproducibility.

We believe these revisions address your concerns and significantly enhance the clarity and strength of the manuscript. We hope the updated version meets your expectations, and we once again thank you for your valuable feedback, which has greatly contributed to the improvement of our study.

The authors report that adding FFA increases FAO and OXPHOS (Figure S2k): basal OCR is increased by FFA, but this is not blocked by Etomoxir, indicating that the FFA-induced increase in OCR is not due to CPT-1 inhibition and questioning the contribution of FAO in basal condition. Etomoxir has only an effect during maximal respiration in the presence of FFA, suggesting that FAO is used in this condition. Prove should be provided that FAO-linked OXPHOS is increased after FFA supply. The same comment applies to the HADHA-K728 mutant.

Response: Thank you for your insightful comment. We agree with your

observation that, although FFA-induced increases in basal OCR were not completely blocked by etomoxir, etomoxir did reduce basal OCR to some extent in the FFA-treated group. This is a phenomenon often observed in other studies, as inhibition of one metabolic pathway can often be compensated by alternative pathways under basal conditions. However, during maximal respiration induced by FCCP, other pathways are unable to compensate for the inhibition of FAO, making maximal respiration a more reliable indicator, as it reduces the compensatory effects of other pathways. This explains why we observed that etomoxir had only a minor effect on basal OCR but significantly suppressed OCR under maximal respiration conditions, as shown in Figure s2k.

To further support this hypothesis, we conducted additional mitochondrial fuel oxidation analysis. Briefly, cells were pre-treated with FFA or BSA for 2 days, followed by incubation in XF DMEM containing 25 mM glucose, 1 mM pyruvate, and 2 mM L-glutamine for one hour at 37°C without CO₂. OCR measurements were taken with the sequential addition of 2 μM UK5099, 3 μM BPTES, and 4 μM Eto as indicated. The results show that FFA pre-treatment upregulated OCR under basal conditions, and the addition of UK5099 and BPTES, which inhibited compensation by glucose and glutamine oxidation, allowed etomoxir to more effectively suppress OCR in the basal condition, particularly in the FFA pre-treatment group. These findings confirm that FFA pre-treatment indeed enhances FAO capacity in chondrocytes. We have included these additional results in revised Figure S3n to further support the conclusion that FAO-linked OXPHOS is increased after FFA pre-treatment.

Similarly, for the HADHA K728R mutant cells, we used the same approach to compare the FAO capacity between HADHA WT and HADHA K728R cells, both were pretreated with 300μM FFA for two days. The results showed that HADHA WT cells exhibited a higher FAO capacity after FFA treatment, whereas HADHA K728R cells were resisted to FFA-induced increase in FAO. This suggests that acetylation at the K728 site of HADHA is essential for its enzymatic activity. We have added these results in revised Figure 2g.

We hope the additional data provided above sufficiently address your concern. Thank you again for your valuable feedback.

The authors highly focus on AMPK activation to explain the regulation of SOX9. Although some of the data report changes in pAMPK, other data only show an effect on p-ACC, which is regulated by pAMPK but also by several other parameters including citrate, long chain FFA and hormones. Therefore, an effect on p-ACC levels is not similar to energy status, but likely reflects an effect on de novo lipogenesis. Either the data should be interpreted in this way or levels of pAMPK, reflecting energy status, should be provided for the different conditions tested.

Response: Thank you for your insightful feedback. We agree that p-ACC can be influenced by various factors beyond AMPK activation. To strengthen our interpretation and better support the role of AMPK in regulating SOX9, we have supplemented the data in revised Figure S7c, S7d, and S7g with p-AMPK levels under the different conditions tested. We believe this addition provides a clearer connection between AMPK activation and the FAO status in chondrocytes, aligning with our overall conclusions.

How do the authors explain that the overlap of FFA-induced phosphorylation sites of SOX9 and A-769662 (AMPK)-induced phosphorylation sites is rather low (Fig 4g), as these data indicate that FFA-induced effect on SOX9 levels is different from AMPK/energy status.

Response: Thank you for this insightful comment. We believe this result aligns

well with our hypothesis. Our data suggest that FFA reduce phosphorylation at specific SOX9 sites by inhibiting AMPK activity. In contrast, A-769662, as an AMPK activator, maintains phosphorylation at these sites, stabilizing SOX9 protein. Thus, the low overlap between FFA-induced and A-769662-induced phosphorylation sites supports our conclusion that these two treatments impact SOX9 phosphorylation differently through opposite effects on AMPK activity. We hope this clarifies our interpretation.

To further prove that FFA supply regulates SOX9 levels through FAO, the authors should show that the effect of FFA supply has no effect when *Hadha* gene expression/activity is blocked.

Response: Thank you for your valuable suggestion. Mechanistically, our data in Figure 4f demonstrates that FAO inhibition can effectively reduce the FFA-induced ubiquitin-mediated degradation of SOX9. You may observe a significant increase in SOX9 ubiquitination following FFA stimulation, which is effectively rescued by the FAO inhibitor TMZ, restoring ubiquitination levels to near-baseline.

To further support this conclusion, we performed additional experiments using FFA stimulation in mouse chondrocytes, combined with TMZ treatment or siRNA knockdown of *Hadha/Hadhb* to inhibit FAO. These interventions successfully rescued FFA-induced SOX9 protein downregulation. We have included these additional results in Figure S8b of the revised manuscript. Thank you again for your valuable feedback.

The authors report that adding free fatty acids (FFA) changes chondrocyte functioning (Figure 1a-c), but the concentrations of FFA used (between 100-500 μ M) are rather high as concentrations higher than 200 μ M can become lipotoxic for some cell types. The authors should therefore provide evidence that the used FFA concentrations do not induce toxicity and that the observed effects are specific. In addition, Western Blots should be quantified, not only in Figure 1 but in all the experiments.

Response: Thank you very much for your valuable comments. We designed the concentration gradient of FFA for treating mouse chondrocytes based on the following rationale. The concentration of fatty acids in plasma is 0.3-0.9 mM in human (clinical reference values) and 0.5-1.5 mM in mouse (PMID:

35124446). Based on our detection of free fatty acids in human synovial fluid (Figure 1m), we found that the concentration of fatty acids in synovial fluid is comparable to that in plasma. Therefore, we hypothesize that fatty acids levels in mouse synovial fluid may be slightly higher than in humans. To further support this design, we conducted additional cell viability assays and observed that FFA concentrations between 100-500 μM did not result in significant cell death or lipotoxicity in both primary mouse chondrocytes and C28/I2 cells. We have incorporated the viability data for primary mouse chondrocytes into **Figure S1c** in the revised manuscript. Thus, we believe these concentrations are well tolerated by chondrocytes and appropriately reflect pathophysiological conditions.

We have also quantified the Western blots across all experiments as recommended. The detailed quantification data can be found in the **“Source Data. Original scans of Western blot.”** file. For experiments where quantification is not applicable or necessary, such as Co-IP assays and ubiquitination analyses, we have provided images from three independent replicates in the same file for reference. Thank you again for your valuable feedback.

Immunohistological analysis (Col2, MMP13, ACC,..) forms an important part of the study, but no information is provided on how the intensity is quantified and whether also the abundance of the signal was quantified. Related to the latter aspect, it is rather surprising that Col2 staining of the articular cartilage is very weak, detected only at certain spots and often intracellular (nuclear), also in the control condition (Sham, normal diet). Normally you should expect positive staining of almost the entire extracellular matrix. Staining should be reassessed, as this information is important for the conclusions.

Response: Thank you for your valuable comment. We acknowledge the importance of immunohistological analysis in this study. To quantify the intensity of the immunofluorescence staining, we used ImageJ software, following standard protocols for cartilage analysis. The key steps for this analysis are as follows:

- Image acquisition: Images were captured under consistent exposure settings to ensure comparability.
- Image processing: Images were converted to 8-bit grayscale for the relevant channel, with background subtraction applied to minimize

noise.

- ROI selection: The region of interest (ROI) was manually outlined for each sample by experienced pathologists, ensuring consistency across comparable samples whenever possible.
- Signal thresholds: Positive staining areas within the total ROI were quantified by setting appropriate intensity thresholds to assess the abundance of positive staining.
- Intensity quantification: Mean fluorescence intensity was measured within the selected ROI after determining the signal thresholds.

These steps ensured that the analysis was both objective and reproducible. We have included a detailed description of this quantification process in the revised manuscript, which is highlighted in red for your convenience.

Regarding the weak Col2a1 staining in the cartilage matrix, we appreciate your careful observation and for pointing out this important issue. We have carefully reassessed our staining procedure and found that the previous results were likely due to an insufficient antibody concentration and decreased antibody efficacy from prolonged storage. To resolve this, we re-performed the Col2a1 staining using a fresh batch of antibody and adjusted the antibody concentration, ensuring that all other conditions were optimized.

As a result, we obtained stronger and more widespread staining of the extracellular matrix, as seen in the revised figures (Figure 3i, S11, 6b). The extracellular matrix is now visibly stained in the healthy cartilage (Sham group). As expected, the extracellular matrix staining in OA cartilage shows varying degrees of reduction depending on the severity of OA, which aligns with prior studies. Furthermore, we also conducted the quantification as described above, and the results still support our original conclusions.

We greatly appreciate your close attention to this detail and hope this clarification, along with the updated data and detailed quantification could address your concern. Thank you again for your constructive feedback

The authors demonstrate that adding FFA results in increased number of lipid droplets and increased gene expression of Lipe and Cd36 and conclude that FFA uptake is increased (Fig 1d and Suppl fig 1c). However, these findings do not prove undeniably that FFA uptake is increased, as lipid droplets can be formed by many mechanisms.

Response: Thank you for your insightful comment. We agree with you that lipid droplets can be formed through various mechanisms. In fact, a previous study by our group demonstrated that de novo lipogenesis is increased in senescent chondrocytes, which could also contribute to intracellular lipid droplet accumulation (PMID: 37640697). However, in the current study, we focus on how chondrocytes sense and respond to lipid overload in joint microenvironment, such as synovial fluid. Considering that obesity is a major risk factor for OA and that our experiments in Figure 1m have demonstrated elevated FFA levels in the synovial fluid of obese patients, we interpreted that a high-lipid environment induces chondrocytes to express proteins associated with FFA transport, thereby enhancing FFA uptake.

To further substantiate this hypothesis, we conducted additional experiments to directly monitor FFA uptake by chondrocytes. The result demonstrated that FFA-pretreated chondrocytes exhibit an enhanced capacity for FFA uptake, which have been included in the revised **Figure 1d**. This finding at least suggests that, under obese and high-lipid conditions, FFA uptake by chondrocytes plays a significant role in intracellular lipid accumulation. Moreover, this observation raises critical scientific questions regarding the metabolic changes triggered by lipid accumulation in chondrocytes and their subsequent impact on OA progression, which constitutes a key focus of the later sections of our study.

We hope this explanation provides clarity regarding our experimental design and adequately addresses your concern. Thank you once again for your constructive feedback.

The lipidomic analysis shows increased FA in the HFD-CCM (Fig S1a). Are these FA found in triglycerides (lipid droplets) or more as phospholipids or other types of lipids which do not function as storage but rather as membrane component, signaling or energy source.

Response: Thank you for your valuable comment. To further address your question, we conducted a more detailed analysis of the lipidomic data. We found that these increased FAs in the HFD-DMM group are primarily enriched in corresponding glycerolipids (GLs), indicating that these FAs are involved in lipid droplet formation. However, when examining other lipid classes, such as phospholipids, we did not observe a similar upregulation pattern like FAs or GLs. Therefore, we conclude that these FAs mainly function as an energy source in this context. That said, we do not rule out the possibility that specific lipids may also be regulated in obesity-related osteoarthritic chondrocytes, and their potential roles warrant further explored in future studies. We have included the changes in GLs in the revised Figure S1j, and the data for other lipid classes have been uploaded to "Supplementary Data 1. Lipidomic data.". Thank you once again for your insightful feedback, which has allowed us to clarify this aspect of our study.

Minor

No information is provided on which ^{13}C -Fatty Acids are used in the in vivo model (Fig 2a).

Response: Thank you for your thorough review and valuable comment. The ^{13}C -fatty acids were purchased from Cambridge Isotope Laboratories (Cat# CLM-8455-PK). We have included this information in Supplementary Table 8. Key resources table, and highlighted it in red. The full product details can be found on the CIL website using this catalog information.

Fatty Acid composition for:

CLM-8455 MIXED FATTY ACIDS (^{13}C , 98%+) PR-24965

Relative Area	Fatty Acid	Carbon Number	Sites of Unsaturation
0.2%	Myristic	14	0
9.4%	Palmitoleic	16	1
38.9%	Palmitic	16	0
0.3%	Margaric	17	0
10.7%	Linoleic	18	2
26.9%	Oleic	18	1
1.6%	Elaidic	18	1
1.6%	Stearic	18	0

Validation of gene deletion and levels of overexpression of *Hadha* or *Hadhb* should be provided, both for the in vitro and in vivo models.

Response: Thank you for your valuable comment. We have now provided validation for the gene deletion and overexpression levels of *Hadha* or *Hadhb* in both in vitro and in vivo models.

For the in vitro experiments, we validated the gene knockdown/knockout and overexpression of *Hadha* or *Hadhb* by Western blotting using specific antibodies. The results are included in the corresponding Western blot images in revised **Figures 3a, 3f, 3g, 4e, 4n, S5a, S5h, and S7g.**

For evaluating chondrocyte secretion function, including Micromass culture and 3D-agarose culture, we assessed the mRNA expression levels of *Hadha* and *Hadhb* by RT-qPCR to confirm shRNA-mediated knockdown. The corresponding validation data are included in revised **Figures 3b and S5b.**

For the in vivo models, we performed immunohistochemistry to detect the expression of *Hadha* and *Hadhb* in joint cartilage and statistically compared the percentage of positive cells to verify gene deletion efficiency. The results are shown in revised **Figures 3h and S6a-S6d.**

Thank you again for your kind reminder.

Reviewer #4 (Remarks to the Author):

The manuscript describes the correlation between fatty acid and the change in metabolic pathway (acetylation, fatty acid oxidation, etc) that induces the severity of osteoarthritis. After careful consideration, there are some issues which authors can consider:

Response: We sincerely thank you for your thoughtful and constructive feedback, which has been extremely helpful in improving our study. We have made several revisions, including additional data to better explain the regulation of metabolites such as FFA and cholesterol in chondrocyte, particularly within the inherently complex context of obesity. We have also clarified the novel insights from our study and provided further details on our experimental design, along with correcting some errors. Detailed responses to your specific comments can be found in the following sections. We sincerely hope that these revisions address your concerns and that the updated manuscript meets your expectations.

1. The role of some compounds, such as acetyl-CoA, free fatty acid, cholesterol, etc, have been widely reported in various scientific literatures, including the detail mechanism of those pathways. Thus, what is the crucial aspect of this manuscript compared to others? Does this manuscript provide new information regarding OA pathogenesis or progression?

Response: Thank you for your insightful comment. We agree that the roles of acetyl-CoA, free fatty acids, and cholesterol in OA have been reported extensively. However, we believe our study contributes further discoveries and novel insights:

1. Research has demonstrated that acetyl-CoA accumulation can promote OA progression (PMID: 37908734; 34987154). In our study, we identified that the acetyl-CoA level within chondrocytes can be modulated by metabolic fuels, specifically FFAs. We show that FFA-derived acetyl-CoA significantly impacts the global acetylation profile in chondrocytes. Importantly, our work provides the first comprehensive landscape of acetylation alterations in osteoarthritic chondrocytes under high-fat stress. These alterations in acetylation not only regulate chondrocyte metabolism but also mediate changes in various biological processes through epigenetic regulation. These findings further broaden the understanding of the role of metabolic abnormalities in the mechanisms underlying OA.

2. Although different FFAs have been shown to impact OA variably (summarized in our response to your Comment #3), and preliminary evidence links fatty acid synthesis in senescent chondrocytes to OA progression (PMID: 34987154; 37640697), the mechanisms by which FFA overload exacerbates OA remain unclear. Our study advances this understanding by showing that fatty acid utilization (FAO) in chondrocytes is a key factor in OA progression. By targeting

this process, we found that cartilage-targeted delivery of trimetazidine, an agent commonly used in ischemic heart disease, had a significant effect in a mouse model of obesity-associated OA, suggesting promising potential for clinical translation. Additionally, we identify a novel mechanism of fatty acid uptake from the synovial fluid as a source of intracellular lipids in chondrocytes, which may have more profound implications in the obesity-related OA subtype.

3. We sincerely acknowledge the significance cholesterol metabolism in OA (PMID: 30728500; 24280247). However, evidence on targeting cholesterol metabolism to prevent OA is inconsistent (PMID: 30144308; 24280247). In our current study, although OA severity induced by HFD was comparable, cholesterol levels remained relatively stable. These disparities may arise from the different experimental models (which has been discussed in our response to your comment #2), but also underscore the complexity of obesity-related metabolic effects. Based on the experimental findings, we shifted our focus from cholesterol metabolism to an alternative possibility and discovered that chondrocyte FAO play a crucial role in OA pathophysiology. These new findings suggest that future research efforts should focus on elucidating how chondrocytes balance fatty acid and cholesterol metabolism, as specific pathways may predominate in distinct subtypes of obesity-related OA, emphasizing the need for tailored therapeutic strategies in clinical practice.

By addressing these points, we hope to clarify the innovative contributions of our study to the existing body of literature. Moving beyond correlative studies to mechanistic validation, our work provides a deeper understanding of the molecular underpinnings of OA and lays the foundation for targeted metabolic interventions. We appreciate your feedback, which has helped us better articulate the significance of our study.

2. The authors stated that cholesterol levels remained stable in OA induced by HFD. However, it is known that the elevation of cholesterol level plays a crucial role in OA. How would you explain your findings regarding this issue?

Response: We sincerely appreciate your valuable comments. Given the significant impact of cholesterol in OA progression (PMID: 30728500), it is essential to apply extra effort and care in analyzing our observations to uncover the underlying relationship between cholesterol metabolism and other lipid metabolic processes. With this acknowledgement, we carefully reviewed our existing data and identified potential indications that obesity-related OA may involve alternative lipid metabolic pathways, particularly in the context of excessive fatty acid accumulation, in addition to cholesterol accumulation. First, our proteomic analysis (Figure S3h) shows that cholesterol efflux-related proteins, such as APOA4 and APOE, were upregulated in cartilage from mice subjected to HFD+DMM compared to ND+DMM. We further validated the upregulation of these proteins *in vitro*, where FFA were used to stimulate

chondrocytes. Moreover, our CUT&Tag-sequencing data revealed that H3K27ac around genes related to cholesterol influx (*Apob*), cholesterol synthesis (*Srebf1*, *Srebf2*) are significantly reduced upon FFA stimulation, suggesting downregulation of these genes, which we confirmed at the mRNA level (Figure 5h). We have also included additional data in the revised manuscript showing that protein levels of SREBF1, SREBF2, and APOB were downregulated in chondrocytes upon FFA stimulation. These data have now been included in the revised Figure S3i and Figure S9f.

Additionally, we measured free and total cholesterol levels in previously collected synovial fluid samples. The data showed no significant changes in cholesterol levels, either with increasing OA severity or in the obese group. This suggests that in obesity-related OA, cholesterol level changes in synovial fluid may lag behind alterations in free fatty acids. These data have now been included in the revised Figure S2d.

These collected findings suggest that, in the context of excessive fatty acid accumulation, chondrocytes may favor a state of increased cholesterol efflux while inhibiting cholesterol influx and synthesis. This shift may partially explain why cholesterol levels in the cartilage of mice remained stable in our HFD-DMM model. More importantly, different experimental models have been used to focus on cholesterol metabolism in chondrocytes (such as the high cholesterol diet-DMM model). Given that obesity itself is a highly complex condition, future research efforts should focus on elucidating how chondrocytes modulate the balance between fatty acid metabolism and cholesterol metabolism and

whether each pathway predominates in specific subtypes of obesity-related OA. It is crucial to clearly distinguish these subtypes during clinical practice to implement more precise therapeutic strategies. We have included this discussion in the "Limitations of Study" section of the revised manuscript, located in the penultimate paragraph of the Discussion.

3. Why did the authors use OA:PA ratio of 2:1 for FFA solutions? Also, is there any difference in FFA-induced OA if the type/chain number of FFA is different?

Response: We deeply appreciate the insightful feedback. The choice of an OA ratio of 2:1 for our FFA solutions was guided by prior studies in liver and adipocyte research (PMID: 33318171, PMID: 36746922) as well as osteoarthritis-related research (PMID: 33378044). This ratio is commonly used to study fatty acids due to OA and PA being the most prevalent free fatty acids in body, especially in obesity contexts. Both OA and PA are fundamental components of animal-derived fats and typically exist in a 2:1 molar ratio in daily diets. Their significance in obesity and metabolic disorders influenced our selection of this formulation for our study.

Regarding the impact of different types or chain lengths of FFAs, fatty acids can be classified into saturated fatty acids (SFAs), monounsaturated fatty acids (MUFAs), and polyunsaturated fatty acids (PUFAs), based on the length of their carbon chain and the number of double bonds. SFAs are generally associated with osteoarthritis exacerbation, and research has shown that their osteoarthritis-inducing effects tend to intensify with longer chain lengths (PMID: 28418007). The relationship between MUFAs and OA, however, remains more complex and debated. For example, a clinical study found higher MUFA levels in the synovial fluid of OA patients compared to non-OA individuals (PMID: 30203669), while a rabbit OA model showed a decrease in MUFAs within the knee joint (PMID: 30885225). In our preliminary experiments, PA, the most abundant SFA in vivo, significantly induced an osteoarthritis-like phenotype in chondrocytes, while stearic acid (SA), with a longer carbon chain, exhibited similar effects even at lower concentrations. However, OA, the most abundant MUFA, increased MMP13 expression at higher stimulation concentrations, yet also increased SOX9 expression, indicating a potentially dual role in osteoarthritis. These findings are consistent with prior research but are not the primary focus of our current study. Therefore, we did not include these results in the manuscript.

As for PUFAs, they serve as precursors to various bioactive compounds like

eicosanoids, resulting in complex and diverse functions (PMID: 38052778). Since this study focused on exploring the general relationship between chondrocyte FAO and OA, we specifically avoided the use of PUFAs to maintain a clearer interpretation of the FAO process.

4. Figure 1f: it is unclear if the mice subjected to ND and HFD diet was carried out in weeks or months. The authors should put that information in the figure itself or in the caption. Please recheck also the text as the authors said that the "mice subjected to a 16-week HFD diet."

Response: Thank you for your thorough review and valuable comments. As described in the Methods section, the mice were switched to the experimental diets at 4 weeks of age and were sacrificed at 20 weeks, meaning they were subjected to 16 weeks of HFD feeding. We have now clarified this information in revised **Figure 1f**, as well as in **Figures S3a and S10a** for consistency.

5. The data shown in the manuscript is too much for one figure. Please reselect the main data and put some data in the supporting information. Also, please consider the font size or clarity of the font in the figures. For example, p values can be just marked by different asterisk, and the actual values can be mentioned in the captions or text.

Response: Thank for your helpful comment. In response, we have moved some of the data to the supporting information to streamline the presentation of the main findings. As a result, the supplemental figures have increased from 8 to 10, and the content in the main data section has been made clearer and more concise. Additionally, we have replaced the p values with asterisks in the figures for clarity. We hope these adjustments improve the readability and clarity of the manuscript. Thank you again for your valuable feedback.

6. The resolution of the figures is low. Please ensure the clarity of the figures.

Response: Thank for your attention to the clarity of the figures. We have uploaded the highest resolution and largest file size versions permitted by the submission system. However, the files available in the "Zip of files for Reviewer" folder may appear blurry due to additional compression by the system, as illustrated in the example below. We appreciate your understanding of this situation, and we will continue to ensure high-resolution images in future versions.

Before compression↑

After compression↑

7. There are many typos in the manuscript. Please revise it.

Response: Thanks for your careful checks and valuable comments. We have carefully rechecked the manuscript for any typographical errors and corrected them. Additionally, prior to submission, the manuscript was professionally edited to ensure the quality of the language.

Reviewer #5 (Remarks to the Author):

This study by Mei et al. investigates the role of fatty acid oxidation (FAO) in OA. The authors report considerable in vitro results implicating the mechanisms of fatty acid induce FAO mediates acetylation of HADHA, which modifies the AMPK activity leading to degradation of SOX 9 and increases production of cartilage matrix degradation enzymes. They confirm their finding in vivo using tissue-specific KO of Hadha, Hadhb, and Hadha k728R conditional knock in mice. Finally, target delivery of Trimetazidine (TMZ) in the joint to alleviate the OA. Overall, this is a comprehensive and well-designed study that presents a large body of work. However, there are concerns.

Response: We deeply appreciate your thoughtful and detailed feedback on our manuscript. Your insights have been invaluable in enhancing the clarity and rigor of our study. We are grateful for your recognition of the comprehensive and well-designed nature of our research. In response to your concerns, we have made comprehensive revisions to the manuscript, which can be found in the following sections. We believe these revisions thoroughly address your concerns, and we hope that the updated manuscript meets your expectations.

Major concern

1.OA caused by obesity/HFD is different from OA caused by injury. Combining the two is not a good model and could have exacerbated the finding. Fig. 1 clearly showed that HFD+ DMM accumulates lipid in joints compared to HFD+sham and ND+sham.

Response: We deeply appreciate your insightful feedback. We designed the combination of HFD with DMM based on specific reasoning. As shown in Figure 1h, HFD alone induced only a mild OA phenotype and mild lipid accumulation in cartilage, a finding consistent with a previous study (PMID: 28418007). This outcome may be due to the relatively short 16-week HFD feeding period. We anticipate that with prolonged HFD feeding, the severity of OA and lipid accumulation in the HFD-Sham group would become more pronounced. Indeed, another study (PMID: 35279877) using a 30-week feeding model demonstrated that diet-induced obesity significantly exacerbates cartilage degeneration and intracellular lipid deposition. However, such long-term feeding experiments could be time-consuming. Due to the timeline considerations and to more clearly detect the impact of HFD on OA, we have chosen to incorporate the DMM model. Through stringent HFD-DMM vs. ND-DMM comparison, the effect of obesity/HFD on OA progression could also be demonstrated effectively. Such design was also influenced by a significant prior study (PMID: 30728500), which utilized the high cholesterol diet (HCD)-DMM model as a significant metabolic-related post-traumatic OA model in scientific research. We appreciate your understanding of our rationale for this design, and we have included this discussion in the "Limitations of Study" section of the revised manuscript, located in the penultimate paragraph of the Discussion.

2. Does loss of *Hadha*^{-/-} prevent mice from developing OA on HFD?

Response: Thank you for your valuable comment. In our previous experiments (Figure 3i and Figure 6b), we found that inhibition of *Hadha* alleviated OA progression in the HFD-DMM model, partially addressing this question. However, your comment highlights the importance of assessing whether loss of *Hadha* specifically prevents OA development under HFD-sham conditions.

To address this, we conducted additional *in vivo* experiments. 4-week-old *Hadha*^{fl/fl} mice and *Hadha*^{fl/fl}; Acan-CreER^{T2} mice were fed an HFD. At 8 weeks of age, tamoxifen injections were administered to induce conditional knockout of the target gene. At 12 weeks, mice were randomized into DMM and sham groups, undergoing DMM surgery or sham operations, respectively. At 20 weeks, mice were sacrificed, and knee samples were collected for histological evaluation as previously described.

The results demonstrated that cartilage-specific deletion of *Hadha* not only alleviated OA progression in the HFD-DMM group, consistent with our previous findings, but also effectively prevented OA development in the HFD-sham group. This new evidence supports our overall conclusion. We have included these results in revised Figure S6d. Thank you again for your constructive suggestion, which has significantly strengthened the robustness of our conclusion.

3. It is not clear why only palmitate was chosen when other pro-inflammatory FFAs are known to play roles in obesity/HFD-derived pathways.

Response: We sincerely appreciate your thoughtful question. In this study, the choice of an OA ratio of 2:1 for our FFA solutions was guided by prior studies in liver and adipocyte research (PMID: 33318171, PMID: 36746922) as well as osteoarthritis-related research (PMID: 33378044). This ratio is widely applied in fatty acid research, as OA and PA are the most abundant free fatty acids in body, especially in obesity contexts. They are essential components of animal-derived fats and generally present in a 2:1 molar ratio in typical diets.

While other pro-inflammatory fatty acids, such as stearic acid (SA) and n-6 polyunsaturated fatty acids (PUFAs), also play roles in obesity-related pathways, our focus on PA stemmed from its prominence and well-characterized effects on cellular metabolism. Our preliminary experiments demonstrated that PA

significantly induced an OA-like phenotype in chondrocytes. Though SA, with its longer carbon chain, showed similar effects at lower concentrations, aligning with findings from an *in vivo* study (PMID: 28418007), PA was chosen for its consistent representation in animal models and human diets, particularly under obesity and HFD-related conditions. These data are consistent with prior research but are not the primary focus of our current study. Therefore, we did not include these results in the manuscript.

Additionally, this study primarily aimed to explore the general relationship between FAO and OA progression in chondrocytes. Given the complexity of PUFAs, which serve as precursors to bioactive compounds like eicosanoids (PMID: 38052778), we excluded them to maintain focus on the role of FAO and OA progression without the additional variables associated with PUFA metabolites.

4. Fig. 2. C14 labeling data was presented for ND+DMM and HFD+DMM. What about ND and HFD alone?

Response: Thank you for your insightful comment. We further analyzed ¹³C-acetyl-CoA levels in the cartilage of ND+Sham and HFD+Sham mice, and this data has been added to revised Figure S3b. Although HFD alone led to a slight increase in ¹³C-acetyl-CoA accumulation in cartilage, the difference was not statistically significant. This finding is consistent with our observations in Figure 1, where HFD alone caused only mild FFA accumulation in cartilage, underscoring the more pronounced effect of combining HFD with the DMM model. This rationale supports our focus on the HFD+DMM vs. ND+DMM

comparison in subsequent analyses. We appreciate your understanding.

5. Figure 3 invitro studies, cells were not treated with FFA?

Response: Thank you for your careful attention to this issue. We apologize for

not providing a clear explanation earlier. In fact, the in vitro studies presented in Figure 3, including Figures 3a, c-d, and f-g, as well as Figures S5a, c-e, and h, involved pre-treatment of the cells with 200 μ M FFA. We have now clarified this in the revised figure legends, which are highlighted in red. We apologize again for not specifying this in the original manuscript.

Other concerns

How long were the cells fed with FFA in Fig. 1 and elsewhere?

Response: Thank you for your question. For the RNA analysis, protein detection, and CUT&Tag assay, cells were harvested after 48 hours of FFA treatment. For proteolysis investigation and ubiquitination-immunoprecipitation, cells were treated with FFA for 12 hours prior to harvest. For the assessment of chondrocyte secretory function, including micromass culture and 3D agarose culture, cells were treated with FFA for one week. This information has been clarified in the revised manuscript within the Methods section. We appreciate your attention to these specifics.

Fig. 2, panel D, is misrepresented.

Response: Thank you for your valuable feedback. We aimed to present a heatmap in Figure 2d highlighting the significantly regulated acetylated lysine sites on proteins involved in lipid metabolic processes in the cartilage of HFD-DMM mice compared to ND-DMM mice. This approach was inspired by a previous study (PMID: 31577934, Figure 6G). However, as you pointed out, the numbers at the bottom of the heatmap were indeed confusing and did not clearly convey our intention. These numbers actually represent the acetylated lysine sites detected on the corresponding proteins. In the revised figure, we have made adjustments by annotating "acetylated lysine sites" to the far left of these values and added a "K" in front of each number to indicate lysine. We hope these changes make the figure clearer and more accurately convey the information we intended. Thank you again for your helpful suggestion.

Fig. 2 panel H: How was HADHA k728 acetylation detected in cartilage tissue? HADHA-specific antibodies were generated? For in vitro studies, the authors mentioned that they used general K728 antibodies. Is this true for tissues too?

Response: Thank you for your insightful question. In this study, we used two antibodies to detect acetylation in HADHA. The general anti-acetyl lysine

antibody (purchased from Abcam) was used to assess overall HADHA acetylation and is labeled as "Kac" in Figure 2e. For specific detection of acetylation at the HADHA K728 site, we used a HADHA-K728ac-specific antibody, labeled as "HADHA K728ac" in Figure 2H. This HADHA-K728ac-specific antibody was custom-prepared by Huaan Biotechnology (Hangzhou, China). We apologize for not detailing this process in the original Methods section. To clarify, we have now included the following antibody preparation details in the revised Methods section, highlighted in red:

Anti-HADHA-K728ac antibody preparation

HADHA - K728ac - specific antibody was prepared by Huaan Biotechnology (Hangzhou, China). The HADHA-K728ac peptide (CFVDLYGAQK[ac]IVDRL) was first coupled to keyhole limpet hemocyanin and BSA, followed by multiple subcutaneous injections to immunize New Zealand white rabbits. Blood was then collected, and the presence of HADHA-K728ac-specific antibodies in the serum was evaluated using enzyme-linked immunosorbent assay (ELISA) and western blot. For final antibody purification, the serum was eluted through the HADHA-K728ac peptide column and then passed through a non-HADHA-K728ac peptide column to ensure specificity.

After antibody purification, we assessed its specificity and efficacy via Western blot and ELISA. As illustrated in the figure below. Equal amounts of wild-type HADHA protein and the K728R mutant were enriched and subjected to Western blot analysis using HADHA-K728ac-specific antibodies. Compared to the wild-type HADHA, the signal for the K728R mutant was significantly reduce.

As shown in the table below, the antibody demonstrated strong affinity for the acetylated peptide within a dilution range of 1:1000–1:16000, with minimal binding to the non-acetylated peptide, further confirming its high specificity and efficacy.

The purified antibody titer					
Coating component	CFVDLYGAQK [ac]IVDRL or CFVDLYGAQK IVDRL	Coating volume	1µg/ml* 50µl	Coating condition	4°C, overnight
Blocking buffer	1%BSA-PBST	Blocking volume	150µl	Blocking condition	37°C, 1h
Primary antibody	Gradient dilution sample	Volume	50µl	Incubation condition	37°C, 30min
Second antibody	Goat Anti Rabbit IgG- HRP	Volume	1/30000 *50µl	Incubation condition	37°C, 45min
Color rendering condition	37°C, 10min				
Instruction	IgG- CFVDLYGAQK[ac]IVDRL		IgG- CFVDLYGAQKIVDRL		
Rabbit No.	RB2596				
1:250	2.199	2.238	0.748	0.790	
1:1000	2.334	2.259	0.349	0.369	
1:4000	2.195	2.072	0.167	0.182	
1:16000	1.600	1.497	0.070	0.077	
1:64000	0.736	0.764	0.027	0.026	
1:256000	0.281	0.283	0.012	0.014	
1:1024000	0.105	0.096	0.011	0.010	
1%BSA	0.010	0.008	0.008	0.007	

Additionally, as shown in Figures 3i, the rate of HADHA K728ac-positive cells in cartilage tissue is significantly reduced in HADHA K728R CKI mice, further supporting the antibody's effectiveness in vivo. The antibody preparation and validation protocol we followed is well-established and widely used in scientific research (e.g., PMID: 31409767, 36374166, 35932761, 36891678). Due to the maturity of this method and its routine application, we did not include the western blot and ELISA results in the manuscript, as they do not impact the study's main conclusions. We hope this clarification addresses your concerns regarding the suitability of this antibody for both in vitro and in vivo detection. Thank you again for prompting this clarification.

Fig. 3 and in other figures. What are lane 1 and 2?

Response: Thank you for your valuable feedback. The numbers in Fig. 3a (and other figures) correspond to siRNA identifiers or mouse cartilage sample numbers. Your observation has indeed highlighted the need for a clearer and more standardized labeling approach. In response, we have updated the figure labels in the revised manuscript to consistently use "#1," "#2," "#3," following a more conventional format. Thank you again for your insightful suggestion.

Page 8 line 2. upregulated SOX9 protein expression. Word expression is inappropriate here.

Response: Thank you for your insightful comment. We understand your concern regarding the phrasing "upregulated SOX9 protein expression." To improve clarity and accuracy, we have revised this to "significantly increased SOX9 protein levels" in the manuscript. Thank you again for your helpful suggestion.

Figure panels should have capital alphabets. For example, Figure 2I should be Figure 2L. And also, the panel should follow an order. It is confusing.

Response: Thank you for your valuable feedback. In most articles published in *Nature Communications*, figure panels are typically labeled with lowercase alphabets. To align with the journal's standard practices, we have retained the lowercase alphabets for the panel labels. But we have bolded them to enhance clarity. Also, we have adjusted the order of the figure panels to ensure a more organized and clear presentation. Thank you again for your helpful suggestion.

Reviewer #6 (Remarks to the Author):

The manuscript "Fatty acids-mediated acetylation drives osteoarthritis via fatty acid oxidation-induced SOX9 degradation and epigenetic regulation" is well written and contains a wealth of interesting data and results. It is easy to follow and results are presented in a clear and insightful way.

Response: We deeply appreciate your thoughtful and detailed review of our manuscript. Your feedback has greatly contributed to enhancing the clarity and rigor of our study. We are grateful for your recognition of the comprehensive nature of our research and the clarity with which the results were presented.

In response to your concerns, we have made the necessary revisions to the manuscript, which we believe address all the issues raised. These changes include the inclusion of updated data for the ubiquitination sites of SOX9, a more detailed description of the UPLC-MS/MS analysis conditions, and clarification of how to distinguish between ¹³C- and ¹²C- acetyl-CoA. We have also made several adjustments in the experimental methods, particularly on how to accurately identify ubiquitination sites, which we greatly appreciate your thoughtful reminder on.

Your feedback has significantly improved the quality and robustness of our work, and we believe the updated manuscript now meets the standards you have outlined. We hope the revised version meets your expectations, and we thank you again for your valuable input.

Major concerns:

Figure 5g shows only the phosphorylation sites but does not indicate the results for the ubiquitination sites. Please include the data for the ubiquitination of the lysine sites.

Response: Thank you for your constructive feedback. In our experiment shown in Figure 4f, we observed that FFA treatment significantly upregulated the ubiquitin-mediated degradation of SOX9 in chondrocytes, and this effect was blocked by the AMPK activator A-769662 and the FAO inhibitor TMZ. Following these findings, we attempted to identify the specific ubiquitination sites of SOX9 via mass spectrometry. However, our initial experiments showed no substantial differences in the identified ubiquitination sites among the groups.

Initially, we hypothesized that the FAO-AMPK axis might primarily regulate the poly-ubiquitination process, explaining why FFA increased the overall ubiquitination levels of SOX9 protein without changes in the specific ubiquitination sites. Consequently, we did not include data for specific ubiquitination sites in the initial manuscript, as it did not affect our primary conclusions.

However, as highlighted in your minor concern #1, we realized that the initial results might have been influenced by iodoacetamide-related artifacts mimicking ubiquitination, which interfered with our analysis. Guided by your valuable feedback, we optimized our methodology and reanalyzed the ubiquitination sites, replacing iodoacetamide with chloroacetamide to eliminate potential artifacts.

The updated analysis revealed that FFA treatment significantly increased the number of ubiquitination sites on SOX9 compared to the control and A-769662-treated groups. Specifically, sites such as K62, K100, K166, K180, K183, K242, and K249 were identified as potential contributors to the FAO-AMPK axis-mediated enhancement of SOX9 ubiquitin degradation. These findings further support our previous conclusions. We have now included the detailed ubiquitination site data in revised Figure S8d for reference.

Thank you once again for your insightful comment, which has significantly improved the quality and robustness of our work.

The description of the UPLC-MS/MS analysis p24I971ff is too short: What were the MS/MS analysis conditions and how were ¹³C acetylated peptides distinguished from ¹²C acetylated peptides? The mass difference is 1 Da so that the ¹³C mono isotopic peptide mass overlaps perfectly well with the first mono-isotopic mass of the ¹²C acetylated peptides. More details are needed to allow the reader sufficient insight into the experiment and its results.

Response: Thank you for your thorough review and valuable comments. We have added a more comprehensive description of the MS/MS analysis

conditions to the Methods section, highlighted in red.

The LC-MS/MS analysis was performed using a TSQ Quantis™ mass spectrometer (Thermo Fisher Scientific) coupled with a Vanquish UPLC (Thermo Fisher Scientific). LC separation was achieved on an Atlantis Waters XSelect® HSS T3 column (2.1 mm×100 mm, 2.5µm). Sample aliquot was injected with the column temperature maintained at 30 °C, the flow rate was set at 0.3 ml/min. Mobile phase A was 2.5 mM ammonium acetate in H₂O and Mobile phase B was acetonitrile. The elution gradient started with 2 % phase B: 0-5min: 2%-95%B; 5-7min: 95%B; 7-7.1min: 95%-2%B; 7.1-10min: 2%B. The sample manager temperature was set at 4°C. Experiments were conducted in positive ionization mode and quantification was carried out in multiple reaction monitoring mode. Mass spectrometric conditions, including: collision energy and RF lens voltage were optimized automatically. The mass spectrometric conditions for analytes and the internal standard were shown in Supplementary Table 7. The ion source conditions were listed as follows: Spray voltage, 4000 V; the ion transfer tube and the vaporizer temperature, 350 °C; sheath gas and auxiliary gas of nitrogen (purity ≥99.99 %) at the flow rate of 35 arb and 10 arb, respectively. Data were analyzed using the Thermo Scientific Xcalibur workstation (Version 4.5.445.18).

Supplementary Table 7. Multiple reaction monitoring parameters

Compound	Precursor (m/z)	Product (m/z)	CE(V)	RF (V)	Lens	Polarity	IS
Acetyl-CoA	810	303.05	31.1	102		+	4-CPA
¹³ C ₂ -Acetyl-CoA	812	305.05	30.4	143		+	4-CPA
4-CPA	200	154	13	59		+	

To distinguish between ¹³C- and ¹²C-acetylated peptides, we used ¹³C₂-Acetyl-CoA as a standard, as all carbon atoms in the backbone of the ¹³C-FFA were labeled with ¹³C, resulting in acetyl-CoA with two carbon atoms labeled with ¹³C upon metabolism. The molecular weight difference between ¹³C₂-Acetyl-CoA and natural Acetyl-CoA is 2 Da. Although the two metabolites are structurally very similar and cannot be easily separated at the chromatographic level, they can be clearly differentiated during MS analysis, as both their primary and secondary ions differ by 2 Da. Below are the EIC images of ¹³C₂-Acetyl-CoA and Acetyl-CoA, demonstrating that the two metabolites can be distinctly identified in cartilage tissue without mutual interference. This

approach, utilizing differences in mass spectrometry ion pairs to distinguish between structurally similar substances, is commonly employed in the analysis of endogenous metabolites. We hope this additional detail addresses your concerns, and we appreciate your valuable input.

(In standard, Upper: Acetyl-CoA;
Lower: $^{13}\text{C}_2$ -Acetyl-CoA)

(In cartilage sample, Upper: Acetyl-CoA;
Lower: $^{13}\text{C}_2$ -Acetyl-CoA)

Minor concerns:

One minor concern is that the ubiquitination of SOX9 was detected with mass spectrometry wherein samples were treated with DTT to reduce Cysteines to sulfhydryl moieties that then were modified with iodoacetamide according to the reference in the methods section pointing to the initial expanded description of the 4D Proteomics experiment. However, iodoacetamide treatment of proteins has been shown to result in 2 acetoamidoacetamide adducts on lysine sites mimicking the di-glycine mass modification of an ubiquitylation with a subsequent endoproteolytic digestion with trypsin (Nat Methods 5, 459–460 (2008). <https://doi.org/10.1038/nmeth0608-459>). Can the authors clarify how they confirmed the ubiquitination sites (other than mutation) or be more specific on the sample preparation protocol used for detecting the ubiquitination sites?

Response: Thanks for your insightful comment and for bringing this important point to our attention. In our initial experiments, we indeed used iodoacetamide as the alkylating reagent. Iodoacetamide modifies the sulfhydryl groups of cysteine residues but can also react with lysine residues, adding a modification of 114.0429 Da, which is similar to the di-glycine remnant mass left by ubiquitin after trypsin digestion. This potential artifact may explain why our initial experiments did not reveal differences in SOX9 ubiquitination sites among the treatment groups.

Based on your valuable feedback, we re-analyzed SOX9 ubiquitination using mass spectrometry. This time, we replaced iodoacetamide with chloroacetamide as the alkylating reagent, which does not produce di-glycine-like modifications during alkylation, thereby avoiding false-positive

ubiquitination site identifications. The updated result, presented in our response to your major concern #1, supports our initial conclusion. We have also updated the Methods section to describe this revised protocol in detail and cited relevant literature, which was highlighted in red. Thank you again for helping us enhance the rigor and clarity of our work.

p8l320f 'aspirate' -> 'aspartate'

Response: Thanks for your careful review and valuable comments. We have corrected "aspirate" to "aspartate" in both the main text and the figure legends, and highlighted the changes in red.

move figure 2g to supplemental data

Response: Thank you for your suggestion. We have moved Figure 2g to the supplemental **Figure S4a** in the revised manuscript as recommended.